# Dominant contribution of Asgard archaea to eukaryogenesis

Victor Tobiasson[1,2 ✉], Jacob Luo[1], Yuri I. Wolf[1] & Eugene V. Koonin[1 ✉]

The origin of eukaryotes is one of the key problems in evolutionary biology[1,2]. The demonstration that the last eukaryotic common ancestor (LECA) already contained the mitochondrion—an endosymbiotic organelle derived from an alphaproteobacterium—and the discovery of Asgard archaea—the closest archaeal relatives of eukaryotes[3–7]—inform and constrain evolutionary scenarios of eukaryogenesis[8]. We conducted a comprehensive analysis of the origins of core eukaryotic genes tracing to the LECA within a rigorous statistical framework centred around evolutionary hypothesis testing using constrained phylogenetic trees. The results show dominant contributions of Asgard archaea to the origin of most of the conserved eukaryotic functional systems and pathways. A limited contribution from Alphaproteobacteria was identified, relating primarily to energy transformation systems and Fe–S cluster biogenesis, whereas ancestry from other bacterial phyla was scattered across the eukaryotic functional landscape, without clear, consistent trends. These findings imply a model of eukaryogenesis in which key features of eukaryotic cell organization evolved in the Asgard lineage leading to the LECA, followed by the capture of the alphaproteobacterial endosymbiont and augmented by numerous but sporadic horizontal acquisitions of genes from other bacteria both before and after endosymbiosis.

Eukaryotes drastically differ from archaea and bacteria (collectively, prokaryotes) by the complex organization of eukaryotic cells. The signature features of this organizational complexity include the eponymous nucleus, the endomembrane system, the elaborate cytoskeleton and the energy-converting mitochondrion, which evolved from an alphaproteobacterial endosymbiont[9]. Early eukaryogenesis models featured an amitochondrial protoeukaryote that captured and domesticated an alphaproteobacterium at a relatively late stage of evolution[10,11]. However, subsequent research revealed no primary amitochondrial eukaryotes although many have secondarily lost mitochondria[12]. Thus, the last eukaryotic common ancestor (LECA) probably already possessed mitochondria along with the other signatures of the eukaryotic cellular organization[1,2,8]. These findings lend credence to scenarios of eukaryogenesis in which mitochondrial endosymbiosis triggered cellular reorganization, giving rise to eukaryotic cellular complexity[1,2,8,13–15].

Phylogenomic analyses show that the core genes of eukaryotes that can be traced to the LECA are a mix originating from archaea and various bacteria and, in early studies, the bacterial contributions quantitatively exceeded the archaeal ones[1]. The archaea-derived genes were found primarily in information-processing systems (replication, transcription and translation), whereas genes of apparent bacterial origin comprised the operational component of the eukaryotic gene complement, in particular encoding metabolic enzymes[16–19].

The study of eukaryogenesis was transformed by the discovery and exploration of Asgard archaea (currently, phylum Promethearchaeota within kingdom Promethearchaeati[20,21], hereafter Asgard) which includes the closest known archaeal relatives of eukaryotes[3–7]. Phylogenetic analyses of conserved genes have been pushing the eukaryotic branch progressively deeper within the Asgard tree. The latest such analysis identified the order Hodarchaeales, within the Asgard class Heimdallarchaeia, as the likely sister group of eukaryotes[4]. However, with the continued influx of Asgard sequence data, the exact affinity of eukaryotes within Asgard remains a moving target[22,23]. Notably, Asgard archaea encode, express and use homologues of a broad variety of eukaryote signature proteins beyond the core information-processing componentry, in particular, cytoskeletal proteins and proteins involved in membrane remodelling[4,5,7,24,25]. Furthermore, these genes of Asgard archaea have undergone extensive duplication and subfunctionalization pre-LECA, suggesting their prominent role in eukaryogenesis[26].

Many models of eukaryogenesis have been proposed, differing with respect to the timing, topology and origin of the eukaryotic cellular organization and genetic repertoire[1,2,8]. The endosymbiotic origin of the mitochondria from an alphaproteobacterium is indisputable, but the nature of the host of the proto-mitochondrial endosymbiont remains a matter of debate. The most straightforward models posit an Asgard host[2,8,27]. However, a principal conundrum for such scenarios of eukaryogenesis is the chemistry of membrane lipids and the enzymology of their biosynthesis, which are unrelated in archaea and bacteria, with eukaryotes using the bacterial type[28]. Accordingly, any model of eukaryogenesis that includes an archaeal host would require a change in membrane composition. More complex, alternative eukaryogenesis

[1]Division of Intramural Research, Computational Biology Branch, National Library of Medicine, National Institutes of Health, Bethesda, MD, USA. [2]School of Infection and Immunity, University of Glasgow, Glasgow, UK. ✉e-mail: victor.tobiasson@glasgow.ac.uk; koonin@ncbi.nlm.nih.gov

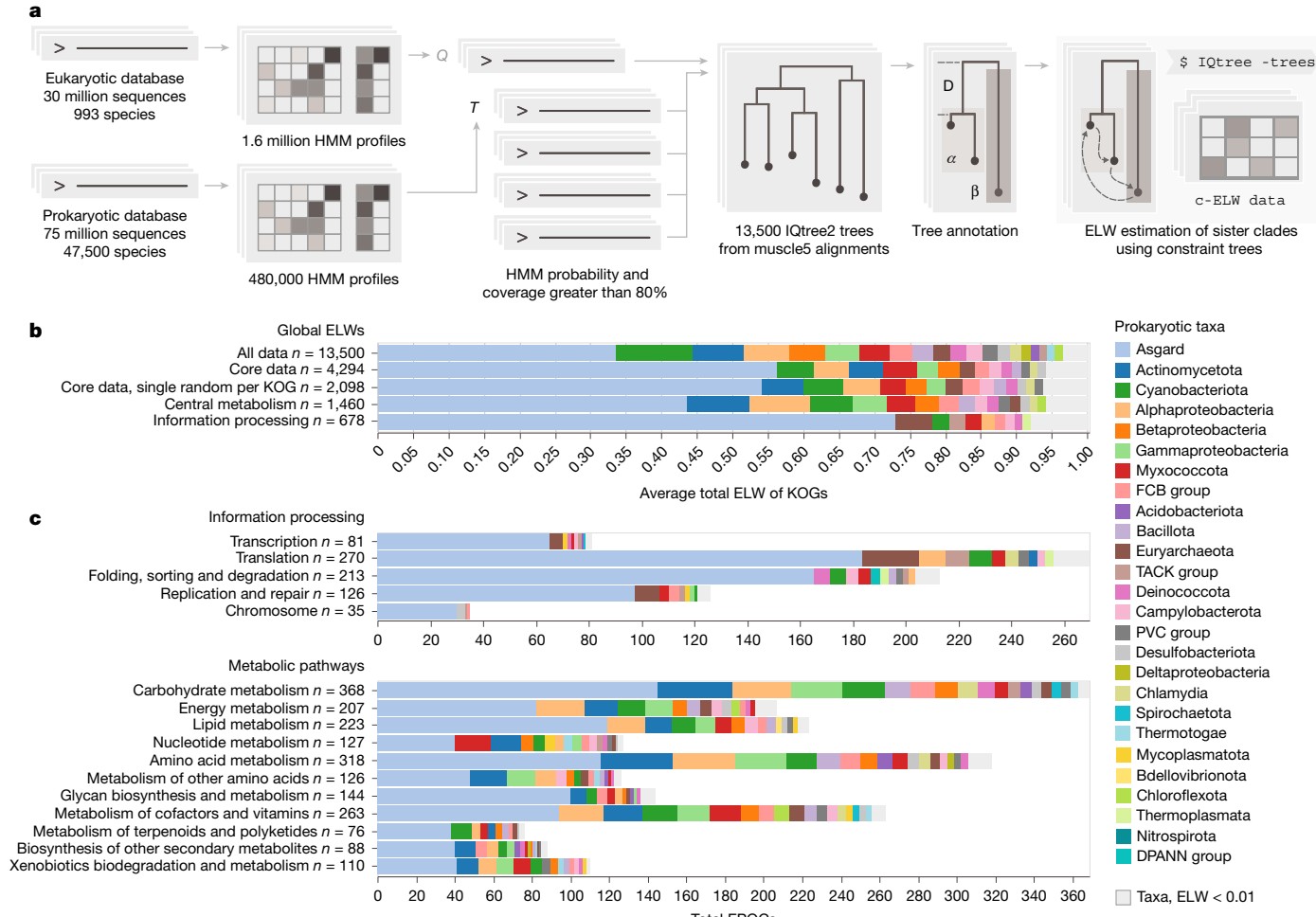

**Fig. 1 | Phylogenetic associations of functional classes of eukaryotic proteins with principal archaeal and bacterial taxa. a**, Simplified EPOC generation and processing pipeline. From left to right, one eukaryotic and one prokaryotic sequence database transformed into HMMs. The query of eukaryotic against prokaryotic HMMs generated 13,500 annotated EPOCs. The detected prokaryotic clades were assessed for their probability to be the closest eukaryotic sister phyla by permutation testing using IQTree2 (Extended Data Fig. 2). **b**, Global aELW for all EPOCs within unfiltered, core and non-paralogous subsets and top-level breakdowns for metabolic enzymes and proteins of core information-processing pathways as defined by the KEGG ontology BRITE at the coarsest A level (KO:09100, KO:09120). Grey markers group prokaryotic taxa with an aELW < 0.01. **c**, Further subdivision of aELW values across more specific functional categories of genes BRITE level B, counting the total number of EPOCs and coloured by relative aELW of taxa per category.

models, underpinned by metabolic symbiosis, postulate two endosymbiotic events whereby an Asgard archaeon was first engulfed by a bacterium, the archaeal membrane being lost in the process, followed by a second endosymbiosis that gave rise to the mitochondria[15,29,30]. Although less parsimonious than binary scenarios, such models of eukaryogenesis account for the continuity of bacterial membranes while requiring retargeting at least part of the archaeal membrane proteome. These evolutionary scenarios appear compatible with the syntrophic lifestyle of at least some Asgard archaea, found in consortia with bacteria, particularly Myxococcota (formerly Deltaproteobacteria)[31,32]. A pivotal study by Pittis and Gabaldon[33], in which the relative timing of events in eukaryogenesis was inferred by comparing stem length from different ancestors in phylogenetic trees of conserved eukaryotic genes, suggested late mitochondrial acquisition, which was apparently preceded by the acquisition of many genes from different bacteria, possibly through one or several earlier endosymbioses[33–35].

We took advantage of the rapidly growing collection of archaeal, bacterial and eukaryotic genome sequences to assess the origins of core eukaryotic genes within a rigorous statistical framework based on evolutionary hypothesis testing using constrained phylogenetic trees. The results reveal a consistent, principal link between Asgard archaea and the origin of most functional classes of eukaryotic genes,

demonstrating a dominant Asgard contribution to eukaryogenesis. These findings are compatible with a scenario of eukaryogenesis where many signature, complex features of eukaryotic cells evolved in the Asgard ancestor of eukaryotes.

## Origins of eukaryotic proteins

To represent prokaryotes, a database of 75 million prokaryotic protein sequences was curated from 47,545 complete prokaryotic genomes obtained from the National Center for Biotechnology Information (NCBI) GenBank in November 2023 (prok2311) and supplemented with proteins extracted from 63 Asgard genome assemblies[4,5]. The initial eukaryotic database consisted of 30 million sequences from 993 species present in EukProt v.3 (ref. 36), cleaned using mmseqs2 (ref. 37) to remove prokaryotic contaminants (Fig. 1a). To accurately infer the origins of genes in the LECA, it is essential to exclude genes present only within narrow subsets of species. These possibly result from post-LECA horizontal gene transfer (HGT) between eukaryotes and prokaryotes, a principal phenomenon given the pervasive cohabitation of eukaryotes with prokaryotes[38–41]. To prevent such genes from affecting our analysis, we reconstructed the 'soft-core' pangenome for each of our assigned prokaryotic and eukaryotic taxonomic classes

(Methods). These pangenomes included only those genes that are present in at least 50% of the species within each class of bacteria and archaea in prok2311 (20.7 million), and at least 20% of the species in eukaryotes (12.7 million), ensuring that our inferences were based only on widespread protein families and would not reflect lineage-specific HGT (Extended Data Fig. 1d).

To identify close connections between prokaryotic and eukaryotic protein families, separate hidden Markov model (HMM) databases for prokaryotes and eukaryotes were constructed using a custom, cascaded, sequence-to-profile clustering pipeline, implemented based on mmseqs2 clustering, followed by a multistep data reduction and multiple sequence alignment (MSA) procedure to generate HMM profiles using diversified muscle5 alignments[42] and HH-suite[43] (Methods). The resulting eukaryotic HMMs were queried against the prokaryotic dataset using HHblits[43] to identify sets of homologous protein sequences. Each eukaryotic cluster sequence and all its significant prokaryotic hits constituted an individual set, hereinafter referred to as a eukaryotic–prokaryotic orthologous cluster (EPOC). Each EPOC contains a unique set of eukaryotic proteins. These EPOCs were used for phylogenetic tree construction, annotation and evolutionary hypothesis testing (Fig. 1a). The final EPOCs include 10.9 million prokaryotic and 1.74 million eukaryotic sequences, mapping to 52% and 14% of the respective non-redundant pangenome datasets.

To infer the most likely prokaryotic ancestry of the eukaryotic proteins given the data in each EPOC, rather than relying on tree topology directly, we used a probabilistic approach for evolutionary hypothesis testing using constraint trees (Fig. 1a). Following the construction of an initial master tree, we carried out further constrained tree calculations, exhaustively sampling all arrangements of the prokaryotic sister clades nearest to the eukaryotic clade(s) and obtaining expected likelihood weights (ELW) for the set of possible sister clade models[44] (Extended Data Fig. 2). Given that the ELW metric is analogous to model selection confidence, here we take it to be proportional to the probability of a sampled prokaryotic clade being the best estimate of the closest sister clade of a eukaryotic clade. For each EPOC, our analysis dynamically accounts for long branch outliers and is capable of resolving eukaryotic paraphyly, treating each resolved eukaryotic clade within an EPOC individually for downstream analysis (Methods). The resulting data included 13,500 EPOCs annotated using profiles generated from KEGG Orthology Groups (KOGs)[45], each with an MSA generated using muscle5 (ref. 42), a maximum likelihood tree inferred using IQtree2 (ref. 46) and associated ELW values for all candidate prokaryotic sister clades. The analysis of prokaryotic ancestry was performed only for those eukaryotic clades that included more than five distinct taxonomic labels, with at least one coming from Amorphea and one from Diaphoretickes—the two expansive eukaryotic clades considered to emit from either the first or the second bifurcation in the evolution of eukaryotes[47,48]. Thus, although we did not attempt to explicitly reconstruct the LECA gene set, these clades represent genes that, given their broad taxonomic distribution of our data, were likely to have had an ancestor in the LECA.

Considering the global distribution of ELW values across all 13,500 EPOCs covering 3,950 unique KOGs, the single greatest average ELW (aELW), here referred to as 'association', was with Asgard archaea (Fig. 1b). Further associations with Cyanobacteria, Actinomycetota, Betaproteobacteria and Alphaproteobacteria, as well as trace associations with many additional bacterial groups, were detected at lower levels. Due to the scope of the analysis, MSA quality and topological diversity of the trees varied across the EPOCs such that some EPOCs showed low ELW values for all clades, indicating that the data in these cases was insufficient to discern their eukaryotic ancestries. Excluding such low-quality EPOCs with a maximum ELW < 0.4 yielded a robust core set of 4,290 EPOCs covering 2,100 KOGs, improving the reliability of the results. This core set covers a wide range of information-processing systems, ubiquitous metabolic pathways and transporters, as well as regulatory and housekeeping proteins. Limiting the analysis to this core subset of well-assigned eukaryotic families with wide taxonomic coverage increased the global Asgard association, which now accounted for 50% of all likelihood weight across more than 4,000 unique data points, covering more than 2,000 unique KOGs (Fig. 1b).

To ensure these results were based on robust sequence clusters, we performed an additional cluster annealing step (Extended Data Fig. 3a; Methods). In brief, starting with initial profiles derived from mmseqs2 clusters we clustered the HMM–HMM search results with a greedy set cover approach. We then further partitioned these super-clusters by constructing trees for each cluster and identifying sets of leaves sharing short pairwise distances, indicating the existence of separate clades. These annealed clades were then used for EPOC construction. Furthermore, we reconstructed our full dataset using the Genome Taxonomy Database (GTDB) taxonomy to verify that the detected eukaryote–prokaryote associations were not affected substantially by our use of the NCBI taxonomy[49]. We note that, although relative contributions from specific taxa varied because of remapping of taxonomic labels (for example, merging of Betaproteobacteria into Gammaproteobacteria), or the stricter annealed EPOC definition limiting the scope of the data, our key observation of a dominant Asgard association for eukaryotes held independent of clustering, taxonomy and eukaryotic scope (Extended Data Figs. 1e, 3b and 10a). As the strong Asgard association was clear in both representations of our data, all subsequent analysis was based on the core EPOC set described above. Finally, although our results suggest the presence of diverse paralogs of apparent Asgard origin in the LECA[5,26,50], paralogy alone cannot account for this broad association of Asgard archaea to eukaryogenesis (Extended Data Figs. 1d and 5).

Averaging ELW scores across EPOCs sharing KEGG ontology supported strong Asgard associations with, among other functional systems and pathways, the ribosome, RNA and DNA polymerases, and Ras-like GTPases, as well as ubiquitin-mediated protein degradation and the proteasome. These findings validated and expanded upon previous observations[2,4,5], extending Asgard associations to large parts of the core metabolic network (Fig. 1b). In contrast, prominent alphaproteobacterial associations were relatively limited, primarily including proteins involved in oxidative phosphorylation and Fe−S clusters biogenesis, with weaker associations across the mitochondrial metabolic pathways. Thus, we observed a far stronger overall association between Asgard archaea and eukaryotes across diverse biological functions and pathways than described previously[4,5] although the association between eukaryotic core metabolism and a wide range of bacterial phyla remained cumulatively substantial. We also observed a consistent association with Cyanobacteria, a known endosymbiont of Archaeplastida and the ancestor of plastids, with a strong impact on the evolution of Eukarya post LECA. However, we note that pathways driving this cyanobacterial association are unrelated to the photosynthetic capabilities of the chloroplast but rather exhibit a broad association with several metabolic pathways, without apparent trends, similar to the observations for other bacterial taxa[51–53] (Extended Data Fig. 7). Thus, we conclude that this cyanobacterial signal most probably stems from pre-LECA events and is distinct from the later contributions from the chloroplast.

## Asgard dominance in eukaryogenesis

In accordance with a dominant contribution of Asgard archaea to eukaryogenesis, we observed consistent and strong associations of Asgard proteins with a wide array of cellular functions. In previous work, the strongest Asgard traces have been noted across the information-processing systems, with unambiguous protein associations with DNA and RNA processing as well as protein expression, trafficking and signalling[2,4,5]. Here we confirmed and extended Asgard associations with genome replication and transcription, and further detected pronounced Asgard traces for nucleotide excision repair,

mismatch repair and homologous recombination. Additional well-known associations, such as ribosomal proteins, were expanded to include translation factors, components of the co-translational insertion machinery, protein targeting and aminoacyl-tRNA biosynthesis (Extended Data Fig. 4b). Thus, all groups of core eukaryotic proteins involved in information processing appear to be almost exclusively of Asgard descent (Fig. 1c).

We further detected Asgard associations extending far beyond the information-processing systems, including prominent contributions of Asgard archaea to the protein machinery involved in nucleocytoplasmic transport as well as downstream protein sorting, glycosylation and targeting. Central components of the endoplasmic-reticulum-associated, N-linked glycan biosynthesis and transfer, including both cytoplasmic and lumenal monoglycosyltransferases, as well as the core of the oligosaccharyltransferase complex, were associated strongly with Asgard (Extended Data Fig. 4a). Enzymes associated with glycosylation maturation in the Golgi complex were not recovered in our core data, possibly because of extensive diversification of domain architecture in eukaryotes, limiting their taxonomic scope and apparent presence in the LECA. The Asgard connections of the eukaryotic glycosylation machinery further included the synthesis of glycosylphosphatidylinositol (GPI)-anchors, which tether targeted proteins to the membrane post-translationally[54], here detected as unambiguously Asgard-derived. We also detected an Asgard origin of the seven-subunit (UDP-GlcNAc)-transferring (GPI-GnT)-monoglycosyltransferase complex responsible for initiating GPI-anchor synthesis and components required for the maturation of the GPI-anchor, as well as the transamidase complex and factors responsible for transfer of the protein onto the mature GPI-anchor (Extended Data Fig. 4b).

Of principal importance to eukaryogenesis is the provenance of the pathways for the biosynthesis of bacterial-type lipids, given that all binary symbiogenesis scenarios with an archaeal host require a transition from archaeal to bacterial lipids in the membranes[8]. Although Asgard associations were observed for large parts of the overall metabolic network, the pathways associated with fatty acid synthesis and decay showed a wide distribution of aELW values, indicating a higher-than-average degree of mosaicism. As we enforce a minimum ELW of 0.4 for the most likely ancestors in our core data, such mosaicism is interpreted as separate, individually likely ancestries, averaged across EPOCs within a pathway, rather than a general uncertainty of the constituent EPOCs. The global aELW values favoured an Asgard origin for these pathways, but there were also associations with Actinomycetota and Alphaproteobacteria, discussed further below. However, higher Asgard aELW was observed within the adjoining endoplasmic reticulum localized pathways for sphingolipid metabolism, a broad class of derived plasma membrane lipids in eukaryotes. Previous studies have highlighted possible convergent origins of this pathway in bacteria and eukaryotes[55,56], but here we detected a global association with Asgard archaea (Extended Data Fig. 4b). Of further note is the endoplasmic-reticulum-associated mevalonate pathway, converting Acetyl-CoA to mevalonate and further to Farnesyl and Geranyl diphosphate, which we also found to be associated with Asgard (Extended Data Fig. 4b). In eukaryotes, mevalonate is required for isoprenoid synthesis, in turn forming the precursor units for both sterols, carotenoids and terpenoids, synthesized in the endoplasmic reticulum lumen by the mevalonate pathway. However, in archaea, isoprenoids are the precursors for the ether-linked membrane lipids and therefore are closely associated with membrane biogenesis[57]. In conclusion, we detected Asgard associations across a wide range of cellular functions and metabolic pathways and noted a relationship between archaeal lipid synthesis and eukaryotic steroid precursor metabolism. A distinctly weaker Asgard signal was detected for pathways involved in bacterial lipid biosynthesis, suggesting a complex evolutionary history.

## Specific alphaproteobacterial traces

In line with the central role of mitochondria in eukaryotic energy metabolism and, as recently reported for central carbon metabolism[58], we observe primarily clear associations between Alphaproteobacteria and mitochondrially localized metabolic pathways. Apart from the components of the mitochondrial translation system, the most prominent alphaproteobacterial associations were evident within complexes involved in oxidative phosphorylation and the associated ubiquinone synthesis (Extended Data Fig. 4c). Outside these central energy-transforming functions, we detected only sparse contributions from core alphaproteobacterial genes. One such prominent association was the pathway for iron-sulfur cluster (ISC) biogenesis. As previously reported[59], the ISC assembly machinery is of alphaproteobacterial origin and, accordingly, we found the 4Fe-4S ISCA platforms, as well as IBA57 and Fe–S cluster binding ferredoxin-1 and 2, to be associated with Alphaproteobacteria (Extended Data Figs. 4c and 6). However, the 2Fe-2S precursor scaffold ISCU showed a mosaic association with diverse prokaryotes (including Alphaproteobacteria) and a weaker but clearly detectable Asgard association. Furthermore, the cysteine desulfurase NFS1 and the upstream pathways for the biosynthesis of sulfur-containing amino acids—cysteine and methionine—were Asgard-associated. The alphaproteobacterial associations with Fe–S synthesis were therefore restricted primarily to the synthesis and integration of 4Fe-4S into the oxidative phosphorylation complexes. Outside the mitochondria, alphaproteobacterial associations with Fe–S metabolism cease. Here, ISC insertion depends on the cytosolic targeting complex CIA, which consists of CIAO1, CIAO2B and MMS19 (ref. 60). For the CIA components CIAO1 and CIAO2B, we observed a clear association with Asgard, whereas MMS19 is not detected in our data. Taken together, these observations indicate that the contributions of Alphaproteobacteria to eukaryogenesis are centred mostly around the mitochondrion.

## Sparse contributions from other bacteria

Although our analysis greatly expands the Asgard contributions to eukaryogenesis, and reveals limited but consistent alphaproteobacterial associations, contributions from other bacteria were detected as well. For some biological functions, this 'diverse' bacterial component accounted for most of the aELW, and roughly one-third of the KOGs (600 of 2,100) and EPOCs (1,580 of 4,290) analysed had their strongest association with neither Asgard nor Alphaproteobacteria. However, in a sharp contrast to the limited but functionally related associations between alphaproteobacterial and oxidative phosphorylation and Fe–S metabolism, EPOCs associated with diverse bacteria showed few functional trends, instead demonstrating low-level association across most metabolic functions. Although Cyanobacteria and Actinomycetota exhibited stronger global aELW compared with Alphaproteobacteria, this association stemmed from several small contributions across disjoint pathways rather than specific functionally related contributions (Extended Data Fig. 7). To emphasize this trend, across the diverse set of bacterial taxa, only Alphaproteobacteria and Myxococcota were associated with a KEGG pathway containing more than 20 EPOCs and with a greater aELW than Asgard (Fig. 2 and see Extended Data Fig. 9a for alternate cutoffs). Specifically, oxidative phosphorylation shows a strong signal for Alphaproteobacteria while Myxococcota (formerly Deltaproteobacteria) showed a consistent association with metabolic pathways involved in nicotinate, nucleotide and nucleoside-sugar modification and metabolism. However, these nucleotide-related associations appear to be limited to phosphatases and phosphoribosyltransferases acting on nucleoside-sugars, specifically 5′ and 3′ nucleotidases (Extended Data Fig. 8b), and are much more limited in functional scope than pathways of alphaproteobacterial associations. Previous studies highlighted a link between Myxococcota and steroid

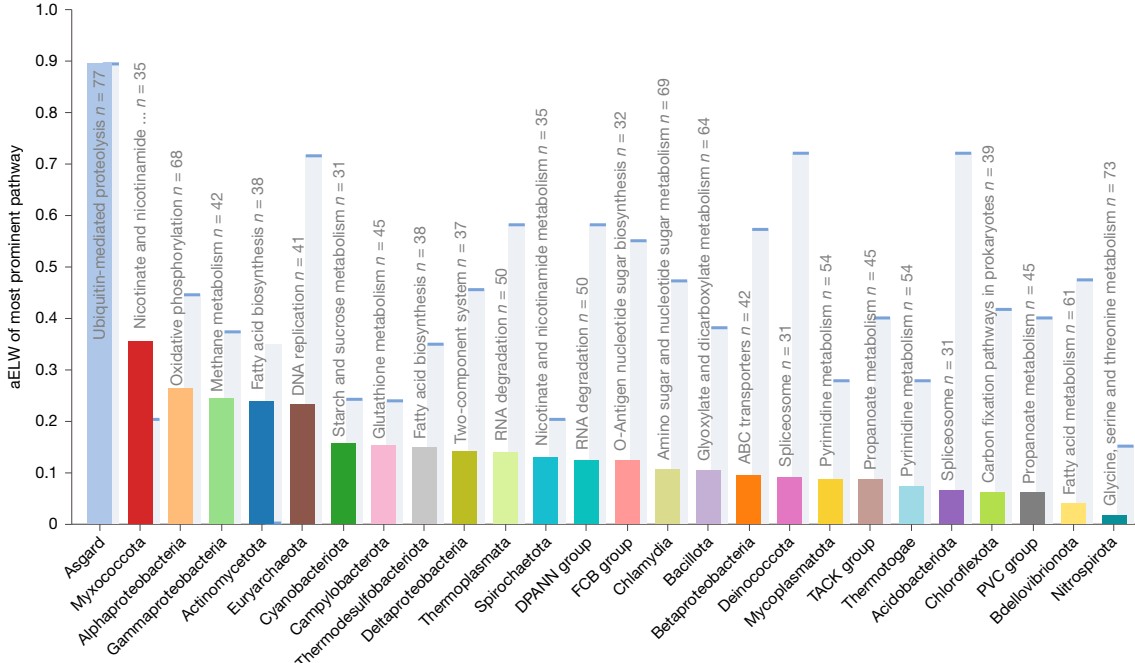

**Fig. 2 | Strongest bacterial contributions to eukaryotic pathways compared with the contributions of Asgard archaea.** Each prokaryotic taxon is represented by the highest aELW of any KEGG BRITE pathway with at least 20 EPOCs, relative to that of Asgard for the same pathway (background, pale blue). The strongest

prokaryotic association relative to Asgard was that of Myxococcota with nicotinate metabolism followed by Alphaproteobacteria with oxidative phosphorylation.

biosynthesis in Eukarya[61]. Here we observed a clear, if narrow, association with Myxococcota for diterpenoid biosynthesis, downstream of the mevalonate pathway (see above) and upstream of steroid synthesis, although our criteria on taxonomic scope limited the representation of downstream steroid pathways in the core data (Extended Data Fig. 8a).

Although few specific associations were observed with diverse bacteria on the level of pathways, the possibility remained that individual proteins with key functions could originate from such taxa. For this analysis, we considered a stricter subset of the core EPOCs, requiring a eukaryotic outgroup containing at least 15 taxonomic clades, to ensure functional centrality, and prokaryotic sister taxa with at least 20 sequences and an ELW > 0.7 to ensure strong association. Of the 130 unique EPOCs meeting these criteria, 33 were found to be associated with diverse bacterial lineages, mostly Actinomycetota and FCB group bacteria (Extended Data Fig. 9b). The remaining 97 EPOCs were found to be Asgard-derived. The 33 EPOCs of prominent bacterial origin covered a wide range of cellular functions from main facilitator superfamily transporters to lipases, to components of core sugar metabolism and cardiolipin synthesis. Once again, we observed no clear functional trend for these bacterial contributions. Thus, although single protein associations were individually significant, no pathways appeared to have been derived from any single bacterial taxon. Instead, we interpret these diffuse associations with diverse bacteria as indications of several bacterial contributions, each with a limited functional scope, probably originating from isolated HGT events rather than additional endosymbioses.

## Ancestral stem lengths

We attempted to use tree stem length analysis to infer the timing of acquisition of core eukaryotic genes, as originally implemented by Pittis and Gabaldon[33], with the mitochondrion as a test case. As the time elapsed post LECA is the same for all genes, it has been proposed that the relative stem length in gene trees is proportional to the time elapsed since the acquisition of the gene from its prokaryotic donor to the LECA. Using this approach, Pittis and Gabaldon found that proteins

of alphaproteobacterial descent had significantly shorter stems than proteins of archaeal descent, consistent with a mitochondria-late scenario for eukaryogenesis[33–35]. The raw stem length is defined as the distance from the First Eukaryotic Common Ancestor (FECA), represented by the Last Common Ancestor (LCA) of Eukarya and the most likely prokaryotic sister phylum[50,62], to the LECA. To account for differences in evolutionary rates, this raw stem length is normalized through dividing by the median eukaryotic branch length, measured from the LECA (Fig. 3a).

Across the entire set of eukaryotic stem lengths for our dataset (5,850 stems), we observed a wide distribution with a sharp maximum close to 0.05, highly reminiscent of previous findings[33] (Fig. 3b,c). In our analysis, alphaproteobacterial stems shorter than 0.3 (70% of all stems) were significantly longer than those of Asgard origin, but alphaproteobacterial stems longer than 0.35 were instead shorter than Asgard on average, showing a non-obvious yet reproducible (Extended Data Fig. 10b) and statistically significant (Fig. 3b) relationship. Stems of cyanobacterial origin followed a similar but weaker trend. Thus, although the observation of globally shorter stem lengths for genes of alphaproteobacterial descent applied only to stems longer than 0.35, we effectively recapitulate the findings of Pittis and Gabaldon, who analysed primarily stems within that range. When we limited our analysis to ribosomal proteins and components of oxidative phosphorylation complexes, that is, genes of well-known and easily identifiable archaeal and alphaproteobacterial ancestries, respectively, the above trends were further accentuated (Fig. 3c). Contrary to the notion of stem lengths as a measure of acquisition timing, we observed variances of more than two orders of magnitude in normalized stem lengths for the oxidative phosphorylation complexes that are assumed to have been acquired simultaneously during mitochondrial symbiogenesis (Fig. 3b). Taken together, these findings suggest that even normalized stem lengths are determined primarily by factors other than the gene acquisition time, at least across the long evolutionary timescales analysed here.

Comparison of the stem length distributions across functional classes of genes (Fig. 3c) prompted a possible explanation for these

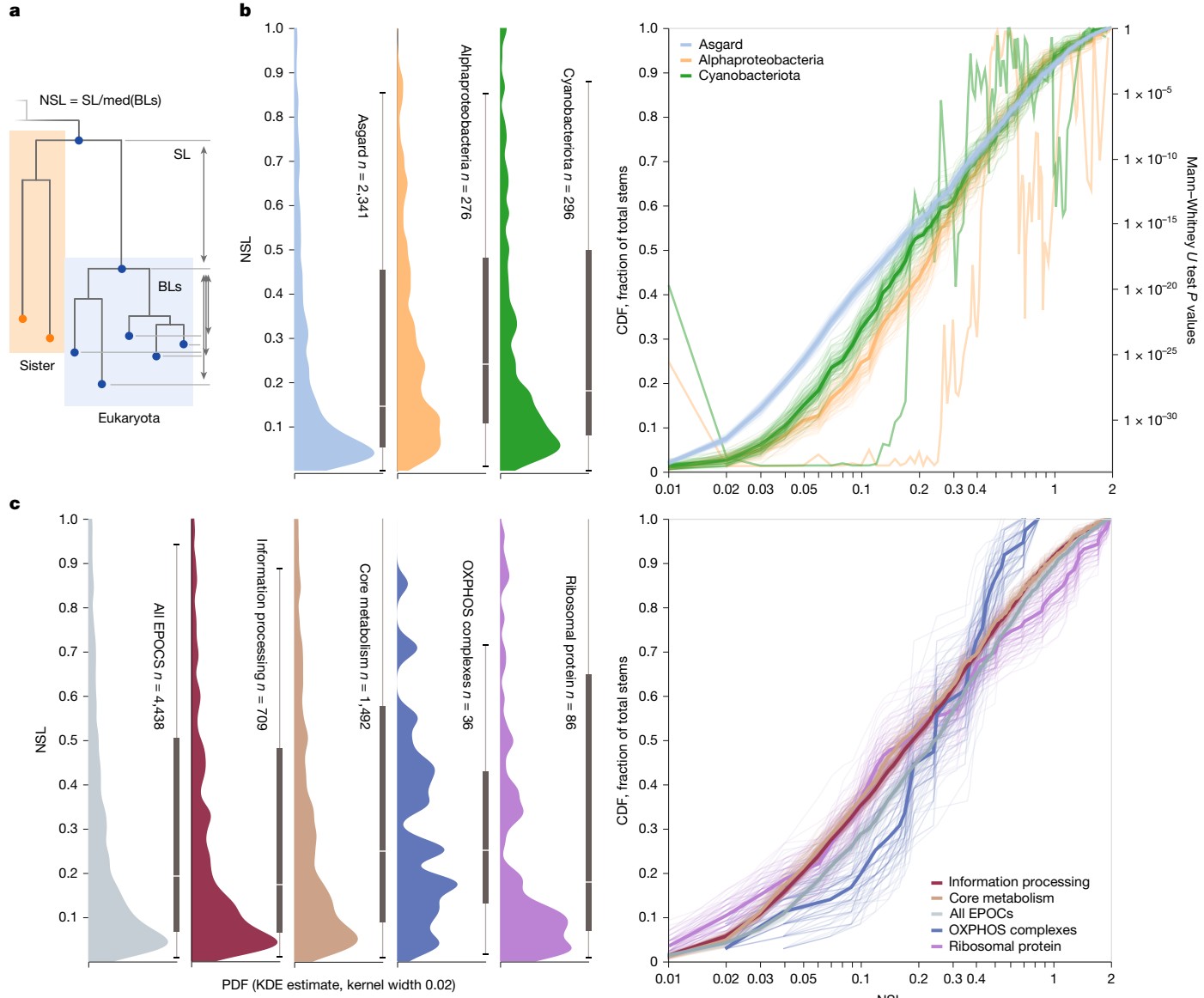

**Fig. 3 | Ancestral stem length distributions for different groups of prokaryotes contributing to eukaryogenesis and different functional classes of genes. a**, Schematic of stem length calculation. Stem lengths (SL) were calculated as the lengths of branches from the shared root of Eukarya and the inferred ancestral prokaryotic sister phylum to the root of the Eukarya clade. Normalized stem lengths (NSL) shown in **a** and **b** were calculated by dividing SL by median branch lengths from the root of Eukarya to individual Eukarya leaves (med(BLs)). **b**, Left, probability density function (PDF) of normalized stem lengths taken from samples of individual core eukaryotic genes associated with prokaryotic taxa, obtained through kernel density estimation. Thin bar, 95th interpercentile range; thick bar, 50th interpercentile range with median indicated. Right, fractional cumulative distribution functions (CDF) of normalized stem lengths together with 20 of 200 bootstrap replicates from EPOCs within each category. Second Y loglinear axis indicates the value of the two-sided Mann–Whitney $U$ test of bootstrap CDF distributions per stem length between Asgard and either Cyano- or Alphaproteobacteria. **c**, As in **b** with genes grouped by BRITE A Metabolism (KO:09100), Genetic Information Processing (KO:09120), Oxidative phosphorylation (map00190 excluding KOGs for V-type ATPases).

observations. Shorter than average stems were observed for genetic information processing, belying the expectation of these genes being the longest-residing genes in the nascent eukaryotic lineage. We suggest that another principal factor determining the relative stem length of a gene might be the amount of evolution post-acquisition, necessary to adjust an acquired gene to a new biological environment. The genes inherited from the Asgard ancestor would have been pre-adapted to the cellular environment of the evolving protoeukaryotes, whereas genes acquired from radically different bacterial sources had to adapt substantially post-acquisition, increasing the apparent lengths of their pre-LECA stems, which is at least partially supported by our general observation of long stems from diverse bacteria (Extended Data Fig. 10c).

## Discussion

Reconstruction of eukaryogenesis is a moving target, as strikingly demonstrated by the discovery and expanding exploration of Asgard archaea, which revolutionized the field[2–7]. Nevertheless, recent progress in prokaryotic and eukaryotic genome sequencing and advances in metagenomics have created unprecedented opportunities for understanding the phylogenomic origins of eukaryotes. In this work we took advantage of this expanded genome collection, coupling it with maximally sensitive HMM profile–profile searches. We then performed a comprehensive phylogenetic analysis of core eukaryotic genes tentatively mapped to the LECA, within a statistical

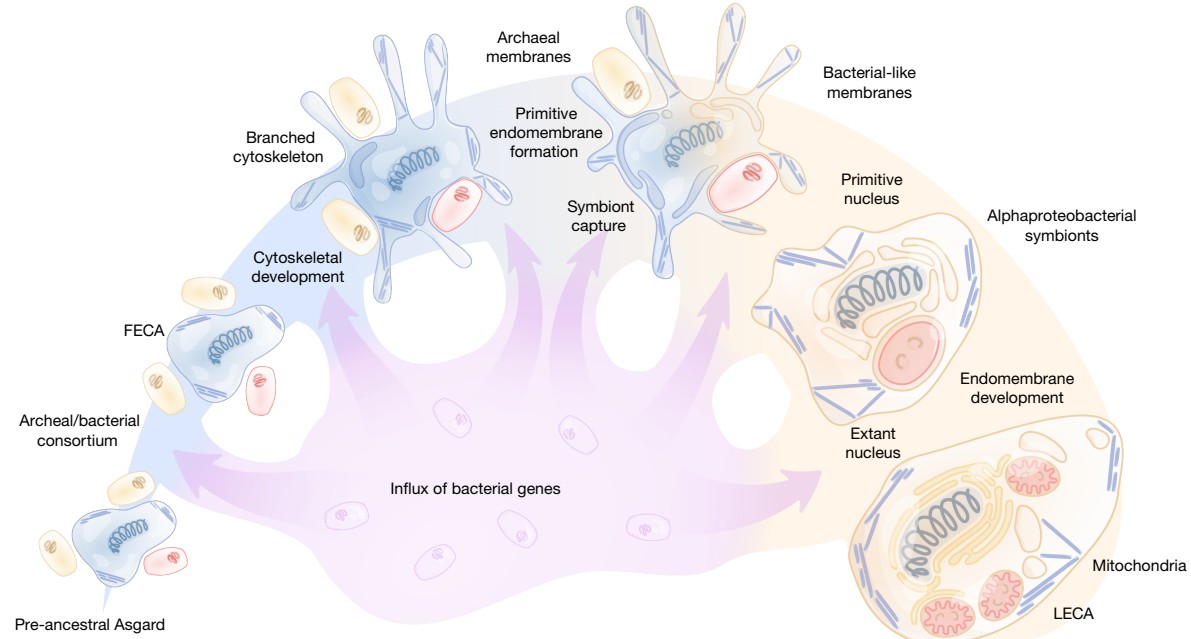

**Fig. 4 | Hypothetical scenario of eukaryogenesis.** Illustration of gene flow along the succession of events leading from a simple archaeal ancestor through the more complex Asgard intermediate capturing the proto-mitochondrial alphaproteobacterial endosymbiont to the LECA. The wide coloured shape denotes the dominant trend of the origin of eukaryotic genes from Asgard ancestors, with light blue portions corresponding to archaeal evolution and the light brown portion to the evolution of protoeukaryotes starting with the alphaproteobacterial endosymbiosis. The purple arrows denote the smaller, piecemeal contributions of gene influx from various bacteria that are posited to have been occurring at all stages of the depicted evolutionary scenario.

framework focused on testing evolutionary hypotheses using constraint trees.

Many previous studies, especially early ones, demonstrated the apparent chimeric origins of the core eukaryotic gene set, with a bacterial contribution quantitatively exceeding the archaeal one, the latter being largely limited to information processing[17–19,63]. The discovery and exploration of Asgard archaea partially changed this notion by demonstrating that many genes involved in various cellular processes and systems, such as membrane remodelling and cytoskeleton, were of apparent Asgard origin[5,6,8]. Indeed, the presence of an elaborate cytoskeleton in Asgard archaea has been demonstrated subsequently experimentally[24]. The present work substantially extends the dominance of Asgard in the ancestry of the eukaryotes by demonstrating a likely Asgard origin for most genes traced to the LECA, and a principal Asgard association for nearly all functional systems and pathways of the eukaryotic cell. In only a handful of metabolic pathways did the detected bacterial contribution exceed that of Asgard archaea. The most conspicuous of the bacterial contributions comes, not unexpectedly, from Alphaproteobacteria and accounts for the core components of the electron transport chain, ISC metabolism and the mitochondrial translation system. However, although prominent and functionally coherent, the alphaproteobacterial contribution is relatively small compared with that of Asgard and is notably almost entirely restricted to mitochondrial functions. The contributions of other bacterial phyla, although cumulatively substantial and, in select cases, strongly supported for individual EPOCs, failed to show any consistent functional trends, with the exception of notable but limited associations between Myxococcota and the synthesis of nucleotides, nucleoside-sugars and steroid precursors.

These findings are most compatible with a scenario of eukaryogenesis in which the Asgard FECA already possessed many signature features of eukaryotic cells, including the cytoskeleton and the endomembrane system as posited in some previous models[26,64]. Under this scenario, mitochondrial acquisition was a relatively late event that made a limited, but functionally crucial, contribution to the gene composition of the emerging eukaryote. Other bacterial contributions appear to be piecemeal, possibly stemming from a continuous but non-specific capture of genes from diverse bacteria during the evolutionary path from the ancestral Asgard archaeon to the LECA (Fig. 4). The syntrophic lifestyle of many Asgard archaea[32] is conducive to extensive gene influx from bacteria, enhancing the general trend of mosaic gene composition of prokaryotic genomes[65,66]. Much of the bacterial gene acquisition likely preceded the emergence of the proto-eukaryotic lineage and more occurred along the path from the FECA to the LECA (Fig. 4). Thus, some of the core eukaryotic genes, although inherited proximally from the Asgard ancestor, might ultimately be of bacterial origin, having been carried by Asgard into the FECA. As such, we obtained no indication of another symbiotic event contributing to eukaryogenesis.

How eukaryotes acquired bacterial-type membranes remains an enigma. Whereas eukaryotic isoprenoid synthesis seems to have been derived from Asgard lipid synthesis, the ancestry of the enzymes involved in membrane lipid biosynthesis appears complex and possibly mosaic, remaining unresolved in the present analysis. Nevertheless, the Asgard affinity of adjoining systems and pathways, such as the synthesis of GPI-anchors, sphingolipids and isoprenoids, indicates that a substantial component of the lipid biosynthesis and modification capacity of eukaryotes persisted from the Asgard ancestor. If an Asgard archaeal cell was the principal scaffold on which the eukaryotic cell evolved, replacement of the archaeal membrane with a bacterial one must have occurred at an early stage of eukaryogenesis, progressing through a mixed membrane stage. Viable bacteria with a mixed bacterial–archaeal membrane have been engineered experimentally, albeit typically with a substantial fitness decrease[66–68], demonstrating the possibility of such a scenario. Membrane replacement might have even predated mitochondrial endosymbiosis, following early capture of the bacterial enzymes involved in membrane biogenesis by the Asgard archaeon (Fig. 4).

A key issue for understanding eukaryogenesis is the origin of the nucleus—the ubiquitous, eponymous organelle of eukaryotic cells[69,70]. Our analysis supports the near-complete origin of

information-processing systems located in the nucleus from Asgard ancestors. However, given that the nuclear envelope membrane is an extension of the endoplasmic reticulum and, as such, is of the bacterial type[71], whereas the nuclear pore consists largely of repetitive proteins of uncertain provenance[72], the origin of the nucleus as a membrane-bounded compartment remains an unresolved question that has to be addressed in conjunction with the membrane replacement problem.

The conclusions of this work are subject to caveats stemming, above all, from the biased and still limited sampling of sequenced archaeal and bacterial genomes. In particular, almost all currently available genomes of Asgard archaea are incomplete metagenomic assemblies, with the exception of only a few closed circular genomes[22–24,31,73], which inevitably leads to uncertainty in the estimation of the Asgard contribution to eukaryogenesis. The same pertains to Alphaproteobacteria, both because of the sequencing bias whereby many of the sequenced alphaproteobacterial genomes come from symbionts and parasites with reduced gene complements, and because the ancestor of the mitochondria apparently belongs outside of the currently known diversity of Alphaproteobacteria[74]. Other bacterial phyla are even more severely under-sampled, potentially leading to underestimates of their contributions to the evolution of eukaryotes. For example, given the limited genomic information, it is difficult to rule out that we capture only part of a principal signal from Myxococcota, proposed as one of the partners in the syntrophic scenario of eukaryogenesis[15,22]. Furthermore, although the dominant core Asgard contribution was fully robust under the two clustering approaches used in this work, the correlations for other taxa were lower, suggesting the possibility of changes once the genome collections for these taxa expand. Our pangenome-based approach to exclude late HGT from ancestral inference potentially could overlook contributions from narrow or severely under-sampled clades and might be insufficient to compensate for extensive and continuous HGT across diverse eukaryotic taxa. A recent preliminary report, using more liberal criteria for core pangenome construction, suggested a far greater bacterial contribution than observed here, potentially due to late HGT[65]. In the future, expanding sampling of prokaryotic genome diversity and further technical improvements in horizontal transfer detection might lead to a substantial modification of the model of eukaryogenesis. Nevertheless, we find it likely that the dominant contribution of Asgard archaea, the main finding of this work, is here to stay.

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

## Methods

### Database curation

For prokaryotes, 75 million sequences were curated from 47,545 fully sequenced prokaryotic genomes obtained from the NCBI GenBank (https://ftp.ncbi.nlm.nih.gov/genomes/) in November 2023 (prok2311) and supplemented with 441,150 proteins sequences from 63 Asgard metagenomes[4,5] (prok2311As), selected to represent all clades in these two publications where available. Protein sequences were either taken directly from GenBank annotations or produced by Prodigal v2.6.3 (ref. 75) trained on the set of 12 complete or chromosome-level assembled Asgard genomes. To include only those sequences that are present widely within prokaryotic families, and to minimize the possibility of post-LECA horizontal transfer from eukaryotes, soft-core pangenomes were constructed for each of our 26 curated taxonomic groups, conforming mostly to the NCBI taxonomy rank 'class' (see below and Extended Data Fig. 1). To construct the soft-core pangenomes, all sequences from each taxonomic group were clustered individually using mmseqs2 (ref. 37) 'mmseqs cluster -s 7 -c 0.9 --cov-mode 0'[37]. All clusters that did not contain sequences from at least 50% of all bacterial species (here defined by lowest rank taxonomic ID within the NCBI taxonomy), or 20% of eukaryotic sequences (see below), were rejected from the database. The effects of alternative pangenome definitions, such as 10%, 25%, 50% and 67% proteins per class, are shown in Extended Data Fig. 1d.

The eukaryotic database was constructed starting from a curated set of 72 genomes available from the NCBI GenBank. To provide a more accurate representation of eukaryotic diversity, this dataset was supplemented with EukProt v.3 (ref. 36)—a curated database aiming to provide a sparse representation of eukaryotic biology, to create Euk72Ep containing 30.3 million sequences. As sequences included in EukProt v.3 come from a diverse range of sources, some known to contain contaminant prokaryotic sequences, the database was screened for prokaryotic contamination. This screen was carried out by first constructing candidate prokaryotic and eukaryotic HMM databases as described below and querying both databases against the eukaryotic sequence database using hhblits[43]. All eukaryotic sequences with a top hit alignment score from a prokaryotic HMM were removed from the database before initial clustering.

### Curation of taxonomic labels

To provide a compromise between taxonomic specificity and accuracy of clade assignment, we curated a set of taxonomic labels for both Euk72Ep and Prok2311As to act as representatives. To construct this set, we started by manually assigning all species in EukProt to their closest relatives in the NCBI taxonomy. Then, each species from Prok2311As and Euk72Ep was assigned a taxonomic label corresponding to their 'Class' rank (for example, Alphaproteobacteria, Thermococci, Mammalia) based on the NCBI Taxonomy of November 2023 using tools from the ete3 v.3.1.3. toolkit[76] as well as custom scripts. For species without a corresponding 'Class' rank, the closest relevant rank was assigned manually. This procedure yielded an initial list of 317 unique taxonomic class identifiers. As the rank 'Class' does not partition biological diversity evenly, identifiers were further mapped manually to a curated set of 45 eukaryotic and 26 prokaryotic taxonomic superclasses, each present in the NCBI taxonomy (for example, Metazoa, Streptophyta, TACK archaea; Extended Data Fig. 1. Due to the partially unresolved taxonomic relationship within Asgard archaea, all candidate Asgard sequences from Prok2311 or from refs. 6,7 were assigned to the class label 'Asgard'. Following the recent reclassification of Deltaproteobacteria into Myxococcota, Desulfobacteriota, Bdellovibrionota and SAR324 species that could not be mapped into either of these clades using existing taxonomic annotation were classified as Deltaproteobacteria (0.025% of total data).

### Sequence clustering and profile database generation

To transform Euk72Ep and Prok2311As into profile databases suitable for sensitive HMM–HMM searches, we implemented an unsupervised, cascaded sequence-profile clustering pipeline using the tools available in the mmseqs2 software suite[37]. In brief, each sequence database was initially clustered at 90% sequence identity and 80% pairwise coverage using 'mmseqs linclust' and cluster representatives were chosen using 'mmseqs result2repseq'. This procedure provides a non-redundant set of 6.3 million prokaryotic and 25.1 million eukaryotic sequences for further analysis. The non-redundant sequences are further collapsed into a set of initial profiles and consensus sequence pairs using 'mmseqs cluster', 'mmseqs result2profile' and 'mmseqs profile2consensus'. All consensus sequences are queried against their profiles using 'mmseqs search' and results clustered using 'mmseqs clust'. Clusters are mapped to their original non-redundant sequences using 'mmseqs mergeclusters' and profiles and consensus sequences are constructed once again. To grow clusters based on their sequence-profile alignments, this procedure of 'mmseqs profile2consensus - mmseqs search - mmseqs clust - mmseqs result2profile' is iterated until cluster sizes converge. An 80% pairwise coverage is maintained throughout all steps of the cascaded clustering protocol to avoid homologous overextension. The final clustered databases contain 14.1 million and 91,000 clusters for Euk72Ep and Prok2311As, respectively.

To transform the unsupervised clusters into a set of HMM profiles, we used a mixed MSA construction strategy coupled with automatic data reduction to create accurate alignments for a wide diversity of cluster sizes and sequence diversities. To construct HMMs, all non-singleton sequence clusters are first aligned using FAMSA v.2.0.1 (ref. 77) to produce an initial alignment. Using the initial alignment, we then reduce the sequence space by calculating an approximate tree using Fast-Tree v.2.1.1. using 'FastTree -gamma'[78], remove long branch outliers as described in 'EPOC tree construction and processing' and iteratively removing the closest pairwise leaves using ete3 (ref. 76), keeping the leaf closest to the root, until the desired sequence set size is reached. For profile generation, we retain no more than 300 sequences for both Prok2311As and Euk72Ep. This reduced sequence set is then realigned with muscle5 v.5.1.linux64 (ref. 42) using 'muscle -diversified' with five replicates, and the maximum column confidence alignment is extracted using 'muscle -maxcc'. This alignment is then trimmed to keep columns with more than 0.2 bits of Shannon information defined as the difference of $\log_2(20)$ and the Shannon entropy of the column amino acid distribution. This process of data reduction/alignment/trimming is referred to as 'prune-and-align' and used later for downstream analysis. Trimmed alignments are turned into an HMM database using HH-suite with 'hhconsensus -M 50', hhmake and 'cstranslate -f -x 0.3 -c 4 -I a3m'. The final HH-suite databases contained 480,000 and 1.6 million profiles for prokaryotes and eukaryotes, respectively.

### Cluster annealing

As the initial mmseqs2 based clustering was carried out in an unsupervised manner, we considered it necessary to ensure that the results were not contingent on our unsupervised partitioning of the data. To this end, a further cluster annealing step was developed which aims to redefine clusters based on the notion of clades as well separated monophyletic groups on phylogenetic trees, rather directly by sequence similarity and coverage. Starting with the existing clustered databases Euk72Ep and Prok2311As, HMM profiles were calculated for each non-singleton cluster and all versus all HMM–HMM searches were performed using HHblits v.3.3.0 (HHblits -n 1 -p 80), retaining hits with at least 80% HHblits probability and 50% pairwise profile coverage. The resulting hits were then clustered using greedy set clustering, as described in mmseqs2, implemented in Python. These superclusters represent our limit of detection for sequence-based methods, but might involve overclustering, possibly merging several similar gene families into one set. To mitigate the potential effect of such overclustering, cluster partitioning was performed based on phylogenetic tree analysis.

To this end, the supercluster sizes were first reduced using the 'prune-and-align' approach described above, after which a master tree was

constructed using FastTree -gamma. Given the resulting set of trees, we sought to identify well separated monophyletic sets of leaves, the presence of which would warrant partitioning of the respective superclusters. To perform this tree partitioning in an unsupervised manner, the all-versus-all pairwise leaf distances were calculated as a square distance matrix and embedded into two dimensions using Uniform Manifold Approximation and Projection v.0.5.3 (n_neighbors=50, min_dist=0.3, distance='euclidean')[79]. This embedding was clustered using HDBSCAN as implemented in scikit-learn v.1.3.0 with default parameters[80]. We explicitly allow single cluster partitions to ensure that trees with uniform distributions of pairwise branch lengths remain as single clusters. Mapping clusters back to the phylogenetic trees showed that uniform manifold approximation and projection embedded HDBSCAN clusters were well separated in the original trees (Extended Data Fig. 3a). These annealed clades were used in downstream processing, representing a stricter definition of EPOCS than the initial cascaded mmseqs2 clusters. The annealing process redistributed sequences from the 1.6 million eukaryotic clusters into 1.5 million annealed clusters and from 480,000 prokaryotic clusters into 280,000 annealed clusters, highlighting the already dense representation of initial mmseqs2 clusters. Because the embeddings were done per tree, the partitioning into subsets is independent of branch lengths and relies merely on the internal distribution of branch lengths and the sequence diversity within each tree, rather than pre-defined values of tree distances. Thus, well-formed subclades could be identified in sequence sets independent of internal diversity, an essential feature when applying this procedure across our global scope.

### Search for prokaryotic homologues of eukaryotic proteins

Given that clade-specific protein families were not of direct relevance to this study, we reduced the search space by excluding HMM profiles from eukaryotic clusters with fewer than ten sequences and a lowest common ancestor with a taxonomic rank below 'Superkingdom' as calculated by 'mmseqs lca'. This requires a sequence cluster to contain at least two sequences from different kingdoms, as defined by the NCBI taxonomy (for example, Amoebozoa and Haptista, or Viridiplantae and Opisthokonta), retaining 142,000 profiles of conserved eukaryotic proteins. To associate eukaryotic sequence clusters with homologous prokaryotic clusters, we queried the Euk72Ep HMM database against all Prok2311As HMM profiles using a single iteration of HHblits v.3.3.0 as 'HHblits -n 1 -p 80' and retaining the hits with at least 80% HHblits probability and 80% pairwise profile length. These filters resulted in a total of 20,700 accepted eukaryotic query profiles targeting a total of 8,300 unique prokaryotic profiles. These clusters contained a total of 5.7 million and 1.9 million sequences.

### EPOC MSA construction

Unless otherwise stated, all subsequent steps were carried out using custom Python v.3.9 scripts, with taxonomy and tree parsing carried out using ete3 (ref. 76). All sequences from a eukaryotic cluster with at least ten sequences and a lowest common ancestor at the rank of 'Superkingdom' (NCBI taxonomy), together with all members of the homologous prokaryotic clusters identified in the HMM–HMM search with at least 80% probability and pairwise sequence coverage, forms an EPOC. As EPOCs varied in size by more than five orders of magnitude (10 to 100,000 sequences), robust data subsampling is necessary for accurate downstream MSA generation and tree construction. We used a variant of the 'prune-and-align' strategy described above for profile generation taking into account the taxonomic distribution of the constituent sequences. Rather than iteratively pruning the closest leaves pairwise, we first label each leaf with its corresponding taxonomic label and only prune monophyletic groups, maintaining a count of all pruned leaves per clade. If the procedure fails to reach the target leaf number before converging, all leaves are ordered by the number of pruned relatives and leaves with the lowest number of relatives deleted (retaining

the largest monophyletic groups) until the target size is reached. The result approximates a maximum diversity representation, preferentially pruning isolated singleton clades. Using this modified protocol, eukaryotic and prokaryotic sequences are cropped to a maximum of 30 and 70 sequences, respectively. Each EPOC therefore consists of a maximum of 100 representative sequences. All EPOCs are aligned using 'muscle -diversified' with five replicates, and the maximum confidence alignment is extracted using the --maxcc argument. This alignment is then trimmed to keep columns with more than 0.15 bits of Shannon information content for tree reconstruction.

### EPOC tree construction and processing

From the pruned and trimmed alignment, we created a maximum likelihood tree using IQtree2 v.2.3.5. 'IQtree2 -B 1000 -bnni -m MFP -mset LG,Q.pfam --cmin 4 --cmax 12' with model parameters estimated by model finder plus[46]. Candidate models and rate category search ranges were selected based on a test set of 200 randomly chosen EPOCs from the filtered core set analysed with full model parameter evaluation by Model finder[46] and ten tree replicates. The LG and Q.pfam models and rate category ranges were chosen as they consistently produced the highest ranking log-likelihoods for a random sample of 200 EPOCs run with full model estimation in IQtree.

As the master trees are built from sequences obtained through sensitive cascaded sequence-profile clustering, we would in rare cases observe long-stemmed clade outliers in addition to more common long stem leaf outliers. To assess and remove such erroneous data, we first estimated the log-normal distribution of all stem lengths using scipy. stats v.1.11.1 and excluded any stems outside the 0–99.5% probability point function interval. This simultaneously removes long terminal leaf branches as well as highly diverged clades. In rare cases where this pruning would remove either the entire eukaryotic clade or more than 30% of all leaves, the EPOC was discarded (387 EPOCs). The resulting trees were then re-rooted using a weighted midpoint approach so that the sum of all branch stem lengths on either side of the root was equal. For clarity, no trees in our current study have been 'phylogenetically rooted' in the biological sense. All references to the 'root' refer to the origin of the data structure, not the biological concept.

To assign taxonomic clades, all leaf nodes were assigned taxonomic labels from our curated set and monophyletic clades were collapsed, taking as the representative the lowest branching leaf. We found that assigning clades purely based on a monophyletic definition of taxonomic purity severely hampered the analysis as leaves can be taxonomically incongruous with their neighbours. This effect most likely stems from cases of locally erroneous tree topology or local HGT, but severely complicates the downstream analysis. We therefore adopted the notion of a 'soft LCA' representing the root of a close-to-monophyletic clade. To greedily rank all the best 'soft LCAs', for each taxonomic label, the tree was traversed from each monophyletic clade root to the tree root, calculating a score for each internal node:

(Number of leaves with label X in clade/clade size)
 × (Number of leaves with label X in clade/total number of label X)

This metric balances taxonomic 'purity' and 'scope' for each possible clade of label X. All such nodes are then ordered by this score. Because this score is sensitive to the total number of taxonomic labels in an EPOC, it was not used as a filtering criterion at any point in the analysis, yet is present in the metadata, mentioned here for completeness. To avoid overinterpreting small clades, clades were considered valid if they represented at least three sequences with clade purity more than 0.8 for prokaryotes, and at least five sequences with purity more than 0.8 for eukaryotes. Trees that fail to identify either any eukaryotic or any prokaryotic clades under these constraints, derived primarily from small EPOCs, are discarded (411 EPOCs). Trees with more than three valid eukaryotic clades were considered to show high paraphyly and likewise discarded (781 EPOCs).

## Evolutionary hypothesis testing using constraint trees

To assess the relative probabilities of each sampled prokaryotic clades to represent the eukaryotic sister clade, we used evolutionary hypothesis testing using constraint trees as implemented in IQtree2 (ref. 46). For each EPOC and for each detected eukaryotic clade, we generated a set of constraint trees, one for each possible prokaryotic sister. Each constraint tree had three defined clades enforced throughout tree calculations: (1) the eukaryotic group as defined by all sequences below the eukaryotic LCA, (2) one of the prokaryotic sister groups and (3) all other prokaryotic leaves. We then construct a set of local trees from slices of the original MSA using IQtree2, forcing the constraint tree topology using 'iqtree2 -g', using the same evolutionary model as the original, unconstrained 'master tree'. The set of constrained trees is then ranked using 'iqtree2 -z' calculating the relative model assignment confidence for each of the constrained topologies and resulting test metrics are saved for downstream analysis. As the number of trees to evaluate scales by the number of eukaryotic clades multiplied by the number of prokaryotic clades, we limit the sampling to a maximum of three clades per tree for eukaryotes, discarding rare cases of high eukaryotic paraphyly, and only consider the 12 closest prokaryotic clades to each individual eukaryotic clade, based on their topological distance calculated as the number of non-root tree bifurcations from one node to another. The procedure results in an ELW[44] for each of the evaluated model trees, here taken as being analogous to model selection confidence that an evaluated prokaryotic clade is the most significant sister to a eukaryotic clade.

## EPOC annotation

To functionally annotate protein families within EPOCs, we generated profiles based on protein sequences in KEGG release v.110 (ref. 45). For each KOG, we extracted KEGG metadata including KEGG pathways and BRITE labels through KEGG's API. At the time of parsing (January 2024), KEGG contained 26,695 KOGs. Due to KEGG API constraints, protein sequences in KEGG were extracted from Uniprot, first by mapping KEGG proteins to Uniprot IDs using Uniprot's ID mapping service (https://www.uniprot.org/id-mapping), then by extracting the sequences for these Uniprot IDs corresponding to each KOG[81].

For each KOG, we generated HMM profiles using the prune-and-align pipeline. To increase specificity of labelling, we partitioned those KOGs, which encompass both prokaryotic and eukaryotic sequences, into separate taxonomic groups and sequence alignments were generated separately for prokaryotes and eukaryotes. Each eukaryotic HMM profile forming the basis for a EPOC was queried against our KEGG profile database using 'HHblits -n 3 -p 80'. Results were subsequently filtered to a pairwise profile coverage of 0.7 using custom scripts. As a result, 12,600 of the 13,500 EPOCs could be annotated by at least one KOG profile at 80% probability and 70% pairwise coverage. The highest probability hits were used for annotation.

## Data filtering and removal of cases of possible late HGT

Unless otherwise stated, all data are interpreted from the core set of EPOCs meeting the following criteria: (1) the eukaryotic constituent cluster profile identifies at least one target profile in our KEGG profile database at a probability of 80% and 70% coverage; (2) the eukaryotic clades include more than five distinct taxonomic labels, at least one from Amorphea and one from Diaphoretickes, encompassing the LECA and (3) only prokaryotic sister clades with 0.4 < ELW < 0.99 were considered reliable and were included in the analysis. We note the independence on aELW values on the number of eukaryotic clades chosen as a cutoff as shown in Extended Data Fig. 10a. Cases of ELW ≥ 0.99 were not considered as these are found to be disproportionately eukaryotic-like, often branching from within a broader eukaryotic clade, and thus likely derived from late horizontal transfer post LECA. Due to the presence of such high ELW values within alphaproteobacterial associations with Oxidative phosphorylation, in this case, as an exception, ELW values of 1 were included.

## Taxonomy remapping and EPOC construction under GTDB taxonomy

To ensure that the results were robust to taxonomy, the prok2311 database was reformatted under the GTDB taxonomy (v.220) at the level of 'phyla'[49]. For taxonomic mapping, first, genomes present in both prok2311 and GTDB were annotated directly with the GTDB taxonomy. For the remaining genomes, taxonomy was assigned through consensus voting by using the set of GTDB marker genes, searching against our data using 'mmseqs search -s 7 -c 0.5 -e 1e-10'. A phyla level taxonomic label was assigned to each genome based on the largest number of total hits. Genomes without significant hits to GTDB marker genes were discarded for the purpose of this analysis (0.92% of all data). This procedure mapped the prok2311 data to 96 GTDB phyla that were used as taxonomic assignment for EPOC calculations as described above. All additional Asgard data taken from ref. 8 were directly assigned as Asgardarchaeota.

## Data visualization and plotting

All plots were prepared using Python v3.9 and pandas v.2.0.3 with Altair v.4.2.2. Layout, annotation and vector editing was done using Inkscape v.1.1.1. All statistical tests were carried out using scipy.stats v.1.11.1.

## Reporting summary

Further information on research design is available in the Nature Portfolio Reporting Summary linked to this article.

## Data availability

All final non-redundant databases with taxonomy annotation, clustering data, HMM profiles, EPOCs data and final parsed and tabulated results and metadata are available at Zenodo (https://doi.org/10.5281/zenodo.15048010)[82].

## Code availability

All code and scripts used to generate the final parsed data are available on Github at https://github.com/VictorTobiasson/eukgen.

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

**Acknowledgements** We thank T. Gabaldon and the Koonin Group members for helpful discussions. The authors' research is supported by the Intramural Research Program of the National Institutes of Health of the USA (National Library of Medicine).

**Author contributions** E.V.K. initiated the study. V.T. and Y.I.W. developed the pipeline for data analysis. V.T. and J.L. collected the data. V.T., J.L., Y.I.W. and E.V.K. performed data analysis. V.T. and E.V.K. wrote the manuscript, which was read, edited and approved by all authors.

**Competing interests** The authors declare no competing interests.

**Additional information**
**Correspondence and requests for materials** should be addressed to Victor Tobiasson or Eugene V. Koonin.

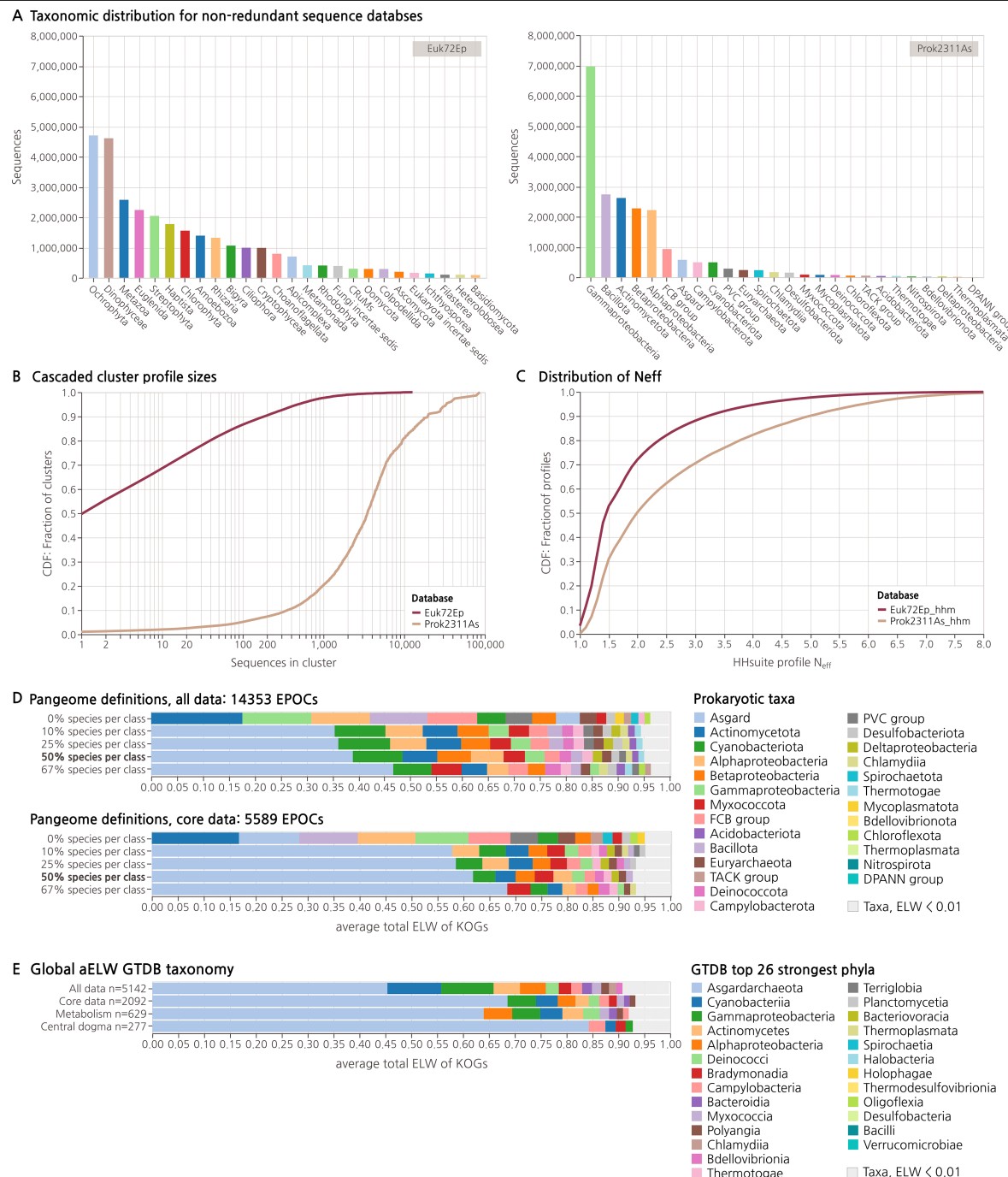

**Extended Data Fig. 1 | Global database statistics. A**) Distribution of taxonomic labels present within non-redundant Euk72Ep and Prok2311As (see Methods). Euk72Ep is limited to only display taxa with more than 100,000 sequences (99.69% of sequences). **B**) CDF fraction of cluster size distribution following cascaded mmseqs profile/sequence clustering. **C**) CDF fraction of $N_{eff}$ values as calculated by HH-suite for resulting profile databases. **D**) The effect of different pangenome regimes on final aELW values. Pangenomes are reconstructed enforcing a protein presence across 0, 25, 50, or 67% of species within clades and data shown for the full dataset of 13,500 EPOCs or core dataset, as described in methods. 50% criteria marked in bold as it is used as reference for the core data present throughout the paper. **E**) Global aELW for data remapped to the GTDB taxonomy at the level of Phylum using marker genes (see Methods). Displaying the top 26 strongest individually contributing phyla out of 92 accounting for 96.4% of all ELW.

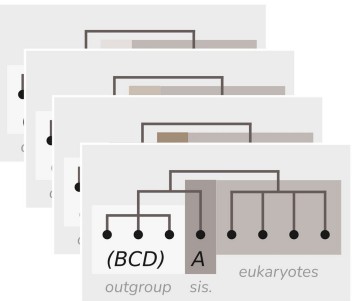

| construct the master tree and label clades | calculate all possible sister clade topologies | evaluate constrained tree with topologies | rank and calculate ELW for each model |

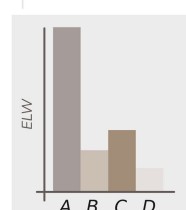

master tree set of contraint trees set of calculated subtrees relative scoring

**Extended Data Fig. 2 | Evolutionary hypothesis testing constraint trees and Expected Likelihood Weights.** Overview of ELW calculation procedure. One unrooted master tree is constructed from all sequences within an EPOC and the closest n sister clades to the eukaryotic outgroup identified. A set of unrooted constraint trees are generated, each enforcing three clades, one eukaryotic, one sample sister and one outgroup with remaining prokaryotic sequences, is present. The set of subtrees are then constructed using IQtree, each forced to conform to a single guide tree. Log likelihood values for resulting trees are compared and evaluated using IQtree -z to produce confidence sets and Expected Likelihood Weight estimations.

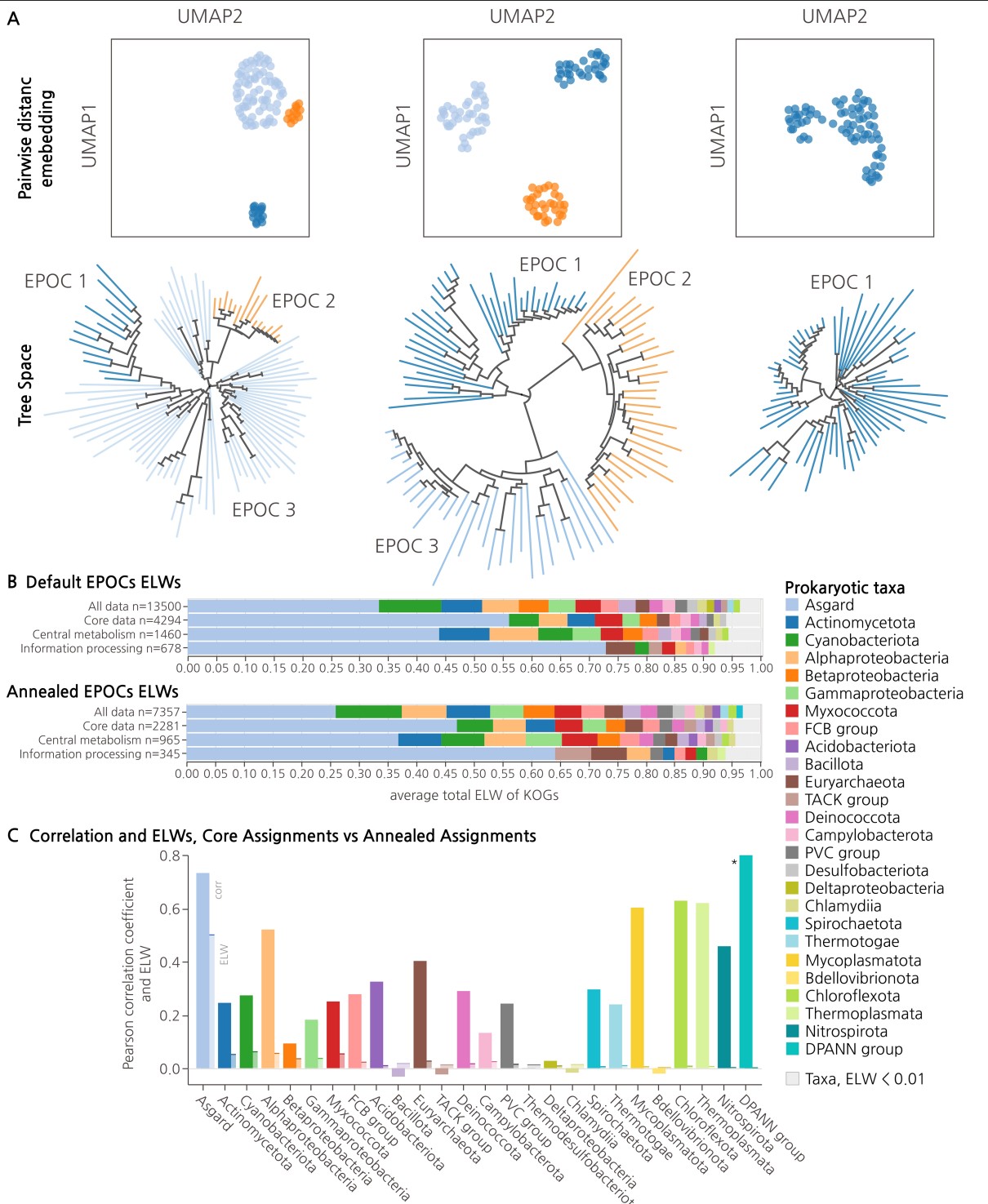

**Extended Data Fig. 3 | Annealed cluster EPOCS. A)** Examples of annealed clades defined based on embedded and clustered pairwise distance matrices of phylogenetic trees. Top, embedding of pairwise leaf distance of phylogenetic tree. Bottom, clustering by HDBSCAN allowing single cluster partitions as shown to the right. Partitions form monophyletic subsets of well-partitioned branches. **B)** Resulting top level aELW values derived from the core data clusters as well as annealed EPOCS showing broad correlation between different core and annealed partitioning schemes. **C)** Correlation of aELW vectors calculated

for all prokaryotic taxa across all KEGG maps using the annealed and core EPOCs indicating good agreement for Asgard assignments but greater variance between prokaryotic taxa. Overlaid on global aELW values. Taxa for which aELW is less than 0.01 are found to be associated only for very few pathways and as such correlate strongly. *DPANN has an aELW of 0 and does not contribute to any pathways, and therefore with a correlation of 1.0 between the two reconstructions.

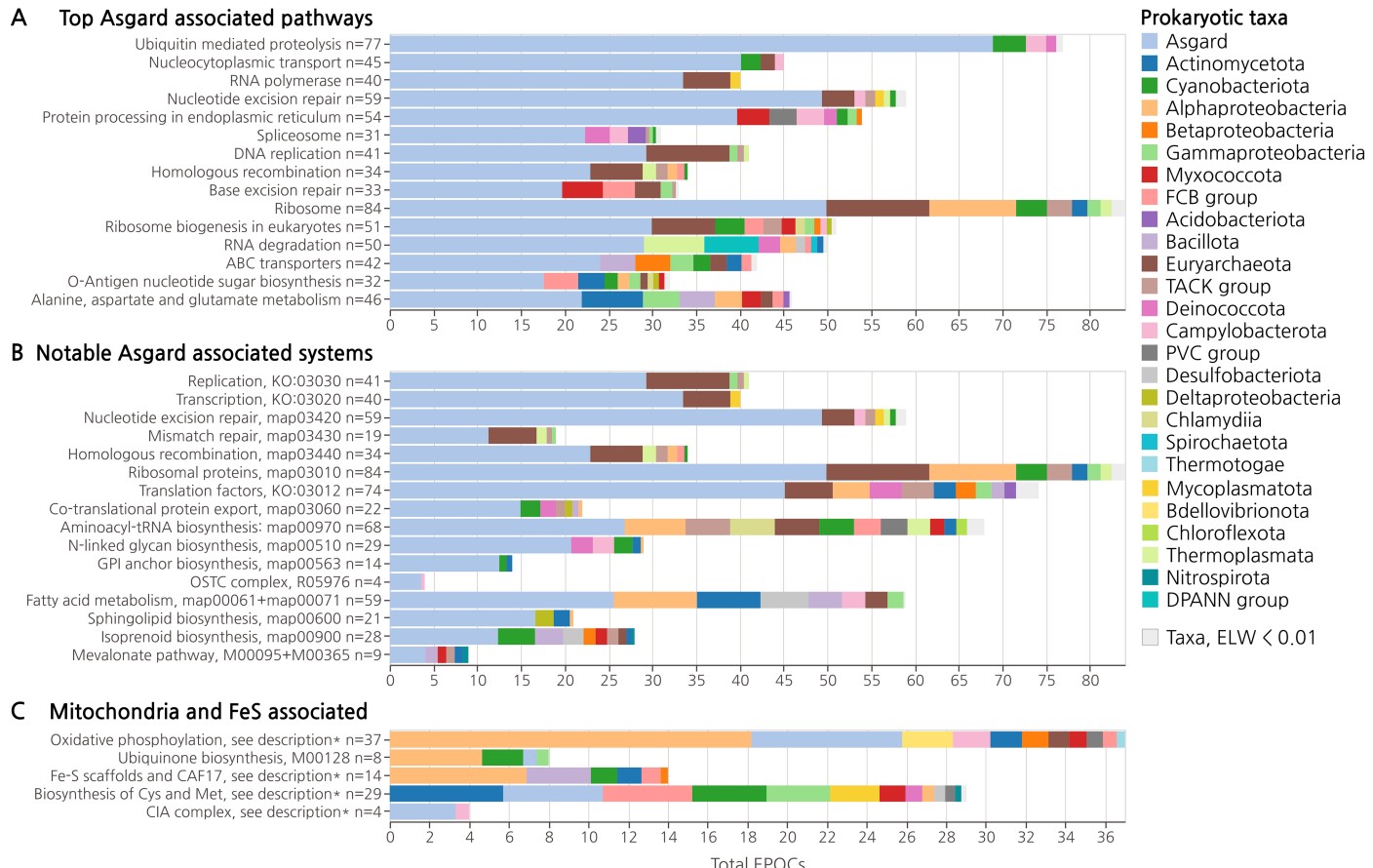

**A  Top Asgard associated pathways**

**B  Notable Asgard associated systems**

**C  Mitochondria and FeS associated**

Total EPOCs

**Prokaryotic taxa**

- Asgard
- Actinomycetota
- Cyanobacteriota
- Alphaproteobacteria
- Betaproteobacteria
- Gammaproteobacteria
- Myxococcota
- FCB group
- Acidobacteriota
- Bacillota
- Euryarchaeota
- TACK group
- Deinococcota
- Campylobacterota
- PVC group
- Desulfobacteriota
- Deltaproteobacteria
- Chlamydiia
- Spirochaetota
- Thermotogae
- Mycoplasmatota
- Bdellovibrionota
- Chloroflexota
- Thermoplasmata
- Nitrospirota
- DPANN group
- Taxa, ELW < 0.01

**Extended Data Fig. 4 | aELW breakdown of additional maps and pathways with prominent Asgard and alphaproteobacterial contributions. A**) Top 15 KEGG pathways with strongest Asgard aELW. KEGG maps with more than 20 EPOCs included. Taxonomy coloured as per Fig. 1. **B**) Further notable Asgard systems, aELW calculated over either KEGG "map" number (ex. map0000X), reaction number (R0000X), or module number (ex. M0000X). **C**) Oxidative

phosphorylation (map00190), also contains entries for V-type ATPases which are excluded here as they are not part of mitochondrial metabolism. Fe-S scaffolds are not members of any map; the manually curated set includes KOGs: K22063, K22072, K22068, K22073, K22070, K22071, K04043, K04044 and K04082. The CIA complex is likewise not grouped within BRITE and the manual set contains KOGs: K24730, K26403 and K15075.

# Most paraphyletic KOGs

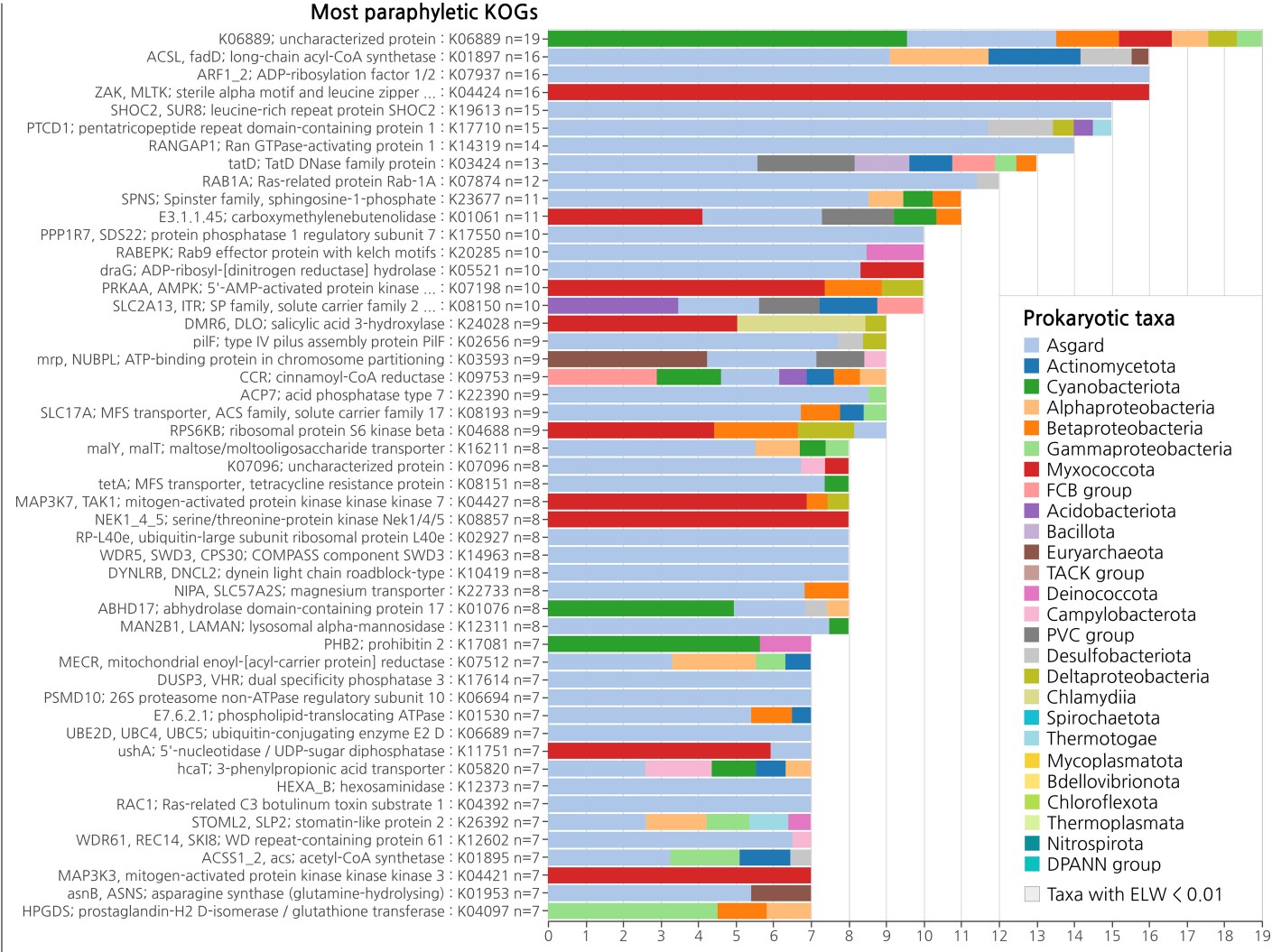

**Extended Data Fig. 5 | Most Paralogous KOGs.** Overview of most paralogous KOGs estimated by number of separate EPOCs traceable to the LECA.

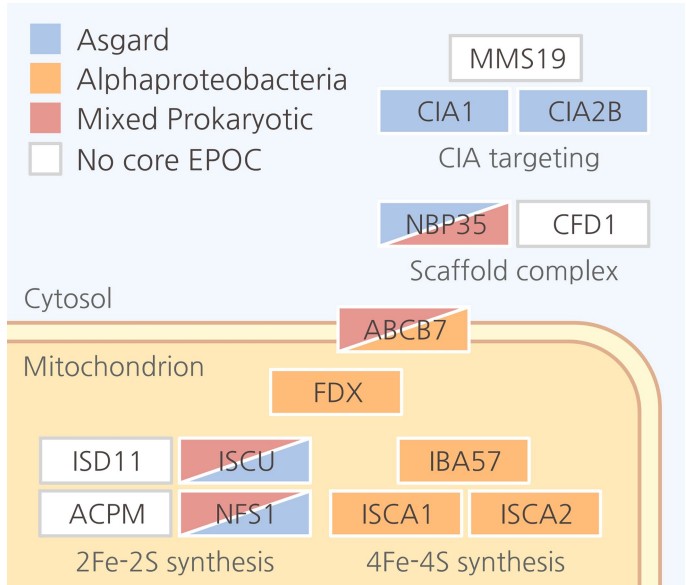

**Extended Data Fig. 6 | Ancestry of eukaryotic Fe-S cluster synthesis pathways.** Schematics for Fe-S cluster synthesis and assembly including downstream cytosolic Fe-S insertion are shown. Boxes indicate individual KOGs covered by EPOCs. Full colour indicates support for ancestry with an aELW > 0.5, boxes with mixed colour indicate unresolved ancestries with ELWs <0.5 for the two most likely estimates. Mixed prokaryotic ancestries represent all possible taxa, including Alphaproteobacteria.

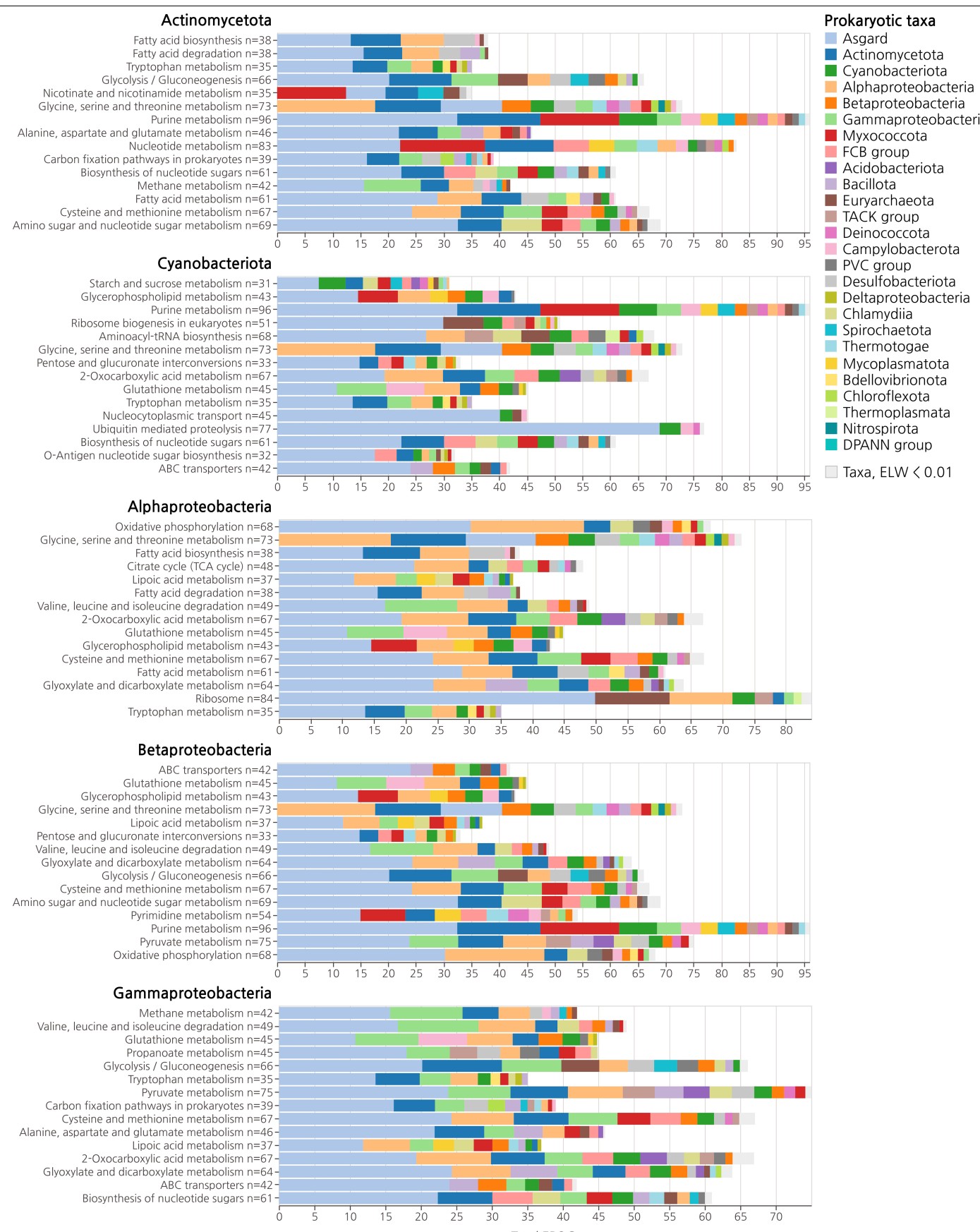

**Extended Data Fig. 7 | Strongest per taxa association of diverse bacteria to eukaryotes.** Top bacterial taxa associated taxa with aELW presented for their 15 most prominent pathways in KEGG Brite B. Pathways with more than 20 EPOCs shown.

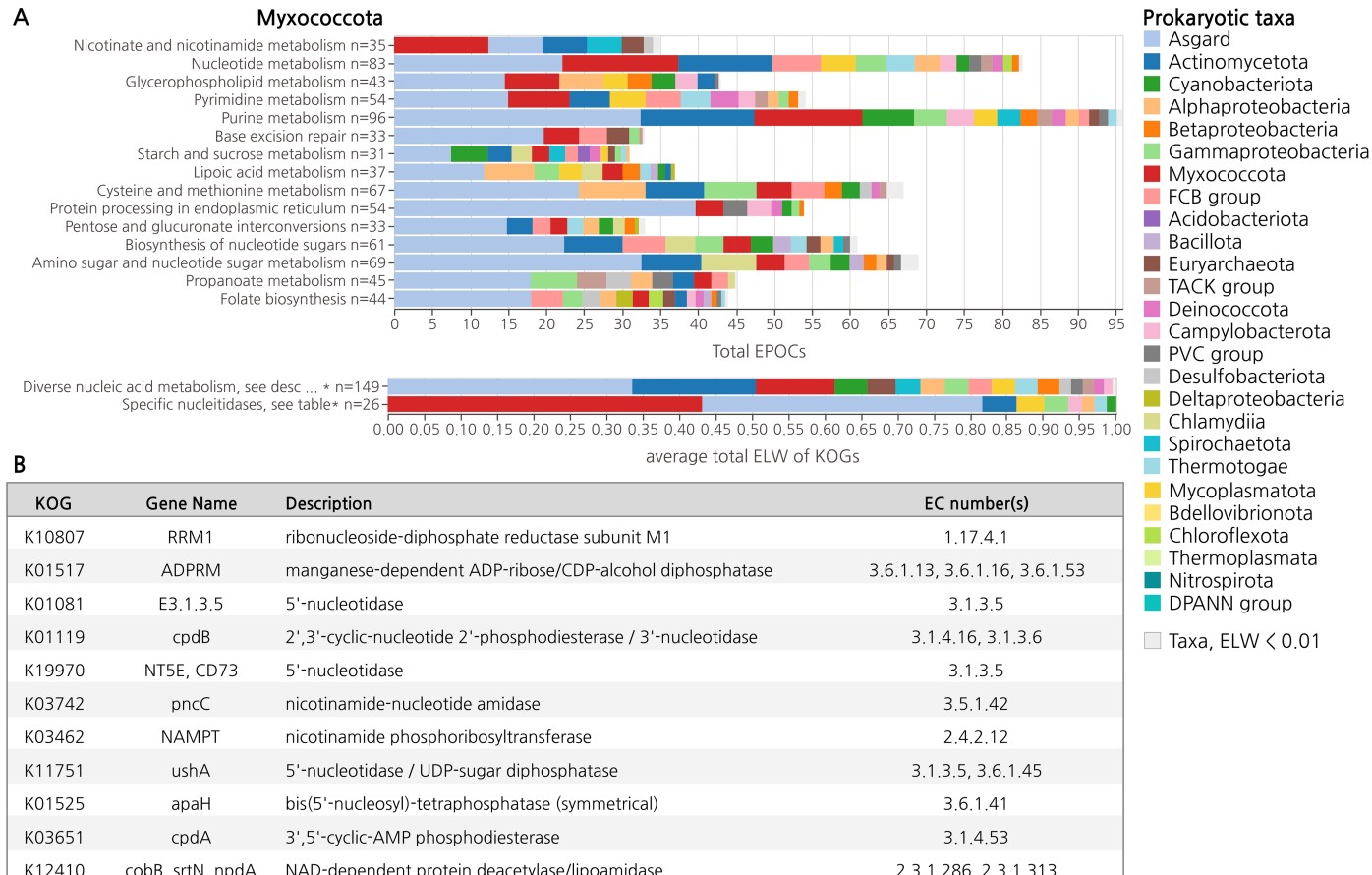

**A**) Myxococcota

| KOG | Gene Name | Description | EC number(s) |
|---|---|---|---|
| K10807 | RRM1 | ribonucleoside-diphosphate reductase subunit M1 | 1.17.4.1 |
| K01517 | ADPRM | manganese-dependent ADP-ribose/CDP-alcohol diphosphatase | 3.6.1.13, 3.6.1.16, 3.6.1.53 |
| K01081 | E3.1.3.5 | 5'-nucleotidase | 3.1.3.5 |
| K01119 | cpdB | 2',3'-cyclic-nucleotide 2'-phosphodiesterase / 3'-nucleotidase | 3.1.4.16, 3.1.3.6 |
| K19970 | NT5E, CD73 | 5'-nucleotidase | 3.1.3.5 |
| K03742 | pncC | nicotinamide-nucleotide amidase | 3.5.1.42 |
| K03462 | NAMPT | nicotinamide phosphoribosyltransferase | 2.4.2.12 |
| K11751 | ushA | 5'-nucleotidase / UDP-sugar diphosphatase | 3.1.3.5, 3.6.1.45 |
| K01525 | apaH | bis(5'-nucleosyl)-tetraphosphatase (symmetrical) | 3.6.1.41 |
| K03651 | cpdA | 3',5'-cyclic-AMP phosphodiesterase | 3.1.4.53 |
| K12410 | cobB, srtN, npdA | NAD-dependent protein deacetylase/lipoamidase | 2.3.1.286, 2.3.1.313 |

**Extended Data Fig. 8 | Myxococcota association with eukaryotes.**
**A**) Top, aELW breakdown of enriched Myxococcota pathways showing primary association with nucleotide synthesis. Bottom, curated set of Myxococcota enriched EPOCs containing general nucleic acid metabolism and modification pathways corresponding to KEGG map00760, map01232, map00240 and map00230 with notable presence of Myxococcota. Further selection of only those KOGs within the above pathways which include Myxococcota. **B**) List of KOGs with Myxococcota presence within the curated set of nucleotide synthesis highlighting phosphatases and phosphotransferases.

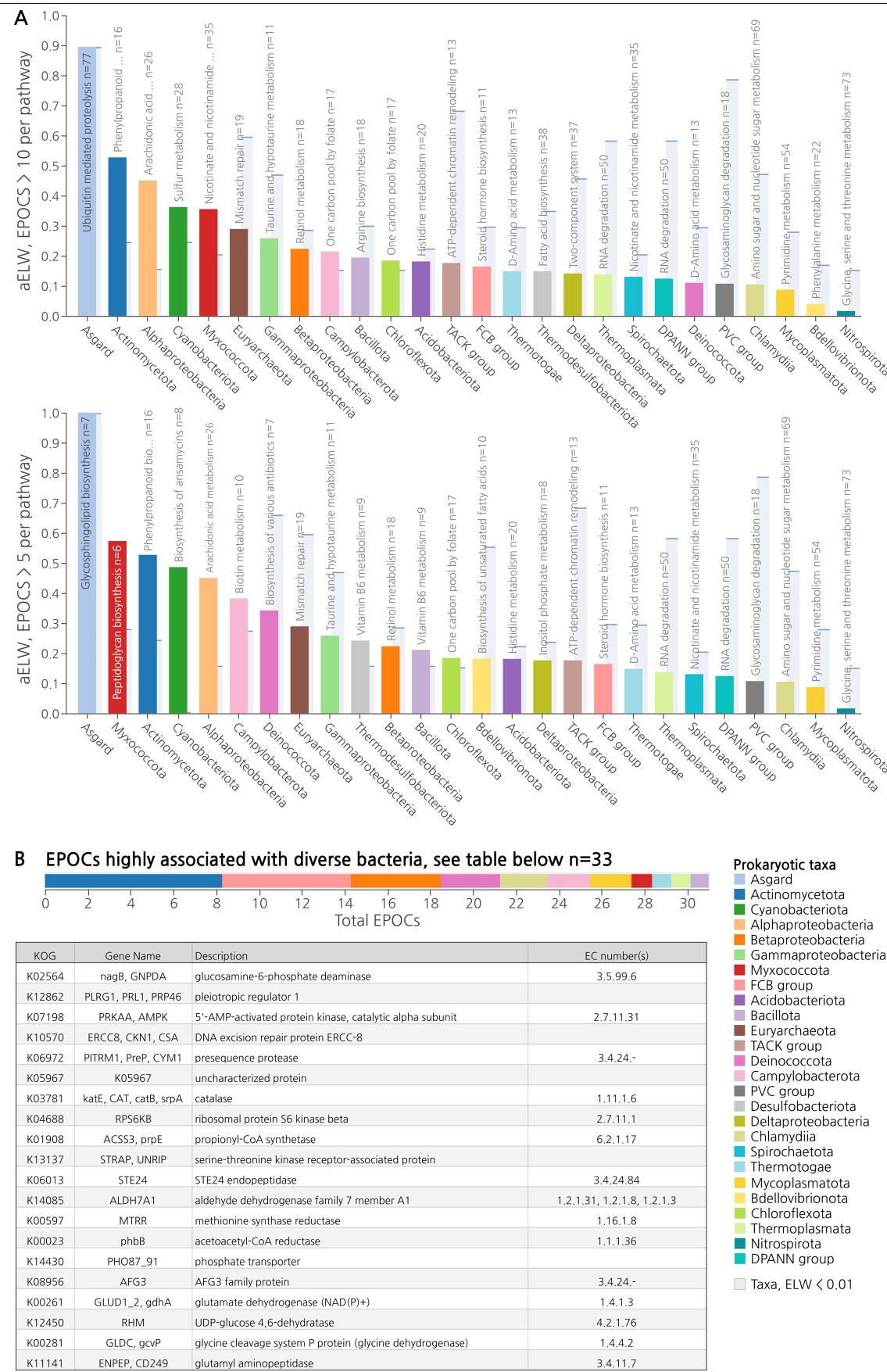

**Extended Data Fig. 9 |** See next page for caption.

**Extended Data Fig. 9 | KOGs with strong prokaryotic association. A)** Same as main Fig. 2 under less stringent selection criteria indicating most prominent prokaryotic pathways with a coverage of at least 10 and 5 EPOCs, aELW shown relative to that of Asgard for the same pathway in pale blue. **B)** Individual EPOCs tightly associated with diverse prokaryotic taxa under stringent selection criteria. EPOCs must contain a eukaryotic outgroup with at least 15 unique taxa encompass the LECA, have a prokaryotic sister clade with an ELW of >0.7 which is not derived from Asgard, Euryarchaeota, the TACK group, DPANN group, Cyanobacteriota or Alphaproteobacteria and contain more than 20 sequences. The EPOCs which meet these criteria are highlighted with respective KOGs indicated, aELW displayed.

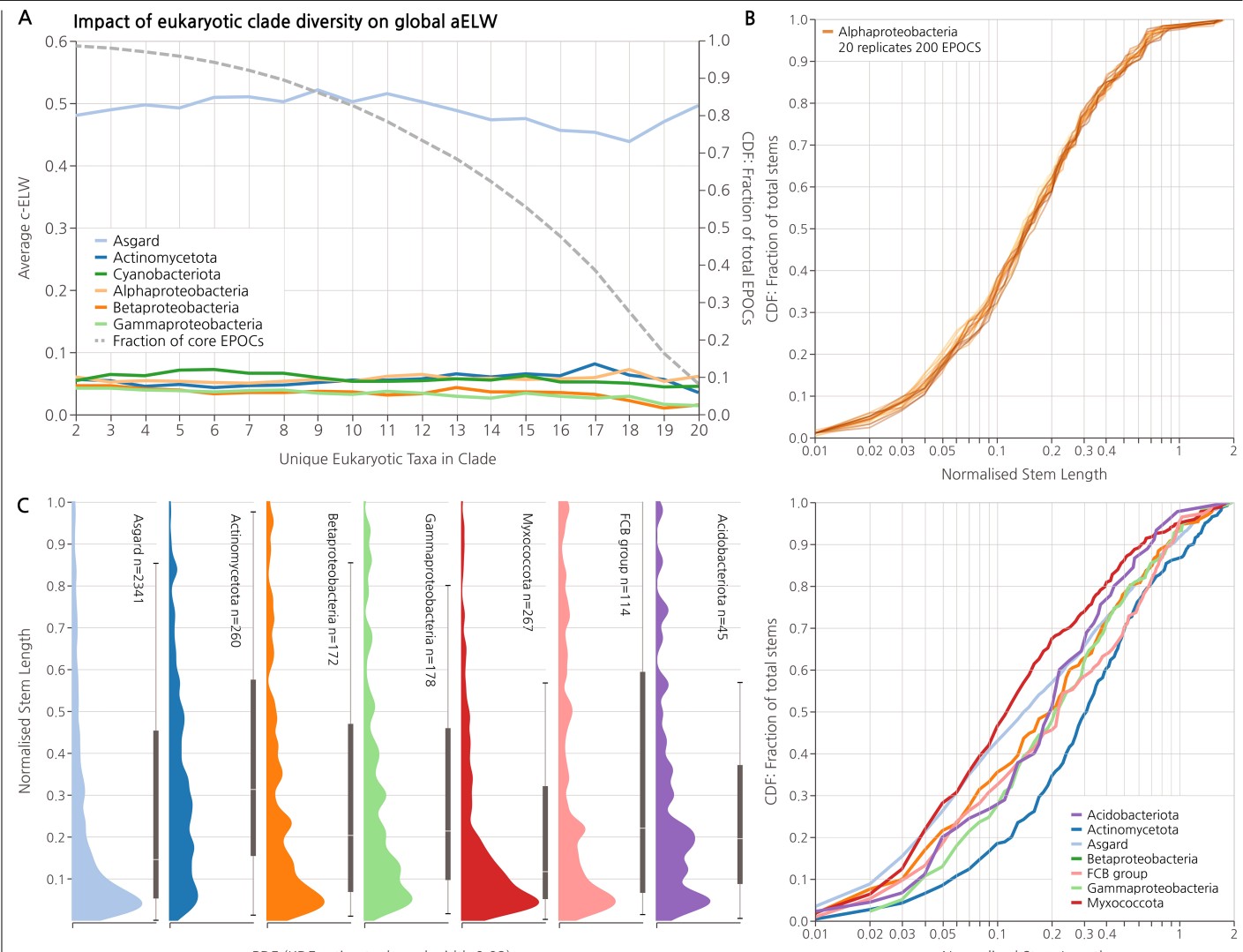

**Extended Data Fig. 10 | Clade scope definition and stem length distribution for diverse bacteria compared to Asgard. A**) Global aELW dependence on choice of clade size cutoff using core data. Estimated aELW are independent on choice of eukaryotic clade scope. **B**) 20 full technical replicates, including muscle5 alignment, IQtree2 calculations, annotation and constraint tree generation, of 200 randomly sampled alphaproteobacterial stems showing reproducibility of stem length distributions. Compared with distributions drawn from Asgard and cyanobacterial stems as per Fig. 3. **C**) Left, Probability Density Function (PDF) of normalized stem lengths taken from samples of individual core eukaryotic genes associated with prokaryotic taxa, annotated as per Fig. 3.

# Reporting Summary

## Statistics

For all statistical analyses, confirm that the following items are present in the figure legend, table legend, main text, or Methods section.

| n/a | Confirmed | |
|---|---|---|
| ☐ | ☒ | The exact sample size (*n*) for each experimental group/condition, given as a discrete number and unit of measurement |
| ☐ | ☒ | A statement on whether measurements were taken from distinct samples or whether the same sample was measured repeatedly |
| ☐ | ☒ | The statistical test(s) used AND whether they are one- or two-sided<br>*Only common tests should be described solely by name; describe more complex techniques in the Methods section.* |
| ☒ | ☐ | A description of all covariates tested |
| ☒ | ☐ | A description of any assumptions or corrections, such as tests of normality and adjustment for multiple comparisons |
| ☒ | ☐ | A full description of the statistical parameters including central tendency (e.g. means) or other basic estimates (e.g. regression coefficient) AND variation (e.g. standard deviation) or associated estimates of uncertainty (e.g. confidence intervals) |
| ☒ | ☐ | For null hypothesis testing, the test statistic (e.g. *F*, *t*, *r*) with confidence intervals, effect sizes, degrees of freedom and *P* value noted<br>*Give P values as exact values whenever suitable.* |
| ☒ | ☐ | For Bayesian analysis, information on the choice of priors and Markov chain Monte Carlo settings |
| ☒ | ☐ | For hierarchical and complex designs, identification of the appropriate level for tests and full reporting of outcomes |
| ☒ | ☐ | Estimates of effect sizes (e.g. Cohen's *d*, Pearson's *r*), indicating how they were calculated |

*Our web collection on statistics for biologists contains articles on many of the points above.*

## Software and code

Policy information about availability of computer code

| Data collection | Sequence data was retreived from NCBI Genome https://www.ncbi.nlm.nih.gov/datasets/genome/ as well as the SRA https://www.ncbi.nlm.nih.gov/sra. Eukaryotic sequences were obtained as a part of Eukprot v3. https://evocellbio.com/eukprot/ |
|---|---|
| Data analysis | Python 3.9, Prodigal 2.6.3, ete3 3.1.3,  mmseqs2 15-6f452, fasttree 2.1.1, famsa 2.0.1, muscle 5.1,  IQtree2 2.3.5 |

For manuscripts utilizing custom algorithms or software that are central to the research but not yet described in published literature, software must be made available to editors and reviewers. We strongly encourage code deposition in a community repository (e.g. GitHub). See the Nature Portfolio guidelines for submitting code & software for further information.

## Data

Policy information about availability of data

All manuscripts must include a data availability statement. This statement should provide the following information, where applicable:
- Accession codes, unique identifiers, or web links for publicly available datasets
- A description of any restrictions on data availability
- For clinical datasets or third party data, please ensure that the statement adheres to our policy

All initial data are avaialable at NCBI Genomes, SRA or via EUkprot as indicated above. Initial parsed and taxonomy annotated databases, intermediate clustering data and HMM profile databases as well as parsed data and all intermediate files been deposited to Zenodo under https://zenodo.org/records/14002645 and https://zenodo.org/records/14004407

# Research involving human participants, their data, or biological material

Policy information about studies with [human participants or human data](). See also policy information about [sex, gender (identity/presentation), and sexual orientation]() and [race, ethnicity and racism]().

| | |
|---|---|
| Reporting on sex and gender | n/a |
| Reporting on race, ethnicity, or other socially relevant groupings | n/a |
| Population characteristics | n/a |
| Recruitment | n/a |
| Ethics oversight | n/a |

Note that full information on the approval of the study protocol must also be provided in the manuscript.

# Field-specific reporting

Please select the one below that is the best fit for your research. If you are not sure, read the appropriate sections before making your selection.

☐ Life sciences       ☐ Behavioural & social sciences       ☒ Ecological, evolutionary & environmental sciences

For a reference copy of the document with all sections, see [nature.com/documents/nr-reporting-summary-flat.pdf]()

# Ecological, evolutionary & environmental sciences study design

All studies must disclose on these points even when the disclosure is negative.

| | |
|---|---|
| Study description | A method based on evolutionary constraint trees constructed from sets of homologous proteins was used to infer evolutionary ancestry for Eukaryotic protein families. |
| Research sample | All initial data consisted of protein sequences taken from prokaryotic and eukaryotic sequence databases. |
| Sampling strategy | Prokaryotes were sampled as widely as possible though manual curation of the NCBI genome database. A wide Eukaryotic representation was ensured though the usage of the Eukprot database. |
| Data collection | Sequences were retrieved from databases either though wed download or the NCBI ftp service. |
| Timing and spatial scale | n/a |
| Data exclusions | n/a |
| Reproducibility | n/a |
| Randomization | n/a |
| Blinding | n/a |

Did the study involve field work?       ☐ Yes       ☒ No

# Reporting for specific materials, systems and methods

We require information from authors about some types of materials, experimental systems and methods used in many studies. Here, indicate whether each material, system or method listed is relevant to your study. If you are not sure if a list item applies to your research, read the appropriate section before selecting a response.

## Materials & experimental systems

| n/a | Involved in the study |
|---|---|
| ☒ ☐ | Antibodies |
| ☒ ☐ | Eukaryotic cell lines |
| ☒ ☐ | Palaeontology and archaeology |
| ☒ ☐ | Animals and other organisms |
| ☒ ☐ | Clinical data |
| ☒ ☐ | Dual use research of concern |
| ☒ ☐ | Plants |

## Methods

| n/a | Involved in the study |
|---|---|
| ☒ ☐ | ChIP-seq |
| ☒ ☐ | Flow cytometry |
| ☒ ☐ | MRI-based neuroimaging |

## Plants

| Seed stocks | n/a |
|---|---|
| Novel plant genotypes | n/a |
| Authentication | n/a |

