## [Peer Review File · Nature]

Dominant contribution of Asgard archaea to eukaryogenesis

Corresponding Author: Dr Eugene Koonin

Version 1:

Reviewer comments:

Referee #1

(Remarks to the Author)

The origin of eukaryotes represents an important knowledge gap in the evolution of life on Earth. The eukaryotic cell evolved via an endosymbiosis between an archaeal host cell and a bacterial (alphaproteobacterial) endosymbiont. Recent work has revealed that the archaeal host was closest related to the Asgard archaea, and that genomes of these archaea encode many genes that are closest related to eukaryotes. Yet, an overall picture of the identity and genetic contribution of the Asgard archaea in the process of eukaryogenesis is still lacking. The manuscript by Tobiasson et al aims to provide new insights into this hiatus, by tracing the evolutionary origin of eukaryotic genes within a 'rigorous statistical framework'. Their analysis reveals a dominant genetic contribution of Asgard archaeal origin, and relatively limited contributions originating from the mitochondrial endosymbiosis. If this would be convincingly shown, this would be an important finding that bears relevance for models of eukaryogenesis.

Several previous studies have focused on the very same subject (e.g. Pittis and Gabaldon, Nature, 2016; Vosseberg et al, NEE, 2020; Bernabeu et al, BioRxiv, 2024). Generally, the Asgard archaeal contribution to eukaryotic gene content has been estimated much lower in these studies. Hence, the claims made in the paper by Tobiasson stand or fall with the rigour of the phylogenetic methodology that has been used to generate the results on which these claims are based. This review will therefore specifically focus on assessing the appropriateness of the methodology that was used. As detailed below, several technical issues have been identified that could potentially account for the inflated Asgard archaeal contributions to eukaryotic gene content. These issues need to be carefully addressed.

Furthermore, as a general comment, the clarity and use of language of the manuscript could be improved. Some of the observed issues (incorrect/missing references, missing figure legend, figure not mentioned in main text, etc) indicate that the manuscript was written in a hurry. In addition, and importantly, the methodology section is not sufficiently detailed to reproduce the analyses. The provided Zenodo link does not work, hence there is no access to supplementary datasets.

Main comments on the methodology:

1. EPOCs/clustering/oversplitting.

A main claim by Tobiasson et al involved the 'dominant contribution' of Asgard archaea to eukaryotic gene content. The quantification of this 'dominant contribution' is rather confusing, as different numbers are provided (e.g. see lines 157-171, and compare with Fig 1). In any case, the alleged Asgard contribution is much higher than was observed in other studies (Pittis and Gabaldon, Nature, 2016; Vosseberg et al, NEE, 2020; Bernabeu et al, BioRxiv, 2024), which raises the question where these higher numbers of Asgard contributions stem from. Could these higher numbers be the result from how eukaryotic paralogs are handled? The applied clustering approach likely results in protoeukaryotic paralogs forming individual eukaryotic clusters. Yet, the homologous prokaryotic clusters are probably combined with each (paralogous) eukaryotic cluster, resulting in multiple EPOCs for these duplicated families. Could it be that, given that eukaryotic genes of archaeal origin have been shown to have predominantly duplicated (Vosseberg et al., NEE, 2021), this explains why the Asgard contribution is inflated compared with previous studies (including a recent preprint, Bernabeu et al, BioRxiv, 2024)? Given massive duplications in several eukaryotic protein families (e.g. small GTPases, actin homologs, ubiquitin(-related) families, etc), effects could be significant.

Another issue involves that many EPOCs are seemingly not taken into account for quantifying the origins of prokaryotic

contributions since they fall outside the chosen parameters/definitions (e.g. soft LCA cut-off; see line 638-645). By omitting these EPOCs, contributions of specific taxonomic classes are artificially inflated.

2. Phylogenetic analyses.

A. The phylogenetic analyses that were used to identify the prokaryotic source of eukaryotic gene content represents a key aspect of the analyses by Tobiasson et al. In several cases it is unclear how the applied definitions and decisions have affected the results and conclusions in the manuscript.

For example, to assign taxonomic lades, a 'soft LCA' score is calculated (defined at lines 635-636), which also includes a division by the 'total number of label X' (line 636). Does this mean that clades that potentially include the sister group in the ML tree are discarded if their main taxonomic label is present in multiple clades in the tree? For example, if the sister clade has 1 gammaproteo and 2 alphaproteos, with a single gamma and 10 alphas in the full tree, the soft LCA score of this clade would be: $\alpha = (2/3) * (2/10) = 0.133$; $\gamma = (1/3) * (1/1) = 0.333$; both are smaller than 0.8 so this clade is discarded. In that case, the results could be simply due to Asgard archaea being much more likely to form a monophyletic group compared with alphas and other bacterial groups. The authors should clarify this in more detail, and show how their results are affected by different cut-offs.

Furthermore, the presented results are likely affected by the choices that the authors have made with respect to taxonomic labels. The taxonomic levels that were chosen by the authors differ for different taxonomic groups. For example, while mostly (super)phylum-level taxonomic groups are used for archaea (Asgard, TACK, DPANN, Euryarchaeota), the choice is rather heterogeneous for bacterial groups (e.g. Alpha/Gammaproteobacteria at class level, but other groups such as FCB and PVC at superphylum level). While the NCBI taxonomy was probably used for practical reasons to define taxonomic classes, it is problematic since the underlying classification is based on a multitude of many different (often conflicting) studies. To make sure that the applied taxonomic labeling does not result in misleading results, the authors should benchmark their observations against a standardized microbial taxonomy such as the GTDB. Surely the taxonomic classification of GTDB is also not perfect, but for sure less arbitrary than the current NCBI taxonomy.

B. Another issue involves the identification of the closest prokaryotic clade of eukaryotic proteins in phylogenetic trees. In order to infer the 12 closest clades (line 661), the tree has to be rooted. It is not described in the manuscript how this is done. Extended Data Figure 2 seems to imply that the tree is rooted on the eukaryotic clade. If this is the case, it is not possible to infer closeness based on topological distance.

C. Finally, the authors surprisingly report that eukaryotic proteins of Asgard archaeal origin display shorter stem lengths compared to eukaryotic proteins of e.g. Alphaproteobacterial descent (Figure 5B). This result not only contrasts with the results reported in previous studies (including the aforementioned preprint from Bernabeu et al.), but it is also contradicting with what one typically observes in single gene trees and trees based on supermatrices: a long eukaryotic stem for archaeal-derived genes and shorter ones for alphaproteobacteria-derived genes. This is also a main reason why mitochondrial genes have been used to root the eukaryotic tree of life in recent studies. It is unclear what causes this aberrant result, but the biological explanation provided in lines 367 and onwards are rather speculative/unconvincing. The authors should exclude that a technical reason is underlying this contrasting result.

3. Figures 2B (not referred to in the main text...) and 3 provide examples of eukaryotic pathways that display dominant Asgard archaeal contributions. The authors should provide phylogenetic trees as supplementary data that convincingly support these claims.

4. The provided Zenodo link does not work, hence it was not possible to access and inspect the data sets underlying the figures and claims.

Minor/specific comments:

Line 41-45. The contribution of de novo gene birth and the numerous gene duplications that occurred along the eukaryotic stem to the evolution of "key features of eukaryotic cell organization" is ignored in this line and the rest of the manuscript. Moreover, what does "the Asgard ancestor" refer to: the last common ancestor of all Asgard archaea, the LCA of eukaryotes and the closest Asgard group, or the host in endosymbiosis?

Line 68. Please note that the taxonomic (re-)classification of Asgard archaea is currently pending, as another proposal for the nomenclature has been published that aimed to retain the 'Asgard' nomenclature (Tamarit et al, Syst. & Applied Microbiology, 2024). I suggest you cite both these papers, or neither of the two.

Line 90-91. Perhaps such models would not require membrane replacement, but besides a second endosymbiosis, it would need complete retargeting of membrane proteome, so I would involve additional challenges compared to 'single-endosymbiosis' scenarios. I suggest mentioning this as well.

Line 113, incorrect reference (2932)?

Line 118. At his point it is unclear where these "26 curated prokaryotic taxonomic classes" come from. Please provide more context.

Line 126. Noun (probably "sequences") missing between "prokaryotic" and "were".

Line 300. EPOCs that are associated with the plastid ancestor are mentioned: how have these post-LECA clusters passed the LECA criteria set by the authors?

Line 438. "likely": based on which findings of their own or other studies do the authors think this is likely?

Line 464. Figure 6 is lacking a legend.

Line 469. Several other studies have provided complete Asgard archaeal genomes (Valentin-Alvarado et al, Genome Research, 2024; Wu et al Nature Microbiol, 2022).

Line 498. "All clusters which did not contain sequences from at least 67% of all bacterial families within a prokaryotic class were rejected from the database." Not all genomes are assigned to a family in the NCBI taxonomy: how did the authors take this into account?

Line 692-693. "Cases of $ELW < 0.99$ ", "<" should be " \geq ".

Figures

Figure 1B-C and similar figures: colours are recycled due to the high number of categories. According to the description, average ELW values are displayed but the figures show the number of EPOCs.

Figure 2. Not referred to in the main text. Colours not explained for panel 2A.

Figure 4. Not clear what the two different bars mean for each taxonomic group. If one of them is relative to the Asgard contribution, why is the Asgard bar then not at 1?

Figure 6. The proposed scenario is not new as similar models for eukaryogenesis have been proposed before, and it is unclear how the analyses/results described in the present paper have contributed to the proposed scenario.

(Remarks on code availability)

I have not inspected the code, but even if I could, the provided Zenodo link did not work, hence it would not have been possible to reproduce the analyses.

Referee #2

(Remarks to the Author)

Tobiasson et al.

Dominant contribution of Asgard archaea to eukaryogenesis

This paper is a very fresh take on the role of Asgard archaea in eukaryogenesis. I really like the approach taken in this paper. The authors take advantage of huge sequence databases available for bacteria and archaea and augment it with the rapidly expanding datasets for asgard (MAGs but also several now confirmed by having sequences from two or three (I believe) asgard in culture, thus ruling out concerns over MAG binning, assembly, etc.).

The take home message is that Asgard appear to have made a much larger and diverse contribution to the genes / proteins / processes present in LECA than originally believed. There may well be quibbles over some of the bioinformatic decisions made (e.g., the relatively small number of eukaryotes required to constitute an EPOC (page 21)), but on balance I don't have any problems with the methodologies. I think the right balance has been struck for the desired high-level questions being asked of the data.

I particularly like the stem-length (re)analysis, which is a way of potentially ordering temporally prokaryotic contributions to eukaryotic evolution the authors conclude that in fact they can't do that reliably with their data. Instead they invoke bursts of sequence divergence associated with varying difficulties associated with adaptation to different cellular environments (in the case of bacterial donors to eukaryotes). The idea of varying rates of adaptive evolution has been around for a long time -- I'm thinking originally when looking at things like actins and tubulins, which are slowly evolving within eukaryotes but when compared to their very distant prokaryotic homologs, appear to have diverged before the origin of life on Earth. I have not come across this invocation recently and it makes a lot of sense in the present context.

One potential oversight is not acknowledging the genome mosaicism that clearly exists in prokaryotic genomes today (and euks to a lesser extent). This is relevant because we can only assume that genomes of all lifeforms were mosaic at the time that euks were first evolving, and this complicates the process of assessing donor genes / pathways / prokaryotic lineages. So I think the authors should at least acknowledge this and cite the paper(s) from Bill Martin's team, who impressed upon me (at least) that this was an issue to take seriously.

Specific questions and comments:

-Fig. 1 and associated text.

Regarding color coding for parts B and C, some colors are used multiple times. For example, orange is both Cyanobacteria

and Chloroflexa. I'm assuming this is because the number of colors is finite, and they loop around the same set. So top to bottom on the right column of prokaryotic taxa is most to least on the bar graphs in terms of average total ELW (B) or Total EPOCs). This is probably worth mentioning in the legend, because at first glance it is possible for someone to miss it, for example, choosing Chloroflexa (orange) as the second highest average total ELW instead of Cyanobacteria.

And what do you do if the trend is reversed for one specific histogram (e.g. Orange = Chloroflexa is higher than Orange = Cyanobacteria) compared to the rest? Would the reader know it? Or does the color coding hierarchy apply consistently to all data presented in B and C? The color orange caught my eye because Cyanobacteria is (I assume) the second highest after Asgard, despite the fact that this lineage should only be a factor for the phototrophic groups of eukaryotes.

Finally, what about the lightest gray bars on the extreme right of most of the histograms? This is the lightest of the three shades of gray used, and it doesn't have a code in the 'Prokaryotic taxa' color scheme on the right, despite the fact that as a total it is much larger than most of the other prokaryotic contributions. I assume it is 'unassigned' or something like that.

-Fig 4. I recommend the authors remove the gray vertical and horizontal hatch lines in the background, which make it difficult to read the vertically oriented functional classifications for all bars except the first one for Asgard.

-Line 230: "Although strong Asgard associations were observed for large parts of the overall metabolic network, we observed a high degree of mosaicism in the pathways for fatty acid synthesis and decay." Do the authors mean mosaicism between the pathways for fatty acid synthesis / decay, and/or mosaicism in the signals of the proteins within individual pathways? This got me thinking about the extent is the ELW/EPOC methodology able to detect evolutionary mosaicism within a pathway, e.g., gene/protein X for first step in a biochemical pathway is clearly Asgard origin for eukaryotes, but gene/protein Y for second step is of bacterial origin? I originally assumed that it can't do that, because for example Carbohydrate metabolism (n=423) considers all enzymes together but does not break them down further.

But then on line 243, the authors say: "Here we found the mevalonate pathway, from Acetyl-CoA to mevalonate and further to Farnesyl and Geranyl diphosphate, to be strongly Asgard-associated (Extended Data Figure 3), with the key enzymes hydroxymethylglutaryl-CoA synthase (HMGCS), mevalonate kinase (MVK), phosphomevalonate kinase (PMVK), mevalonate diphosphate decarboxylase (MVD) being clearly Asgard-derived." So what does "strongly associated" here mean? It looks like all enzymes in this particular pathway, but how does the reader infer this from Extended Data Fig 3 (which has n=41 for isoprenoid biosynthesis).

All this is to say that the authors should clarify how the issue of mosaicism is inferred and presented (or not presented), and mosaicism at what level. Figure 2 shows "split backgrounds", but Fig. 2 is not referred to in the text, nor is it clear (to me) how the mosaicism identified and elucidated. The issue of mosaicism comes up again in the context of Figure 3 and associated discussion as well.

-Line 315: "The nucleotide-related associations with Myxococcota were primarily limited to phosphatases and phosphoribosyltransferases acting on nucleotide sugars including 5 and 3' nucleotidases, and the respective EPOCs showed little to no competing Asgard association. While noteworthy, these associations were limited to a few unique KOGs whereas all other associations with Myxococcota remained scattered across various pathways (Extended Data Figure 5)." Looking at that figure, I can see "competing Asgard association" in all data presented. This needs to be better described so that the reader can understand how the inferences have been made. It's not obvious from the figure alone (at least part A; part B doesn't allow one to tell, other than that this list of KOGs has Myxococcotan presence). Also, is "Myxococcotan" a typo?

-Line 430. At this point the authors should cite the papers actually showing that extant asgard that have been studied actually have a cytoskeleton, namely Rodrigues-Oliveira et al. Nature 2022 (this paper is cited later for a different reason). As written, it implies that this is a prediction from the results in the paper, when in fact it is also supported by experimental data. The field has come a long way from the first description of Asgard from MAGs, the presence of ESPs, and the debate about what those proteins actually did (or did not do) in the cell, and I would encourage the authors to underscore this.

-Line 73—Just like above. Yes, cytoskeletal proteins and membrane proteins, but make it clear that these actually DO eukaryote-like things. This was far from clear early on when they were flagged as ESPs when the first Asgard MAGs were published.

- Fig S1 –

-I find it hard to believe that there are more euk. sequences from Dinophyceae (dinoflagellates) than for all of Opisthokonta. Is this because of approach of / composition of Richter et al. 2022 EukProt dataset? Maybe it's massive numbers of RNA-seq data from a relatively small number of dinoflagellates with very large, gene-rich genomes? Dinophyceae are one of the more poorly sampled groups of algae, so this stands out as an outlier.

-Another point for these data is Fungi incertae sedis, Microsporidia, and Choanoflagellata listed separate from the Opisthokonta. The Opisthokonta include animals AND fungi (and microsporidia are related to fungi) AND choanoflagellates. I am not sure of the implications here for downstream analyses but it should be noted and (if need be) addressed.

-Where are the Rhodophyta (red algae)? I see Rhodelpheia listed, but not Rhodophyta. There are more Rhodophyta genomes / transcriptomes available than Rhodelpheia, so this is an oversight. Maybe they are accidentally included under Chlorophyta?

- Line 145. What happens when all eukaryotes are not monophyletic, e.g., a particular euk. lineage has a non-orthologous replacement from a prokaryote? This section underscores the robustness of the EPOC procedure as being tolerant to LBA

and non-homogenous clades, as well as paraphyly. Since the EPOC procedure ultimately treats euk. Clade as a “single datapoint”, I suspect this is taken into account and tolerated. But I would suggest the authors explicitly call out the robustness of their methodology to lineage-specific LGT / non-orthologous replacement of otherwise broadly distributed genes in the “soft-core” pan-genome. Perhaps this is all taken into account in the section on page 686 (Data filtering and removal of likely HGT cases).

(Remarks on code availability)

Referee #3

(Remarks to the Author)

In this article, Tobiasson and colleagues attempt to evaluate the relative contributions of several lineages in eukaryogenesis. To this end, they first collect from public databases a dataset of nearly a thousand eukaryotic genomes, and about 47 000 complete genomes of 7900 prokaryotic species enriched with nearly 150 metagenome assembled genomes of Asgard archaea. They then compute the LECA (last eucaryotic common ancestor) proteome and use phylogenetic analyses to find the prokaryotic clade of origins of this ancestral eucaryotic proteome. They find a largely dominating contribution of Asgard archaea to the proteome of LECA – as expected were this lineage to be the host of eucaryogenesis, and second, of Alphaproteobacteria, which are thought to be close to the ancestor of the mitochondrion. The authors look at the relative contribution of different prokaryotic lineages at the taxonomic and then functional level and mostly find no evidence for a consistent contribution to coherent functions from other lineages than Asgard and Alphaproteobacteria. They then re-employ a tree metric called ‘stem length’ used in a previous study by Pittis and Gabaldón (Nature 2016), which analysis had then led to the proposal of a late mitochondrion acquisition along eukaryogenesis and to the proposal that several other bacterial lineages contributed genes to ancestral eucaryotes, not only Alphaproteobacteria. The revisited analysis by Tobiasson and colleagues, in addition to the ones mentioned above lead the authors to contradict the conclusions of the 2016 article, as Tobiasson and colleagues do not seem to find consistent results. Instead, they say their results rather point at a dominant Asgard contribution first, and secondly, a Alphaproteobacterial contribution to the ancestor of eucaryotes, whenever other bacterial lineages had only a little, non-significant contribution.

The article is well-written on a hotly debated topic that would be of interest to the broad audience of Nature, but I have several major concerns regarding the methodology and dataset used, which I think could affect the main conclusions of this study. Further, I sometimes did not agree with the conclusions of the authors drawn from their results, as I found them unjustified. Please find below my major comments.

Major comments:

1. The dataset

- The authors employ complete genomes of prokaryotes from the NCBI database, and only include MAGs (metagenome assembled genomes) for Asgard archaea, which I found not justified. Indeed, this makes the known Asgard diversity much more well covered, while this also artificially inflates the number of Argard genomes considered comparatively to that of other lineages. More and more organisms are covered solely by sequence evidence (this is the case for Asgard, but also numerous bacterial lineages) and their genomes are never closed. Hence, most of the recently discovered diversity of bacteria and archaea is not covered by complete genomes, and by the dataset being used in this study. By solely focusing on complete genomes, the dataset is highly biased towards certain groups (often those being cultured), while ignoring others (this is visible on extended Figure 1). This would directly affect the taxonomic identification of prokaryotic donors to ancestral eucaryotes as some lineages are not covered at all, or barely covered. Solving what happened during eucaryogenesis is a very difficult endeavor, and I think that to this end, the most should be done of the data available. What was done here does seem not to be it, given the high amount of available genomic information that was not included. The authors acknowledge themselves in the conclusions on line 466-467 that a limited sampling is an issue, but I believe this is a serious shortcoming here.

- Regarding the eucaryotic dataset, could the authors provide some information on how well they think the chosen dataset covers the known eucaryotic diversity and the lineages with sequence data available?

2. The reconstruction of the LECA proteome and of its origins

- The authors claim they reconstructed the LECA proteome, but I found the method employed quite raw, and I believe this method should be better justified, or adapted to the difficulty of the task. The authors use a census criterion, but do not seem to really use a phylogenetic approach to compute the LECA proteome. Could the authors explain how the proposed approach could clearly identify vertically inherited gene families that trace back to the LECA?

- About the inference of the procaryotic origins, I find unjustified the choice to limit the analysis to families that are highly distributed within a procaryotic class (Methods, line 493-496 “All clusters which did not contain sequences from at least 67% of all bacterial families within a prokaryotic class were rejected from the database.”). Again, this choice made in the families to be considered in the analysis could bias the results, as the fact that a gene is not broadly taxonomically spread does not mean that it could not have been contributed to the LECA proteome. Moreover, the choice being based on the NCBI taxonomy that may have uneven phylogenetic depths makes it even more questionable. Could the authors please comment on this?

- I am surprised to see so many genes inferred that could be of Cyanobacterial origins (Fig 1B), while this is not discussed in the text. For instance, porphyrin metabolism is linked to a Cyanobacterial origin (Fig 4). If this corresponds to genes linked to

the later secondary endosymbiosis of the ancestor of the plastid, how could the authors explain this was inferred to be present in LECA? This high number of genes coming from Cyanobacteria is not really discussed, as it should be. To me the potential inference of genes of plastid origins in LECA could be a clear sign that the method to infer the LECA proteome is not accurate enough.

- I did not understand why the strong signal for a contribution of Actinomycetota was dismissed and judged irrelevant in the text without further explanations (Line 234-235). The authors should more strongly argue on why they do not think it is relevant, otherwise it could be interpreted as cherry-picking. Anyway, the fact that the results from the proposed methodology are questioned by the authors themselves makes me even more unsure of the validity of the proposed approach. Could the authors please explain?

- The case of the mevalonate pathway is also another point that questions some of the obtained results: the authors present it as a clear case of acquisition of a pathway from Asgard, but (1) the pathway seems to be not fully complete in the inference of LECA if I understood the figure correctly (Is it only missing PMD? Or are there other essential enzymes missing?) and (2) there are several enzymes that harbour a mixed signal (Asgard and other bacterial origins). Adding to my confusion (but maybe is this just a mistake or typo?), the main text contradicts Figure 2B regarding the origins of MVK (line 246): on one hand, it is said to be "strongly Asgard-associated" whereas on Fig 2B the enzyme is displayed as having a "mixed" signal for its origins. Could the authors please clarify this case as I'm unsure what to think about it?

- On line 471 and subsequently, could the authors justify why they think that the Asgard contribution is most likely an underestimate? Given the points discussed above, I'd rather say (and agree with the authors with what is said on lines 475-479) that it is rather that of the other bacterial lineages' that could be underestimated.

3. Functional interpretation

- The above-discussed case of the mevalonate pathway questions the unit of observation/integration that should be considered. Should individual gene families be considered, or rather sets of functionally coherent gene families pointing towards a same origin? The latter could enable to gather more phylogenetic signal. Could this be accounted for in the process of calculation of the relative prokaryotic lineages' contribution, rather than in an a posteriori manner? The authors make some steps in this direction by attempting to integrate the history of individual genes that are part of a same function, but I find the cutoff employed to investigate potential functional modules very drastic: on line 307, the authors consider only lineages having contributed more than 20 EPOCs to a pathway, and thus found only Alphaproteobacteria and Asgard. It would be interesting to investigate what comes out if this threshold is lowered.

- For instance, for the case of the Myxococcota, it is unclear if the authors could reproduce the results of previous studies where genes involved e.g. in steroid biosynthesis were found to come from this lineage (Hoshino PNAS 2021). Could the authors add a sentence about this in the paragraph on lines 312-310, or elsewhere?

- Later, another threshold is employed to look into prokaryotic contributions to individual EPOC, that of a number of sequences (lines 325-326). This directly points at the potentially strong impact of the biological sampling in the initial genomic dataset, and how it could artificially diminish the apparent contribution of under-sampled lineages. Same here, what if the threshold was lowered and the dataset enriched? Would more significant contributions from other prokaryotic lineages be leveraged?

4. The phylogenetic approach

I would need clarifications regarding the topology tests. For instance, as pictured on Figure 1A (or ext Data Fig 2), the ELW topology tests are ran as if the root of the gene trees was known, which is not the case. Could the authors please clarify the impact of an arbitrary choice of the outgroup on the results?

Minor comments:

- Figure 1. The color code should be changed as here it is a cyclic set of colors being used and the same color (for instance, orange) is employed more than once, making the identification of the prokaryotic lineages difficult and ambiguous.

- While membrane lipid origins is discussed, the origins of the nucleus is not. Could the authors add a couple of sentences about what is their view of the origins of the nucleus in the frame of a dominant contribution of Asgard and then Alphaproteobacteria to eucaryogenesis?

(Remarks on code availability)

Version 2:

Reviewer comments:

Referee #1

(Remarks to the Author)

Please see the attached file for details.

(Remarks on code availability)

Referee #2

(Remarks to the Author)

I thank the authors for having carried out a very thorough revision of their manuscript. Regarding my original comments, the authors have addressed my analytical and interpretive concerns, as well as those relating to data presentation. I have read the other two sets of comments and the authors responses as well. There were many important concerns raised, and it is clear that authors have taken them seriously, and when warranted, carried out additional analyses, some of which appear in the revised manuscript.

(Remarks on code availability)

Referee #3

(Remarks to the Author)

I thank the authors for this revised version and the discussion in the rebuttal, the methods are now more clearly described and the results are more nuanced and discussed in the manuscript. However, there are some of my concerns about the methodology employed that are still standing.

Major points:

- Effect of the taxonomy

Given the employed approaches that heavily rely on taxonomic assignments, it was much needed to investigate the effect of the taxonomy and test the genome-diversity normalized taxonomy from the GTDB. The authors have established that using the GTDB taxonomy was not affecting their main results, even if I think the proposed origins of eucaryotic core genes would have been more precise if the GTDB taxonomic grouping would have been kept all along, as the NCBI taxonomy can be somehow arbitrary for certain lineages. However, given the high rate of taxonomic assignment to the GTDB mentioned in the Methods (line 705), I do not understand why only a portion of the dataset seem to have be analysed on Extended Data Figure 1 (all data: "n = 5142", vs all data in Figure 1B: "n=14353"). Or is this a typo? Could the authors please explain?

- Ability of the phylogenetic method to trace genes back to the LECA.

In my initial review, I pointed out this issue and mentioned as an example the case of the Cyanobacterial contributions that seem to be at least partly false positives. On lines 98-100: "post-LECA horizontal transfer between eukaryotes and prokaryotes » are claimed to be dealt with ("prevent such genes from affecting our analysis »), but this is not the case, since e.g., the Cyanobacteria case is still prominent. The authors somewhat diverted my comment by answering they were not reconstructing the LECA proteome (I apologize for the semantic issue, I meant, as the authors state more precisely, "eukaryotic core genes tracing back to LECA"), leaving the case of transfers vs vertical transmission unanswered. Further, transfers also occur between eukaryotes. In the end, my question is: how can the proposed method really ensure that genes shared by at least 5 distinct taxonomic labels within eukaryotes trace back to their common ancestor if the vertical transmission is not really tested for? To me, this point remains unaddressed, and relying simply on taxonomic distribution to infer ancestry of the genes is too simplistic given the complexity of the question. Showing some phylogenetic trees for several cases could help convince about the validity of the approach (I believe this was also a request of reviewer #1). These points constitute severe limitations of the Method, which I think deserve at least a point in Discussion. Thus, the confidence of assessment of genes to FECA should be added in Discussion, along with a discussion on the case of the inference of genes of Cyanobacterial origins, maybe on lines 483-484 where the authors say "Further, our pangenome-based approach to exclude late HGT events from ancestral inference may overlook possible contributions from narrow or severely under-sampled clades".

Minor comments:

- On line 426, the authors write: "Other bacterial contributions appear to be piecemeal, coming from a continuous but non-specific capture of genes from diverse bacteria, occurring along the entire evolutionary path from Asgard to the LECA (Figure 6). ". I am not sure that the authors have much evidence from their analysis regarding the timing of acquisition, maybe nuance a bit and add "occurring possibly along the entire evolutionary path..."?

- Some of the figures in Sup Mat are not of good quality, for instance Extended Data Figure 6.

- In figure 5C, the legend for "Ribosome" genes is missing.

- Line 86-: "link" instead of "links"?

- Line 341: remove "astronomic" as unprecise and exaggerated?

- Line 676: "HMM" instead of "HHM".

- Line 717: typo at "redundant"

(Remarks on code availability)

Code/scripts are available, even though I have not reviewed the code extensively .

Version 3:

Reviewer comments:

Referee #1

(Remarks to the Author)

I thank the authors for their extensive rebuttal and efforts to alleviate the concerns that were raised previously. The authors state in their rebuttal that part of these concerns were considered "erroneous and unfair", but should realise that my intentions are sincere, and merely aimed to make sure that, given the contrasting results with previous works, the obtained results are not the result of methodological choices and/or artefacts, as this would do the scientific community no good. This is, simply put, my task as reviewer. It is therefore unnecessary for the authors to use a harsh tone in their rebuttal.

I would like the authors to kindly consider the following comments, related to my previous concerns; I apologise if I have not explained myself clearly.

1. I previously expressed my concerns about paralogy and KEGG representation. In the rebuttal, the authors focus at length on post-LECA duplications. However, my concern are related to stem (pre-LECA) duplications.

2. As previously indicated, the main result ("dominant contribution of Asgard archaea to eukaryotic gene content") significantly deviates from other studies that revealed limited Asgard contributions to eukaryotic gene content, which the authors were asked to investigate/explain. The authors point out limitations (limited Asgard taxon sampling amongst others) of a previous study that could explain the obtained differences. However, a more recent study, described in Bernabeu et al, use a comparable Asgard diversity and obtain far less Asgard-dominant contribution to eukaryotic gene content, indicating that taxon sampling alone cannot explain the difference between both studies. Could the authors comment on this?

3. I previously pointed out two examples of proteins where the authors claimed Asgard ancestry, but for which previous studies have suggested Alphaproteobacterial ancestry (NFS1 and ISCU). Given that these examples featured in a main figure, I found it relevant to follow up on this. By providing a "quick" phylogenetic analyses I merely wanted to clarify my point, and urge the authors to have a more careful look at this. Rather than 'shooting the messenger' I was hoping the authors could simply provide a more sophisticated phylogenetic analyses of these protein families to alleviate my concerns. It would be great if the authors consider doing this.

(Remarks on code availability)

Referee #3

(Remarks to the Author)

I would like to thank the authors for carefully considering raised methodological criticisms and in making constant efforts to alleviate them along the different rounds of revision, which led among others to include the "annealed clustering analysis" and further analyses that tested whether more stringent definitions of the "pangenome" would affect the results. I am now overall satisfied with the ways the methods are described and used, and the way the results are reported, interpreted and discussed, in the frame of a highly debated question in evolutionary biology.

I would just have one point left about the newly introduced "annealed clustering" analysis and perhaps how its results enlighten the difficulty to automate such complex evolutionary analysis: I believe this should be part of the Discussion.

Based on the new analyses including that of « Annealed cluster of EPOCs » that is supposed to better account for the overclustering of families (Sup Fig 8), it seems that the proportion of EPOCs of Asgard origins tend to decrease, while the relative order of the major bacterial contributors' changes. This is reflected by the low level of correlations between the contributions assessed via the initial and the annealed clustering of EPOCs (Sup Fig 8C) that seem consistent only for Asgard. While a dominant contribution of Asgard does still hold here as pointed by the authors, this demonstrates that when refining the methodology to better account for the complexity of the data, the results change. This point should be raised in Discussion to emphasize the potential limitations of the current methodology (probably in its last section). Hence, I don't totally agree with the conclusion of the authors in the Results (line 167) that there is a «robustness of the results to both cluster specification and taxonomy », while I agree that there is more consistency for the « core data » subset that are subsequently analysed.

Minor points:

- Line 44: Maybe start already including corresponding references?

- Sup Fig 1 => I know there is not an exact match between the NCBI and the GTDB taxonomy, but they mostly overlap...

Would have been great to see similar/overlapping groups being coloured the same way between panel E and the others (as

the authors did for the Asgard).

- Sup fig 8A => y axis, an « e » is missing in « pairwise distance »
- Sup Figures should be renumbered by order of appearance in main text.
- Line 324: a space is missing
- Several references are missing from the Methods, in particular for the new section about “Cluster Annealing”. Could the authors please ensure to rightly cite all used software and methods? For instance, on line 599 => should “sklearn” be entirely spelled “scikit-learn” (and cited?). And HDBSCAN also cited? Etc...
- Line 733: please check the sentence are it may include several typos.

(Remarks on code availability)

Version 4:

Reviewer comments:

Referee #1

(Remarks to the Author)

-

(Remarks on code availability)

Referee #3

(Remarks to the Author)

(Remarks on code availability)

Referee #1 (Remarks to the Author):

The origin of eukaryotes represents an important knowledge gap in the evolution of life on Earth. The eukaryotic cell evolved via an endosymbiosis between an archaeal host cell and a bacterial (alphaproteobacterial) endosymbiont. Recent work has revealed that the archaeal host was closest related to the Asgard archaea, and that genomes of these archaea encode many genes that are closest related to eukaryotes. Yet, an overall picture of the identity and genetic contribution of the Asgard archaea in the process of eukaryogenesis is still lacking. The manuscript by Tobiasson et al aims to provide new insights into this hiatus, by tracing the evolutionary origin of eukaryotic genes within a 'rigorous statistical framework'. Their analysis reveals a dominant genetic contribution of Asgard archaeal origin, and relatively limited contributions originating from the mitochondrial endosymbiosis. If this would be convincingly shown, this would be an important finding that bears relevance for models of eukaryogenesis.

Several previous studies have focused on the very same subject (e.g. Pittis and Gabaldon, *Nature*, 2016; Vosseberg et al, *NEE*, 2020; Bernabeu et al, *BioRxiv*, 2024). Generally, the Asgard archaeal contribution to eukaryotic gene content has been estimated much lower in these studies. Hence, the claims made in the paper by Tobiasson stand or fall with the rigour of the phylogenetic methodology that has been used to generate the results on which these claims are based. This review will therefore specifically focus on assessing the appropriateness of the methodology that was used. As detailed below, several technical issues have been identified that could potentially account for the inflated Asgard archaeal contributions to eukaryotic gene content. These issues need to be carefully addressed. Furthermore, as a general comment, the clarity and use of language of the manuscript could be improved. Some of the observed issues (incorrect/missing references, missing figure legend, figure not mentioned in main text, etc) indicate that the manuscript was written in a hurry. In addition, and importantly, the methodology section is not sufficiently detailed to reproduce the analyses. The provided Zenodo link does not work, hence there is no access to supplementary datasets.

Major comments

1. EPOCs/clustering/oversplitting.

A main claim by Tobiasson et al involved the 'dominant contribution' of Asgard archaea to eukaryotic gene content. The quantification of this 'dominant contribution' is rather confusing, as different numbers are provided (e.g. see lines 157-171, and compare with Fig 1). In any case, the alleged Asgard contribution is much higher than was observed in other studies (Pittis and Gabaldon, *Nature*, 2016; Vosseberg et al, *NEE*, 2020; Bernabeu et al, *BioRxiv*, 2024), which raises the question where these higher numbers of Asgard contributions stem from. Could these higher numbers be the result from how eukaryotic paralogs are handled? The applied clustering approach likely results in protoeukaryotic paralogs forming individual eukaryotic clusters. Yet, the homologous prokaryotic clusters are probably combined with each (paralogous) eukaryotic cluster, resulting in multiple EPOCs for these duplicated families. Could it be that, given that eukaryotic genes of archaeal origin have been shown to have predominantly duplicated (Vosseberg et al., *NEE*, 2021), this explains why the Asgard contribution is inflated compared with previous studies (including a recent preprint, Bernabeu

et al, BioRxiv, 2024)? Given massive duplications in several eukaryotic protein families (e.g. small GTPases, actin homologs, ubiquitin(-related) families, etc), effects could be significant. Another issue involves that many EPOCs are seemingly not taken into account for quantifying the origins of prokaryotic contributions since they fall outside the chosen parameters/definitions (e.g. soft LCA cut-off; see line 638-645). By omitting these EPOCs, contributions of specific taxonomic classes are artificially inflated.

We appreciate this detailed and constructive comment. Given the concerns regarding the term “dominant”, we have defined and motivated our reference to Asgard archaea as the “consistent and major contributor” in the text and clarified throughout. We further define our usage of “association” as equivalent to average ELW value, further disambiguating our language. As the term “dominant” is only sparsely used throughout the text, outside the title, we hope this is sufficient to alleviate concern. We have further corrected the inconsistently reported number of EPOCs and thank the reviewer for catching this error.

Regarding the work of Vosseberg et al, NEE, 2021, our results are fully compatible with and complementary to theirs. We do detect a significant amount of Asgard derived paralogs in our EPOCs, but this is not enough to make up for the prominence of weaker but more prevalent prokaryotic paralogs among a wider range of genes. In accordance with this observation, if we retain only a single randomly selected EPOC for each KOG (KEGG Orthologous Group), limiting ourselves to a single possible paralog per KOG (in turn having limited coverage of paralogy), we obtain only a marginally lower ELW value for Asgard (0.599 rather than 0.612) demonstrating that the Asgard contribution is not significantly inflated. We have properly cited and mentioned the findings of Vosseberg in several places in the main text (line 60, 153, 442), noted the concordance of our findings with theirs, and illustrated the (minor) effects on our data in main figure 1. We also provide additional data in supplementary figure 3A to indicate the ELW breakdown of paralogs present to further show our agreement.

With regard to the parameterisation of the core dataset we consider our choices to be minimally restrictive yet strictly necessary for the results to be credible. To recapitulate explicitly, for inclusion in the core set, we require each analysed clade to include at least 3 prokaryotic sequences and a taxonomic label purity of 0.8 (i.e 3/3, 4/4 or 4/5 etc. with the same label) to be included. We consider the inference of eukaryotic ancestry based on fewer than 3 of potentially thousands of competing prokaryotic sequences to be considered unreliable and unjustified. We do note the restriction this imposes on so-far poorly sampled prokaryotic taxa and have clarified this point in the Discussion. In addition, we require the ELW of the most likely sister clade to be greater than 0.4. This is necessary as lower values would indicate multiple equally likely ancestries, indicating a lack of resolving power in the data which is a strong cause for exclusion. While we do not consider such data reliable, current manuscript *does* present the full dataset in the absence of any ELW filtering in figure 1B as “All Data, n=14353” for purely demonstrative purposes, and provides all raw and parsed data as supplementary material for the attentive reader to investigate. Finally, regarding the EPOC soft-LCAs, see the discussion below. In brief, while the soft LCA is used to rank the importance of nodes internally, its value is not

actually used to determine the clades selected, an ambiguity in the text which has been corrected (line 654).

2. Phylogenetic analyses.

A. The phylogenetic analyses that were used to identify the prokaryotic source of eukaryotic gene content represents a key aspect of the analyses by Tobiasson et al. In several cases it is unclear how the applied definitions and decisions have affected the results and conclusions in the manuscript. For example, to assign taxonomic lades, a 'soft LCA' score is calculated (defined at lines 635-636), which also includes a division by the 'total number of label X' (line 636). Does this mean that clades that potentially include the sister group in the ML tree are discarded if their main taxonomic label is present in multiple clades in the tree? For example, if the sister clade has 1 gammaproteo and 2 alphaproteos, with a single gamma and 10 alphas in the full tree, the soft LCA score of this clade would be: $\alpha = (2/3) * (2/10) = 0.133$; $\gamma = (1/3) * (1/1) = 0.333$; both are smaller than 0.8 so this clade is discarded. In that case, the results could be simply due to Asgard archaea being much more likely to form a monophyletic group compared with alphas and other bacterial groups. The authors should clarify this in more detail, and show how their results are affected by different cut-offs.

The observation of the reviewer is correct, we internally rank LCAs in order of purify the results and the "scope" is defined as the number of leaves with the given taxonomy within the clade to analyse, divided by the total number of such labels on the tree. The soft LCA value is never used as a filter (for those very reasons the reviewer highlights) but rather as an internal metric. We rely exclusively on the measures of size (more than 3 in a clade) and purity (> 0.8) for the 12 "nearest" (see below) clades to Eukaryotes. The logic and formula for soft LCA are included in the Methods for the sake of completeness and documentation as it is present in the metadata on Zenodo and are not standard in the literature. We have revised the Methods section to more clearly define how clades are selected and clarify the role of the soft LCA values.

B. Furthermore, the presented results are likely affected by the choices that the authors have made with respect to taxonomic labels. The taxonomic levels that were chosen by the authors differ for different taxonomic groups. For example, while mostly (super)phylum-level taxonomic groups are used for archaea (Asgard, TACK, DPANN, Euryarchaeota), the choice is rather heterogeneous for bacterial groups (e.g. Alpha/Gammaproteobacteria at class level, but other groups such as FCB and PVC at superphylum level). While the NCBI taxonomy was probably used for practical reasons to define taxonomic classes, it is problematic since the underlying classification is based on a multitude of many different (often conflicting) studies. To make sure that the applied taxonomic labeling does not result in misleading results, the authors should benchmark their observations against a standardized microbial taxonomy such as the GTDB. Surely the taxonomic classification of GTDB is also not perfect, but for sure less arbitrary than the current NCBI taxonomy.

To assess the robustness of our results to the choice of taxonomy, we de-novo reconstructed the full set of 14000 EPOCs using the GTDB at the phylum level, with taxonomic classification mapped to our data

through either direct comparison of accession numbers or through the marker gene set provided by GTDB 220. We provided the data in the supplementary material and explicit aELW calculations in the supplementary figures. The dominance of the Asgard signal remains, indicating the findings to be mostly independent of the choice of the taxonomy. Obviously, the distribution of the association of the other prokaryotic taxa differs due to the significant taxonomy changes for example the merger of all Betaproteobacteria into Gammaproteobacteria, increasing their overall aELW. We present the comparison in the supplementary material, note the differences in the main text (line 176) and provide all final data on Zenodo.

C. Another issue involves the identification of the closest prokaryotic clade of eukaryotic proteins in phylogenetic trees. In order to infer the 12 closest clades (line 661), the tree has to be rooted. It is not described in the manuscript how this is done. Extended Data Figure 2 seems to imply that the tree is rooted on the eukaryotic clade. If this is the case, it is not possible to infer closeness based on topological distance.

While no trees in the current study are rooted in the phylogenetic sense, see comments from R3, we do describe the computational rooting procedure in the text, under “EPOC tree construction and processing” (line 635) as such; *“The resulting trees were then re-rooted using a weighted midpoint approach so that the sums of all branch stem lengths on either side of the root were equal”* (line 637). We would, however, respectfully contest that a tree must be rooted in order to assess topological distance. Based on the data structure of a tree, a topological distance can be defined between any two points as the number of internal nodes separating the two. This is independent of both a phylogenetic “root” (which we do not have) and the computational “root” of the data structure, as long as the computational “root” node is not counted. This is our current definition of topological distance which we have clarified in the Methods (line 677).

The reviewer further notes that Supplementary Figure 2 was potentially misleading as trees were drawn as rooted, which we agree with. We have modified the figure to make it clear that the tree is phylogenetically unrooted and clarified this in the legend and in the Methods. We further removed the element indicating a tree root from main Figure 1.

D. Finally, the authors surprisingly report that eukaryotic proteins of Asgard archaeal origin display shorter stem lengths compared to eukaryotic proteins of e.g. Alphaproteobacterial descent (Figure 5B). This result not only contrasts with the results reported in previous studies (including the aforementioned preprint from Bernabeu et al.), but it is also contradicting with what one typically observes in single gene trees and trees based on supermatrices: a long eukaryotic stem for archaeal-derived genes and shorter ones for alphaproteobacteria-derived genes. This is also a main reason why mitochondrial genes have been used to root the eukaryotic tree of life in recent studies. It is unclear what causes this aberrant result, but the biological explanation provided in lines 367 and onwards are rather speculative/unconvincing. The authors should exclude that a technical reason is underlying this contrasting result.

The surprise of the reviewer is well founded as our results as presented in the original manuscript do not support the previous findings of Pittis and Gabaldon 2016. The reviewer further posits stem lengths are commonly and typically observed as longer in archaeal-derived genes in comparison to alphaproteobacteria-derived. We respectfully question the generality and robustness of this claim as we are unaware of much consistent and comparable data to this effect outside of the 2016 Gabaldon study. Regarding the paper of Bernabeu et al., we cannot find in it measurements of any new alphaproteobacterial or Asgard stems as the paper aims to validate the stem length methodology by working with mammalian sequences. Their PLOS Biology paper does refer to general literature in regards to Archaeal stems compared to prokaryotes but note that these are reported as being typically short:

"In the latest ToL reconstructions [9,10] (Fig 1A), the branch separating eukaryotes from their closest archaeal relatives (where the transfers in [6] occurred) was found to be relatively short; in addition, the inferred bacterial donors in [6] branch deep in the bacterial phylogeny [9–11]."

(<https://doi.org/10.1371/journal.pbio.3002460>)

further indicating that the reviewer's statement on "a long eukaryotic stem for archaeal-derived genes and shorter ones for alphaproteobacteria-derived genes" is not a consensus in the field. Although our methodologies are not strictly comparable, such observations of short archaeal stems as cited by Bernabeu is in concordance with our findings as well.

With regard to our current data, we emphasize that we analysed a much wider sampling of both Asgard sequences and Eukaryotes compared to the 2016 paper. Therefore, we are not only able to sample a much larger set of protein functions among the proteins of Alphaproteobacterial origin compared to the previous study, but also provide both a deeper estimate of LECA and a more diverse FECA, narrowing the FECA to LECA gap from both sides for Archaea and broadening the functional view of Eukaryogenesis.

The reviewer further suggests ensuring that our stem length measurements are not aberrant due to "a technical reason". This comment is hard to fully address given its generality. As far as we can judge, our tree construction approach (maximum likelihood trees built using state of the art methods, with full model evaluation and selection) are as rigorous as the current methodology allows for single gene trees, especially given that we recover numerous known phylogenetic affinities across a significant geological time and considering a wide range of protein sequences. Nevertheless, trying to address this comment, we investigate whether sampling artifacts or reproducibility in stem length calculations could have any substantial effect on the results. To this end, we have carried out two checks. 1) We ask whether there are any non-deterministic steps in our EPOC reconstructions which would consistently bias the reliability and reproducibility of stem lengths. To address this, we calculated 20 technical replicates for each of 175 Alphaproteobacterial EPOCs, randomly sampled from those presented in Figure 5B. Subsampling was done to expedite calculation and such subsampling had no meaningful impact on the outcome as evident below. 2) We further augmented the single Mann-Whitney U test for statistical significance with

a battery of tests, each derived from 200 bootstrap replicates drawn from stem lengths of Alphaproteobacteria, Cyanobacteria and Asgards respectively. We included the results of these additional tests in the revised manuscript Figure 5 and supplementary figure 7. These results show that the stem length difference between Alphaproteobacteria and Asgards remains as initially presented across both technical replicates and independent on sampling of EPOCs, providing further support for our claims.

However, as a result of the expanded statistical testing, we now note that whereas Asgard stems shorter than 0.3 (~70% of all stems) are significantly shorter than those of Alphaproteobacteria ($p < 1e-25$), the trend is reversed for stems longer than 0.35 (~15%), as can be seen in Figure 5, and clarified in the revised manuscript. In essence, whereas the majority of Asgard stems are shorter than the Alphaproteobacterial ones, the longest stems are longer. When trying to correlate our data to the Gabaldon 2016 study, we were unable to verify exactly which proteins were used for stem length calculations. However, their Table 1 notes that their Alphaproteobacterial stems were primarily drawn from proteins involved in “Energy production and conversion” whereas their archaeal stems were drawn from “Translation, ribosomal structure and biogenesis”, If we restrict our calculations to only EPOCs within a representation those categories, here taken as oxphos complexes and the ribosome, we further exacerbate the observed trend. Furthermore, since those stems used by Gabaldon were predominantly longer than 0.3, as evident from their Figure 2, this could explain their observations as they primarily work with a subset of proteins with longer stems. Coupling this trend with our deeper representation of LECA and more diverse FECA, we believe we can adequately explain the difference between the two studies while demonstrating the robustness of our estimates.

This more nuanced view further accentuates our conclusion that normalised stem lengths are unlikely to correlate well with the geological time over long timescales and a wide range of sequences and functions, as is our primary emphasis in the revised manuscript. We appreciate the reviewer’s comments which made us pay closer attention to these relevant issues, hopefully providing some perspective on the results of Pittis and Gabaldon, and include elements of the above discussion in the main text as well as supplementary figure 7 and revised figure 5.

3. Figures 2B (not referred to in the main text...) and 3 provide examples of eukaryotic pathways that display dominant Asgard archaeal contributions. The authors should provide phylogenetic trees as supplementary data that convincingly support these claims.

We apologize for not referring to Fig. 2B in the submitted manuscript. In the revision, this image has been removed as per discussion below. A file with annotated newick tree files have been provided in the supplementary.

4. The provided Zenodo link does not work, hence it was not possible to access and inspect the data sets underlying the figures and claims.

As far as we could verify from several independent IP addresses, the provided link is and has been fully functional and both Zenodo and github repositories are accessible. The Zenodo repository has also been indexed by search engines and is easily seen via any up to date web browsers. Its initial deposition date shows it to have been available at the time of the initial submission and, at the time of latest revision (050325), the archive has been downloaded over 200 times (<https://zenodo.org/records/14013997>). We apologize for any inconvenience in accessing supplementary data, but to the best of our knowledge, the data is and has been accessible at the provided link since initial submission. The latest version of the supplementary, extended with data included with the revisions, is available on <https://zenodo.org/records/15048010>

Minor comments

1. Line 41-45. The contribution of de novo gene birth and the numerous gene duplications that occurred along the eukaryotic stem to the evolution of “key features of eukaryotic cell organization” is ignored in this line and the rest of the manuscript. Moreover, what does “the Asgard ancestor” refer to: the last common ancestor of all Asgard archaea, the LCA of eukaryotes and the closest Asgard group, or the host in endosymbiosis?

The contribution of the numerous duplications (incidentally, first documented, to our knowledge, in an early paper coauthored by two of the current authors Ref. 49 in the present manuscript) was certainly crucial in the early evolution of eukaryotes, and de novo gene origin, although less readily demonstrable, was in all likelihood important as well. However, these issues are not central to the current study and therefore we believe do not belong in the Abstract. In the revision, we do mention this in the Introduction. In the revised Abstract, instead of referring to “the Asgard ancestor” which indeed can be perceived as ambiguous, we write “the Asgard lineage leading to the last common ancestor of eukaryotes” which is regrettably longer but more precise. Further, in the main text, we refer to this lineage as the First Eukaryotic Common Ancestor (FECA). We appreciate the reviewer bringing our attention to these issues.

2. Line 68. Please note that the taxonomic (re-)classification of Asgard archaea is currently pending, as another proposal for the nomenclature has been published that aimed to retain the ‘Asgard’ nomenclature (Tamarit et al, Syst. & Applied Microbiology, 2024). I suggest you cite both these papers, or neither of the two.

We appreciate this comment. This is indeed relevant work and both papers are properly cited in the revision.

3. Line 90-91. Perhaps such models would not require membrane replacement, but besides a second endosymbiosis, it would need complete retargeting of membrane proteome, so I would involve additional challenges compared to ‘single-endosymbiosis’ scenarios. I suggest mentioning this as well.

This is not quite clear to us. If the original host was a bacterium, seemingly, no complete membrane proteome retargeting would be required inasmuch as the bulk of that proteome was of bacterial origin. Only surviving archaeal membrane proteins would need retargeting to the bacterial membrane. We mention this briefly in the revised manuscript, but given the complexity of the issue, we thought it was best not to discuss it in detail.

4. Line 113, incorrect reference (2932)?

This error (2932 instead of 29,32) has been removed in the revised version as reference numbering changed to accommodate new references.

5. Line 118. At this point it is unclear where these “26 curated prokaryotic taxonomic classes” come from. Please provide more context.

The selection of these taxa is described in the Methods, in the section “Curation of taxonomic labels”. In short, the process is described as:

“As the rank “Class” does not evenly partition biological diversity, identifiers were further manually mapped to a curated set of 45 eukaryotic and 26 prokaryotic taxonomic superclasses, each present in the NCBI taxonomy (for example, Metazoa, Streptophyta, TACK archaea, see supplementary Data 1)”.

For the readers interested in details, the full mapping is described in the supplementary files and available on Zenodo.

7. Line 126. Noun (probably “sequences”) missing between “prokaryotic” and “were”.

“Proteins” was missing, inserted.

8. Line 300. EPOCs that are associated with the plastid ancestor are mentioned: how have these post-LECA clusters passed the LECA criteria set by the authors?

This is due primarily but not exclusively to secondary endosymbioses that are known to have been common in the evolution of eukaryotes as discussed in more detail in our responses to the comments of reviewer 3 below and pointed out in the revised manuscript.

9. Line 438. “likely”: based on which findings of their own or other studies do the authors think this is likely?

This is based on the generally recognized high rates of HGT among prokaryotes. Revised in the updated version of the manuscript.

10. Line 464. Figure 6 is lacking a legend.

We initially considered this figure to be self-explanatory but in the revision, a legend was added, also in response to the other reviewer's comment below.

11. Line 469. Several other studies have provided complete Asgard archaeal genomes (Valentin-Alvarado et al, Genome Research, 2024; Wu et al Nature Microbiol, 2022).

These publications are cited in the revision.

12. Line 498. "All clusters which did not contain sequences from at least 67% of all bacterial families within a prokaryotic class were rejected from the database." Not all genomes are assigned to a family in the NCBI taxonomy: how did the authors take this into account?

We appreciate the reviewer bringing our attention to this point. This line in the Methods was erroneously left unchanged from an early version of the manuscript and has been changed to more accurately reflect the current reconstruction protocol for the pangenomes. The pangenome definition relies on "species" level assignment defined as the lowest ranked unique NCBI taxonomy id per assigned class, not family. Further the cutoff was incorrectly specified as 67% while the final data uses a 50% cutoff. We have corrected this statement to bring it in line with the methodology used.

13. Line 692-693. "Cases of ELW < 0.99", "<" should be ">=".

Fixed.

14. Figure 1B-C and similar figures: colours are recycled due to the high number of categories. According to the description, average ELW values are displayed but the figures show the number of EPOCs.

The colors have been changed to address this issue.

15. Figure 2. Not referred to in the main text. Colours not explained for panel 2A.

We apologize for failing to refer to this figure in the manuscript. This figure has subsequently been altered following other comments and is referred to in the revision.

16. Figure 4. Not clear what the two different bars mean for each taxonomic group. If one of them is relative to the Asgard contribution, why is the Asgard bar then not at 1?

The figure legend has been revised in order to clarify the representation of the elements.

17. Figure 6. The proposed scenario is not new as similar models for eukaryogenesis have been proposed before, and it is unclear how the analyses/results described in the present paper have contributed to the proposed scenario.

Indeed, the scenario depicted in this figure is not really new. We have clarified this in the added legend and discussion to avoid any confusion. Nevertheless, we find that including this figure was highly desirable to indicate which scenario of eukaryogenesis our results were best compatible with. With regard to the contribution of our results, we believe they significantly contribute to the choice between the evolutionary scenarios. These contributions are also reflected in the specifics of the figure whereby the dominant contribution of Asgard archaea is reflected in the large area depicting the vertical gene flow in contrast to the thinner arrows denoting HGT from bacteria which we propose to have been occurring continuously along the entire evolutionary path from Asgard to the LECA. This is emphasized in the revised manuscript and in the legend to Figure 6.

Referee #1 (Remarks on code availability):

I have not inspected the code, but even if I could, the provided Zenodo link did not work, hence it would not have been possible to reproduce the analyses.

See above comment regarding data and code availability.

Referee #2 (Remarks to the Author):

Tobiasson et al.

Dominant contribution of Asgard archaea to eukaryogenesis

This paper is a very fresh take on the role of Asgard archaea in eukaryogenesis. I really like the approach taken in this paper. The authors take advantage of huge sequence databases available for bacteria and archaea and augment it with the rapidly expanding datasets for asgards (MAGs but also several now confirmed by having sequences from two or three (I believe) asgards in culture, thus ruling out concerns over MAG binning, assembly, etc.).

The take home message is that Asgards appear to have made a much larger and diverse contribution to the genes / proteins / processes present in LECA than originally believed. There may well be quibbles over some of the bioinformatic decisions made (e.g., the relatively small number of eukaryotes required to constitute an EPOC (page 21)), but on balance I don't have any problems with the methodologies. I think the right balance has been struck for the desired high-level questions being asked of the data.

I particularly like the stem-length (re)analysis, which is a way of potentially ordering temporally prokaryotic contributions to eukaryotic evolution the authors conclude that in fact they can't do that reliably with their data. Instead they invoke bursts of sequence divergence associated with varying difficulties associated with adaptation to different cellular environments (in the case of bacterial donors to eukaryotes). The idea of varying rates of adaptive evolution has been around for a long time -- I'm thinking originally when looking at things like actins and tubulins, which are slowly evolving within eukaryotes but when compared to their very distant prokaryotic homologs, appear to have diverged before the origin of life on Earth. I have not come across this invocation recently and it makes a lot of sense in the present context.

One potential oversight is not acknowledging the genome mosaicism that clearly exists in prokaryotic genomes today (and euks to a lesser extent). This is relevant because we can only assume that genomes of all lifeforms were mosaic at the time that euks were first evolving, and this complicates the process of assessing donor genes / pathways / prokaryotic lineages. So I think the authors should at least acknowledge this and cite the paper(s) from Bill Martin's team, who impressed upon me (at least) that this was an issue to take seriously.

We thank the reviewer for these positive comments. In the revised Discussion, we emphasize the mosaic nature of prokaryotic genomes, citing Martin's publications. This is indeed highly relevant for the study of eukaryogenesis.

Specific questions and comments:

1. Regarding color coding for parts B and C, some colors are used multiple times. For example, orange is both Cyanobacteria and Chloroflexa. I'm assuming this is because the number of colors is finite, and they loop around the same set. So top to bottom on the right column of prokaryotic taxa is most to least on the bar graphs in terms of average total ELW (B) or Total EPOCs). This is probably worth mentioning in the legend, because at first glance it is possible for someone to miss it, for example, choosing Chloroflexa (orange) as the second highest average total ELW instead of Cyanobacteria.

And what do you do if the trend is reversed for one specific histogram (e.g, Orange = Chloroflexa is higher than Orange = Cyanobacteria) compared to the rest? Would the reader know it? Or does the color coding hierarchy apply consistently to all data presented in B and C? The color orange caught my eye because Cyanobacteria is (I assume) the second highest after Asgards, despite the fact that this lineage should only be a factor for the phototrophic groups of eukaryotes.

The coloring has been updated in all figures, ensuring that all taxa are consistently denoted with unique colors.

2. Finally, what about the lightest gray bars on the extreme right of most of the histograms? This is the lightest of the three shades of gray used, and it doesn't have a code in the 'Prokaryotic taxa' color scheme on the right, despite the fact that as a total it is much larger than most of the other prokaryotic contributions. I assume it is 'unassigned' or something like that.

The legend has been clarified to indicate that the grey portions of average ELW plots are for prokaryotic taxa with less than 1% of ELW, hidden in order to reduce the visual weight and increase the interpretability of the figure.

3. Fig 4. I recommend the authors remove the gray vertical and horizontal hatch lines in the background, which make it difficult to read the vertically oriented functional classifications for all bars except the first one for Asgards.

The figure has been updated as suggested for clarity, we thank the reviewer for this suggestion.

4. Line 230: "Although strong Asgard associations were observed for large parts of the overall metabolic network, we observed a high degree of mosaicism in the pathways for fatty acid synthesis and decay." Do the authors mean mosaicism between the pathways for fatty acid synthesis / decay, and/or mosaicism in the signals of the proteins within individual pathways? This got me thinking about the extent is the ELW/EPOC methodology able to detect evolutionary mosaicism within a pathway, e.g., gene/protein X for first step in a biochemical pathway is clearly Asgard origin for eukaryotes, but gene/protein Y for second step is of bacterial origin? I originally assumed that it can't do that, because for example Carbohydrate metabolism (n=423) considers all enzymes together but does not break them down further.

But then on line 243, the authors say: “Here we found the mevalonate pathway, from Acetyl-CoA to mevalonate and further to Farnesyl and Geranyl diphosphate, to be strongly Asgard-associated (Extended Data Figure 3), with the key enzymes hydroxymethylglutaryl-CoA synthase (HMGCS), mevalonate kinase (MVK), phosphomevalonate kinase (PMVK), mevalonate diphosphate decarboxylase (MVD) being clearly Asgard-derived.” So what does “strongly associated” here mean? It looks like all enzymes in this particular pathway, but how does the reader infer this from Extended Data Fig 3 (which has n=41 for isoprenoid biosynthesis).

All this is to say that the authors should clarify how the issue of mosaicism is inferred and presented (or not presented), and mosaicism at what level. Figure 2 shows “split backgrounds”, but Fig. 2 is not referred to in the text, nor is it clear (to me) how the mosaicism identified and elucidated. The issue of mosaicism comes up again in the context of Figure 3 and associated discussion as well.

We have clarified the usage of “mosaic” in the text (line 229) and updated the description and legends of Figure 2, to better indicate the criteria for coloring and inclusion. The resolution of our analysis is variable depending on the pathway, and although we generally prefer to interpret pathway level associations, for certain pathways, the confidence of the ancestor inference is high enough down to the gene level and across functionally related EPOCs, allowing the definition of “mosaicism” with the pathway.

However, while working on the revised version of the manuscript, we realized that the multifaceted arguments underlying which pathways to analyse on the KOG level were convoluted and might not be clear to the reader. Furthermore, we found that the discussion of the mevalonate pathways yielded little further insight when discussed on the KOG level. Thus, we removed figure 2B breaking down the mevalonate pathway, presenting aELW data for the pathway instead and moving a previous supplementary figure to the main (new 2B). However, we chose to retain the Fe-S synthesis discussion as it does provide a clearer view and is better supported in the text.

5. Line 315: “The nucleotide-related associations with Myxococcota were primarily limited to phosphatases and phosphoribosyltransferases acting on nucleotide sugars including 5 and 3’ nucleotidases, and the respective EPOCs showed little to no competing Asgard association. While noteworthy, these associations were limited to a few unique KOGs whereas all other associations with Myxococcota remained scattered across various pathways (Extended Data Figure 5).” Looking at that figure, I can see “competing Asgard association” in all data presented. This needs to be better described so that the reader can understand how the inferences have been made. It’s not obvious from the figure alone (at least part A; part B doesn’t allow one to tell, other than that this list of KOGs has Myxococcotan presence). Also, is “Myxococcotan” a typo?

As correctly observed by the reviewer, the pathways which Myxococcota associate with are still mostly dominated by Asgard. However, for each Myxococcota associated EPOC, the combined ELW values show low Asgard associations. The supplementary figure 5 text has been modified to clarify this and the corresponding part of the main text was revised. All references to “myxococcotan associations” have been corrected to “myxococcota associated”.

6. Line 430. At this point the authors should cite the papers actually showing that extant asgards that have been studied actually have a cytoskeleton, namely Rodrigues-Oliveira et al. Nature 2022 (this paper is cited later for a different reason). As written, it implies that this is a prediction from the results in the paper, when in fact it is also supported by experimental data. The field has come a long way from the first description of Asgards from MAGs, the presence of ESPs, and the debate about what those proteins actually did (or did not do) in the cell, and I would encourage the authors to underscore this.

We appreciate this comment and, in the revision, refer to the experimental studies of cytoskeleton in Asgard archaea.

7. Line 73—Just like above. Yes, cytoskeletal proteins and membrane proteins, but make it clear that these actually DO eukaryote-like things. This was far from clear early on when they were flagged as ESPs when the first Asgard MAGs were published.

Again, we thank the reviewer for raising this important point. The text (line 56) has been revised in response to the reviewer comments to explicitly mention eukaryotic-like functions tied to current experimental evidence, in addition to previous citations.

8. I find it hard to believe that there are more euk. sequences from Dinophyceae (dinoflagellates) than for all of Opisthokonta. Is this because of approach of / composition of Richter et al. 2022 EukProt dataset? Maybe it's massive numbers of RNA-seq data from a relatively small number of dinoflagellates with very large, gene-rich genomes? Dinophyceae are one of the more poorly sampled groups of algae, so this stands out as an outlier.

The reviewer is correct in their interpretation. We have more sequences from Dinophyceae than from Opisthokonta, and this is due to their wide representation in EukProt. The high abundance of sequences from Dinophyceae results from their large genome size but more importantly, from the wide sequence diversity.

**9. Another point for these data is Fungi incertae sedis, Microsporidia, and Choanoflagellata listed separate from the Opisthokonta. The Opisthokonts include animals AND fungi (and microsporidia are related to fungi) AND choanoflagellates. I am not sure of the implications here for downstream analyses but it should be noted and (if need be) addressed.
-Where are the Rhodophyta (red algae)? I see Rhodelphea listed, but not Rhodophyta. There are more Rhodophyta genomes / transcriptomes available than Rhodelphea, so this is an oversight. Maybe they are accidentally included under Chlorophyta?**

We apologize for a plotting error convoluting several taxa including Rhodophyta. This has been addressed – again, we appreciate the reviewer bringing this to our attention. In the revised figure it's clear that there are less than 100,000 poorly assigned “Opisthokonta” sequences.

Nevertheless, the reviewer points to an important issue, namely, that working with a reducible taxonomic representation risks overcounting the number of unique clades present on trees. Here we require more than 5 unique labels to be present, including at least one from Amorphea and one from Diaphoretickes. For the overcounting to affect our inclusion criteria, either “Fungi incertae sedis”, “Microsporidia”, or “Choanoflagellata” or “Opisthokonta” would have to co-occur, a rare occasion given the limited number of sequences from these groups. We verified the extent to which this overcounting occurred in our data and observed 13 eukaryotic EPOCs including one such clade. Given the total of 6048 clades in our core set, we consider this overcounting (0.2%) to be negligible.

10. Line 145. What happens when all eukaryotes are not monophyletic, e.g., a particular euk. lineage has a non-orthologous replacement from a prokaryote? This section underscores the robustness of the EPOC procedure as being tolerant to LBA and non-homogenous clades, as well as paraphyly. Since the EPOC procedure ultimately treats euk. clade as a “single datapoint”, I suspect this is taken into account and tolerated. But I would suggest the authors explicitly call out the robustness of their methodology to lineage-specific LGT / non-orthologous replacement of otherwise broadly distributed genes in the “soft-core” pan-genome. Perhaps this is all taken into account in the section on page 686 (Data filtering and removal of likely HGT cases).

This situation is indeed taken care of by our criteria that require both deep divergence and a minimal representation of 5 eukaryotic sequences. If non-orthologous displacement occurred in a single eukaryotic lineage, these criteria would not be met. It might potentially happen that distant homologs from the same eukaryotic group replaced the eukaryotic gene independently in multiple lineages. However, we are unaware of such cases and consider this highly unlikely.

Referee #3 (Remarks to the Author):

In this article, Tobiasson and colleagues attempt to evaluate the relative contributions of several lineages in eukaryogenesis. To this end, they first collect from public databases a dataset of nearly a thousand eukaryotic genomes, and about 47 000 complete genomes of 7900 prokaryotic species enriched with nearly 150 metagenome assembled genomes of Asgard archaea. They then compute the LECA (last eucaryotic common ancestor) proteome and use phylogenetic analyses to find the prokaryotic clade of origins of this ancestral eucaryotic proteome. They find a largely dominating contribution of Asgard archaea to the proteome of LECA – as expected were this lineage to be the host of eucaryogenesis, and second, of Alphaproteobacteria, which are thought to be close to the ancestor of the mitochondrion. The authors look at the relative contribution of different prokaryotic lineages at the taxonomic and then functional level and mostly find no evidence for a consistent contribution to coherent functions from other lineages than Asgard and Alphaproteobacteria. They then re-employ a tree metric called ‘stem length’ used in a previous study by Pittis and Gabaldón (Nature 2016), which analysis had then led to the proposal of a late mitochondrion acquisition along eukaryogenesis and to the proposal that several other bacterial lineages contributed genes to ancestral eucaryotes, not only Alphaproteobacteria. The revisited analysis by Tobiasson and colleagues, in addition to the ones mentioned above lead the authors to contradict the conclusions of the 2016 article, as Tobiasson and colleagues do not seem to find consistent results. Instead, they say their results rather point at a dominant Asgard contribution first, and secondly, a Alphaproteobacterial contribution to the ancestor of eucaryotes, whenever other bacterial lineages had only a little, non-significant contribution. The article is well-written on a hotly debated topic that would be of interest to the broad audience of Nature, but I have several major concerns regarding the methodology and dataset used, which I think could affect the main conclusions of this study. Further, I sometimes did not agree with the conclusions of the authors drawn from their results, as I found them unjustified. Please find below my major comments.

Major comments:

1. The dataset

A. The authors employ complete genomes of prokaryotes from the NCBI database, and only include MAGs (metagenome assembled genomes) for Asgard archaea, which I found not justified. Indeed, this makes the known Asgard diversity much more well covered, while this also artificially inflates the number of Argard genomes considered comparatively to that of other lineages. More and more organisms are covered solely by sequence evidence (this is the case for Asgard, but also numerous bacterial lineages) and their genomes are never closed. Hence, most of the recently discovered diversity of bacteria and archaea is not covered by complete genomes, and by the dataset being used in this study. By solely focusing on complete genomes, the dataset is highly biased towards certain groups (often those being cultured), while ignoring others (this is visible on extended Figure 1). This would directly affect the taxonomic identification of prokaryotic donors to ancestral eucaryotes as some lineages are not covered at all, or barely covered. Solving what happened during eukaryogenesis is a very difficult endeavor, and I think that to this end, the most should be done of the data available.

What was done here does seem not to be it, given the high amount of available genomic information that was not included. The authors acknowledge themselves in the conclusions on line 466-467 that a limited sampling is an issue, but I believe this is a serious shortcoming here.

B. Regarding the eucaryotic dataset, could the authors provide some information on how well they think the chosen dataset covers the known eucaryotic diversity and the lineages with sequence data available?

We appreciate these constructive comments and agree with the reviewer that utilising available data to the fullest possible extent is necessary. However, the statement of artificially inflating the Asgard presence is difficult to verify as we cannot approximate the true taxonomic diversity within the Asgard, or any clade for that matter. Thus, although it does not seem possible to provide a direct answer to this comment, we would like to elaborate on the reasoning behind our decisions and the cost/benefit trade-offs that are incumbent to working with Asgard data and prokaryotic MAGs.

Generally, including MAGs in an analysis of gene origins is quite undesirable if closed, well annotated alternatives exist, as any result from such data is bound to be incomplete and open to questioning. When it comes to Asgard, however, there was simply no choice as the number of complete genomes is so small. We would likewise respectfully contest the reviewer's general notion that in our data the true Asgard diversity is better sampled than that of other prokaryotes through inclusion of metagenomes. For the major bacterial groups, notably, Alphaproteobacteria, we include many more complete and diverse genomes than there were high quality MAGs from Asgard. Obviously, some bacterial groups that are particularly poorly among the currently available complete genomes would be under-represented, but any such genome analysis inevitably is a moving target and therefore continuously open to criticism regarding sampling.

On a more practical note, the analysis presented in the current manuscript is already a massive computational effort, the EPOC computation alone requiring around ~250,000 single core cpu hours an HPC cluster per reconstruction, not including database formatting and searches. Adding complete metagenomic information on the scale of all known prokaryotes would make searches and alignments unrealistic to carry out at the quality level we provide here. Nevertheless, in order to illustrate this issue, we compared the sequence diversity for Myxococcota, one bacterial phylum that came out prominently in our analysis, with Asgard, using either a collection of complete genomes or data which included a broad MAG sampling (detailed below).

For *Asgard*, the closed genomes represented at NCBI (technically, assembly levels "Complete" and "Chromosome", the latter allowing unsequenced gaps of known length that do not impinge on the assembly continuity) and accordingly included in our prok2311 database encompass 36,101 protein sequences encoded in 12 genome assemblies. If these sequences were to be subjected to our clustering pipeline, we would base our inference on 2021 non-redundant protein clusters. The Eme 2023 dataset which is used in our paper extends prok2311 to "prok2311_as", adding ~550,000 sequences from 64 proposed species, increasing the number of non-redundant protein clusters to 10,349, a 5-fold increase, and adding more than 8,000 new protein clusters requiring "only" 500,000 more sequences. In contrast,

we compare our prok2311 representation of Myxococota to that present in GTDB (prominently including MAGs). Here we increase the raw protein count from 344,197 sequences from complete genomes, forming 1823 clusters, to 3,200,000 sequences from metagenomes, forming 2,138 clusters, a ~15% increase, only adding about 200 “new” protein clusters. Yet 2,800,000 extra sequences were required for this limited gain. The relative gain compared to the cost, given the computing constraints and allocations, is minor compared to Asgard and hence we chose to limit ourselves to only using MAGs from Asgard.

With regard to the Eukaryotic diversity, we used Eukprot v3.0, which to our knowledge is the widest available representation of extant eukaryotic diversity without overrepresentation of commonly sequenced phyla. Eukprot contains manually curated species data from both fully sequenced and single cell genomes and transcriptomes. We would like to note that the data present within Eukprot does contain a significant amount of non-culturable organisms outside known model species. Reflecting this, of the 993 species included in the dataset, 601 are based on RNA sequencing technology, including transcriptomes and single cell transcriptomes. However as opposed to the prokaryotic side of the analysis, the total sequence space is smaller, allowing these inclusions at a comparatively low cost.

To summarise, we agree that the initial database selection and curation are of high importance. Given the scope of the data and the in-depth methodology we employ, there are limitations to the number of sequences we can deal with. As such, we worked with the highest quality data available, only including MAGs where they were essential. In the revised Discussion, we modified the paragraph on the effects of sampling towards more caution and agnosticism regarding potential under- and over-estimates.

2. The reconstruction of the LECA proteome and of its origins

A. The authors claim they reconstructed the LECA proteome, but I found the method employed quite raw, and I believe this method should be better justified, or adapted to the difficulty of the task. The authors use a census criterion, but do not seem to really use a phylogenetic approach to compute the LECA proteome. Could the authors explain how the proposed approach could clearly identify vertically inherited gene families that trace back to the LECA?

We apologize for any confusion caused, perhaps, by imprecise language in our submitted manuscript. We make no claim of reconstructing the LECA proteome, that was not among the goals of this study. We state that, based on their taxonomic distribution of sequences and assumption of homology, the EPOCs we analyse “represent genes tracing to the LECA”, not that the EPOCs represent the complete gene set of the LECA. Further, given the coverage of KOGs relative to the known extant metabolic capacity of Eukaryotes, we know we cover a sizeable portion of cellular functions, enough to identify trends. The best possible reconstruction of the gene set of the LECA would probably include a larger number of genes while likely excluding some genes from our set (see below). As the reviewer notes, reconstructing such a set would undoubtedly require more advanced methodology, with questionable applicability at such evolutionary distances we are working with. In the revision, we have clarified the methodology in line with this correction and emphasized that reconstruction of the complete gene set of the LECA was not among our aims in this work.

B. About the inference of the procaryotic origins, I find unjustified the choice to limit the analysis to families that are highly distributed within a procaryotic class (Methods, line 493-496 “All clusters which did not contain sequences from at least 67% of all bacterial families within a procaryotic class were rejected from the database.”). Again, this choice made in the families to be considered in the analysis could bias the results, as the fact that a gene is not broadly taxonomically spread does not mean that it could not have been contributed to the LECA proteome. Moreover, the choice being based on the NCBI taxonomy that may have uneven phylogenetic depths makes it even more questionable. Could the authors please comment on this?

In the original manuscript, we failed to sufficiently stress the importance of sequences having at least some limited taxonomic presence across Prokaryotic clades in order to distinguish true vertical inheritance from horizontal transfer. It is well known that many eukaryotic genes have most closely similar homologs in prokaryotes, and vice versa, many prokaryotic genes appear to be “eukaryotic” . This can stem from three main causes: contamination of eukaryotic genomic assemblies, horizontal transfer, and vertical inheritance. In our analysis, distinguishing the first two cases from the third one is crucial as these are irrelevant for eukaryogenesis, and if not accounted for properly, could dramatically distort the results. In particular, local horizontal transfer would appear as a strong prokaryotic association with a sub-clade of eukaryotes, both providing an erroneous signal and masking the true sister clade. We highlighted this issue in the revised manuscript and cited relevant literature, notably Van Etten 2020 (from line 96). If not handled properly, as we experienced firsthand in the earlier stages of the project, the analysis is dominated by inferences dictated by small prokaryotic sister clades (3-5 sequences) which originate from a very narrow range of species. This inevitably leads to an analysis where the majority of the alphaproteobacterial association with eukaryotes is dictated by *Rhizobium*, *Bradyrhizobium* and *Agrobacter*, all symbionts or parasites of plants, and where the eukaryotic transporter proteome is dominated by *Legionella*, *Vibrio* and *Clostridium*, common pathogens. Therefore, we are convinced that any inference of global ancient events which are dominated by data of limited taxonomic presence would be highly error-prone and unreliable.

Consequently, we required that all proteins present in our analysis must have sufficient presence across their prokaryotic Class. As noted in the Methods, we addressed this both through the initial removal of trivial bacterial contaminations from EukProt, and second, through analysis of prokaryotic pangenomes. We observed that mild to moderate pangenome definitions at the level of species effectively removed the vast majority of the suspicious associations, i.e. those related to known post-LECA pathogens or ectosymbionts and those with limited taxonomic presence. We do note that this procedure prevents us from observing any possible true contributions from highly specific taxonomic origins. However, if such cases indeed existed, we would not be able to separate these from latter cases of HTG we know are artificial. We have further emphasized this point in the discussion (line 491).

The pangenomes also naturally depend on the choice of taxonomy. In the current manuscript, we have further reconstructed the full set of EPOCs using the GTDB taxonomic classification and show the

consistency of the Asgard signal, indicating that our main findings are retained while using the GTDB taxonomy as requested by Reviewer 1 and shown in supplementary figure 1E.

In the revision, we expanded the Results section to highlight the critical importance of pangenomes as well as regarding the prevalence and necessity for removal of suspected cases of HGT and provided statistics for alternate pangenome definitions in the supplementary material. We now explicitly show in supplementary figure 1D that simply enforcing a protein to be present in 10% of species within a clade is enough to reveal a strong Asgard signal which does not substantially depend on further, stricter pangenome definitions. Nevertheless, we believe that a 50% pangenome definition is desirable as the distribution of species per class makes any cutoff imperfect, and therefore, we chose to err on the side of caution.

C. I am surprised to see so many genes inferred that could be of Cyanobacterial origins (Fig 1B), while this is not discussed in the text. For instance, porphyrin metabolism is linked to a Cyanobacterial origin (Fig 4). If this corresponds to genes linked to the later secondary endosymbiosis of the ancestor of the plastid, how could the authors explain this was inferred to be present in LECA? This high number of genes coming from Cyanobacteria is not really discussed, as it should be. To me the potential inference of genes of plastid origins in LECA could be a clear sign that the method to infer the LECA proteome is not accurate enough.

The observation of a cyanobacterial signal is due to the broad occurrence of secondary endosymbiosis across Eukaryotes. However, some of it might stem from genuine acquisition of cyanobacterial genes on the path to the LECA as is the case for all diverse prokaryotic ancestry. Removal of such associations (like porphyrin) would either require a functional blacklisting on an ad-hoc case-by-case analysis, providing additional levels of methodology subject to interpretation, or a stricter filtering which in turn yields false negatives and substantially reduces the gene set available for the inference of ancestry. To avoid such cases, we kept the cyanobacterial contributions, noting their prominent role post-LECA and their impact on our analysis. As noted above, an accurate reconstruction of the LECA gene set is a separate task that is outside the scope of the present study.

Incidentally, the detection of the cyanobacterial contribution also serves as a test for sensitivity of our analysis. We should necessarily have detected the cyanobacterial signal if we expect to be able to identify the contributions of other bacterial taxa such as Alphaproteobacteria, and this is indeed the case.

D. I did not understand why the strong signal for a contribution of Actinomycetota was dismissed and judged irrelevant in the text without further explanations (Line 234-235). The authors should more strongly argue on why they do not think it is relevant, otherwise it could be interpreted as cherry-picking. Anyway, the fact that the results from the proposed methodology are questioned by the authors themselves makes me even more unsure of the validity of the proposed approach. Could the authors please explain?

We do not dismiss the contribution of Actinomycetota but downplay its importance for eukaryogenesis because there is no specific, functionally coherent signal from this (and many other) bacterial phylum. As opposed to Asgard or Alphaproteobacteria, which show unquestionable association to several specific and functionally related pathways, the Actinomycetota association is uniformly spread across all pathways, in a sense, being ever present as a background. This trend is in turn true for the vast majority of diverse prokaryotes. This does not at all rule out a continuous, non-specific gene transfer between Actinomycetota and eukaryotes, as we point out in the revised manuscript. However, these observations do suggest that this and other bacterial taxa did not play a defining role in eukaryogenesis individually. We certainly agree that the combined influx of genes from diverse prokaryotes was an important factor of early evolution of eukaryotes, but as a background, non-specific force, rather than a singular, functionally significant event. We thank the reviewer for highlighting this ambiguity in presentation and have clarified and expanded this discussion in the revised section on diverse transfers (line 291 and onwards) to better convey our reasoning. We have also reworded several statements throughout which might have been perceived as disregarding the presence of individually well supported prokaryotic contributions.

E. The case of the mevalonate pathway is also another point that questions some of the obtained results: the authors present it as a clear case of acquisition of a pathway from Asgard, but (1) the pathway seems to be not fully complete in the inference of LECA if I understood the figure correctly (Is it only missing PMD? Or are there other essential enzymes missing?) and (2) there are several enzymes that harbour a mixed signal (Asgard and other bacterial origins). Adding to my confusion (but maybe is this just a mistake or typo?), the main text contradicts Figure 2B regarding the origins of MVK (line 246): on one hand, it is said to be “strongly Asgard-associated” whereas on Fig 2B the enzyme is displayed as having a “mixed” signal for its origins. Could the authors please clarify this case as I’m unsure what to think about it?

While abiding by the analysis presented in original figure 2B, we acknowledge that the reasoning for the KOG-level mapping might not be clear to readers. As elaborated in a comment to reviewer 2, our reasoning behind including KOG level analysis for the mevalonate pathway specifically is based on several factors derived from manual inspection of EPOCs, and thus difficult to concisely convey in the limited space we have. As we found that relatively little of the discussion required understanding of the individual KOGs, we have removed all KOG-level analysis from the mevalonate pathways and accordingly dropped figure 2B, replacing it with a pathway level analysis previously presented in the supplementary material. We therefore agree with the reviewer that such a presentation is more in line with the strengths of our methodology.

F. On line 471 and subsequently, could the authors justify why they think that the Asgard contribution is most likely an underestimate? Given the points discussed above, I’d rather say (and agree with the authors with what is said on lines 475-479) that it is rather that of the other bacterial lineages’ that could be underestimated.

Our argument is based on the partial assembly state of Asgard genomes which is far less complete compared to the other prokaryotes in our data. The MAGs from Eme 2023 range between 77 and 97% completeness (25 and 75th percentile units) according to CheckM indicating that many genomes have proteins missing and are unable to contribute to the analysis. This is in contrast to the data on other prokaryotes which consists of closed assemblies. Furthermore, since the full distribution of Asgards is still being massively expanded further sampling is highly likely to provide examples of more eukaryotic-like proteins. However, in the revised Discussion, we chose to remove this argument, emphasizing more generally the uncertainty caused by the uneven and insufficient sampling of various taxa.

3. Functional interpretation

A. The above-discussed case of the mevalonate pathway questions the unit of observation/integration that should be considered. Should individual gene families be considered, or rather sets of functionally coherent gene families pointing towards a same origin? The latter could enable to gather more phylogenetic signal. Could this be accounted for in the process of calculation of the relative prokaryotic lineages' contribution, rather than in an a posteriori manner? The authors make some steps in this direction by attempting to integrate the history of individual genes that are part of a same function, but I find the cutoff employed to investigate potential functional modules very drastic: on line 307, the authors consider only lineages having contributed more than 20 EPOCs to a pathway, and thus found only Alphaproteobacteria and Asgard. It would be interesting to investigate what comes out if this threshold is lowered.

We agree that a single cutoff of 20 EPOCs is somewhat arbitrary. We balanced this number to only include pathways for which many EPOCs can be averaged, maximising the strength of our claims by avoiding small sets of data, while retaining as many core metabolic pathways as possible. Here, 20 EPOCs was found to be a good trade-off. In the revised manuscript, we provide examples of different limits of 10 and 5 EPOCs in supplementary figure 6 showing a more diverse view with stronger variance for smaller EPOC limits. Although this core message remains, very few pathways have stronger associations with diverse prokaryotes than with Asgard. For readers interested in details, we would refer to the full dataset on Zenodo.

B. For instance, for the case of the Myxococcota, it is unclear if the authors could reproduce the results of previous studies where genes involved e.g. in steroid biosynthesis were found to come from this lineage (Hoshino PNAS 2021). Could the authors add a sentence about this in the paragraph on lines 312-310, or elsewhere?

Based on the expanded data presented in response to the previous reviewer's comment, we describe a previously excluded diterpenoid pathway association with Myxococcota and contrast our findings to the data presented in Hoshino PNAS 2021, as per the reviewer's request. In short, our clustering data only sparsely covers the steroid synthesis pathway post isoprenoid biosynthesis, and we can neither support nor reject the observations presented in Hoshino's much more specialized analysis. However, the tentative observation of Myxococcota derived diterpenoid biosynthesis could lend some support to their data which as indicated in the text (line 310).

C. Later, another threshold is employed to look into prokaryotic contributions to individual EPOC, that of a number of sequences (lines 325-326). This directly points at the potentially strong impact of the biological sampling in the initial genomic dataset, and how it could artificially diminish the apparent contribution of under-sampled lineages. Same here, what if the threshold was lowered and the dataset enriched? Would more significant contributions from other prokaryotic lineages be leveraged?

The reviewer is correct that the cutoffs employed are numerically different, but we note that these are two strictly different comparisons assessing two different hypotheses and requiring different approaches. The prior is an observation of a general lack of consistent contribution of any one prokaryotic clade across different pathways, shown in Figure 4 and supplementary figure 6A. The cutoff employed there is therefore on the level of systems in a pathway, 20 EPOCs (now also presenting 10 and 5, at the *reviewer's* request). However, we also considered the possibility that the contributions from diverse prokaryotes could not be revealed when averaging across pathways if several key EPOCs were associated with diverse putative ancestors, each with a locally strong association, perhaps even functionally coherent on the level of enzymology. To address this question, we instead examined individual EPOCs to see if we could find any traces of widely present association with diverse prokaryotes. Here we applied a different and more stringent cutoff on the level of sequences and ELW within an EPOC as we cannot rely on averages across a pathway as in the previous case. The cutoffs presented are such that any single data point retained was, by our methodology, highly unlikely to be derived from any other prokaryotic clade than the one presented. We observed few such cases but present those we did in the supplementary material, noting no consistent functional trends. The problem of parameterisation obviously remains here as there is no agreed upon standard. Here we present only strongly supported associations that would not be substantially affected by the specifics of parameterisation. We concur that the initial distinction between hypotheses was not clearly presented and have revised the text accordingly (line 318).

4. The phylogenetic approach

D. I would need clarifications regarding the topology tests. For instance, as pictured in Figure 1A (or ext Data Fig 2), the ELW topology tests are ran as if the root of the gene trees was known, which is not the case. Could the authors please clarify the impact of an arbitrary choice of the outgroup on the results?

We here refer to the methodology presented in Rambaut 2002 which is invariant on tree rooting and calculated on unrooted trees. The ELW metric is derived from a ratio between the log likelihood values of trees constructed under different evolutionary scenarios. In our implementation each evolutionary hypothesis is enforced by a corresponding constraint tree, one for each of the 12 closest prokaryotic clades present on the tree, retaining all leaves, the computational root and the evolutionary model of the original master tree. As such, the variables which affect the ELW metric are all kept constant between hypotheses tests. The choice of computational root is irrelevant for the ELW metric as long as it is kept consistent between competing hypotheses, as all likelihoods are calculated as differences from

the most likely hypothesis. As per a comment from reviewer 1, we further clarify that all trees are unrooted.

Minor comments:

Figure 1. The color code should be changed as here it is a cyclic set of colors being used and the same color (for instance, orange) is employed more than once, making the identification of the procaryotic lineages difficult and ambiguous.

The colors have been changed to address this issue.

- While membrane lipid origins is discussed, the origins of the nucleus is not. Could the authors add a couple of sentences about what is their view of the origins of the nucleus in the frame of a dominant contribution of Asgard and then Alphaproteobacteria to eucaryogenesis?

As per the reviewer's request, we have added a brief discussion on the origin of the nucleus.

Referee #1

We thank the reviewer for their continued careful review of our work and for the time investment in independently verifying our results. We are pleased that our response to the reviewers' initial comments regarding the LCA representation and selection, taxonomy label selection, clade selection, and stem lengths appear resolved. We have further addressed our data to try to alleviate the reviewer's concerns but need to first clarify matters regarding the current methodology in response to the reviewers' further comments.

The reviewer mentions in their review that they suspect our data suffers from:

- **Oversplitting of EPOCs (resulting in inflated taxonomic contributions)**
- **Inadequate taxon sampling (resulting in taxonomic misclassification)**
- **Inclusion of out paralogs and contaminated eukaryotic sequences (resulting in taxonomic misclassification)**
- **Inadequate handling of paralogs, e.g. different small GTPases, actin paralogs and ubiquitin(-related) families in LECA (these form their own KOG and random selection of one EPOC per KOG does not address this properly, leading to incorrect taxon contributions of LECA content).**

Here we address these comments in turn. For a general summary of our final statements please refer to part 6 below.

1. Paralogy and KEGG representation

We agree with the reviewer that proper partitioning of the eukaryotic (and prokaryotic) sequence space is important to try to represent biology. In this work, we carry out extensive sequence profile clustering in an attempt to, as far as possible, collapse paralogs into single clusters, a problem that is common in phylogenomic studies. In our current version of the data, which now includes a 20 % pangenome criterion for eukaryotic sequences, in addition to the previously criteria, we resolve 4300 EPOCs covering 2100 KOGs, a ratio of 2 to 1. However, we must emphasize that this number is skewed towards a few EPOCS with high observed paraphyly. The median number of EPOCS per KOG is 1 and only 150 of all KOGs are represented by more than 5 EPOCs. Furthermore, with the latest 20% pangenome definition for Eukaryotes (see also the response to reviewer 3), few EPOCs have multiple Eukaryotic clades, with a total of 4600 analyzed eukaryotic clades across 4300 EPOCs. Thus, there is one-to-one-to-one mapping for most KOG/EPOC/clades, indicating that our partitioning agrees with that in KEGG and in general there is no great amount of paraphyly. Nevertheless, the reviewer suggests that, if we do not accurately account for unevenness of this mapping, certain gene families that have undergone extensive paralogization post LECA would account for more weight than they should, and because these are predominantly derived from Asgard (according to Vosseberg et al), we might be overestimating the ratio of Asgard contributions.

There are, in effect, two levels to this issue. First, we need to be sure that mapping of several EPOCs to a single KOG does not affect our results (potential oversplitting relative to KEGG), and second, we have to show that KOGs themselves are not highly paralogous relative to LECA (oversplitting of KEGG relative to LECA), a far more difficult task given that the true LECA gene set is not known.

For the first issue, we rely on the well-established KEGG ontology to be our representation of biology. As such, all aELW is and have always been normalized per KOG and we can never attribute more than 1 unit of ELW per KOG. For the purpose of our calculations, it does not matter whether a KOG maps to one or multiple EPOCs, the total ELW of all prokaryotic weights still sums to 1 and the total weight is equal to the number of KOGs within the analyzed set. So, if an Asgard-enriched KOG maps to several EPOCs, it will only contribute a total ELW of 1 for the purpose of averaging across maps no matter if there were 1 or 10 EPOCS annotated as Asgard for that KOG. In addition, as shown in supplementary figure 3, Asgard are not specifically enriched in the highly paraphyletic KOGs, showing a large contribution of Myxococcota and other diverse bacteria, made more significant by their globally low aELW. We more explicitly show this independence through the example of one randomly selected EPOC per KOG in Figure 1. This eliminates any possibility of normalization errors during aELW calculations, and the resulting data is highly similar to the core data. Thus, we would note that there is no specific trait of Asgard related KOGs, nor any KOGs for that matter, which would cause them to be overestimated due to the number of EPOCs per KOG.

The second aspect, the possibility of a significant mismatch between the KOGs in KEGG and the genes represented in LECA is much more difficult to assess. If LECA possessed a single copy of a gene serving a single function, which later proliferated into multiple paralogs, each with its own KOG, then, this might not be accounted for accurately in our data. Here we simply do not know how KOGs, as defined by KEGG, map onto the true LECA gene repertoire. Unfortunately, due to specifics of the Vosseberg study, it cannot help us here, see below. We can, however, point to the fact that certain proteins with diverse paralogs across Eukaryotes, for example ubiquitin, actin and tubulin do not contain many paralogous KOGs in KEGG. Ubiquitin is represented as 2 KOGs, with ubiquitin-like ribosomal proteins representing another 2. All tubulins are mapped to 5 KOGs, and all actins are under a single KOG. Within each of these KOGs, there are multiple genes, and many sub-functions, but because our mapping relies on the KOG id, we cannot be overcounting by a large margin. Actually, we might be undercounting as all eukaryotes have many paralogs of these proteins, suggesting the same was the case in the LECA. Yet we can never attribute more weight than the number of unique KOGs. This is one of the reasons why the KEGG ontology was chosen in the first place, as opposed to GO which has a much more fine-grained description of biology, including multiple different accessions for paralogs. If we were using GO, we would be far more concerned with the situation the reviewer is describing but with KEGG even the most paralogous families are condensed into relatively few KOGs.

Finally, and as stated previously, we do not attempt to define an accurate and complete view of the gene complement of the LECA. Such an endeavor is beyond the current methodology and, given the continuous development of databases, a constantly shifting goal. Rather, we are trying to provide a broad view of the origins of eukaryotic protein families we can trace to LECA. Here we highlight the consistent role of Asgard in association with most if not all components of the Eukaryotic cell. In this context we would note that even if a single copy gene was present in the LECA, and subsequently diversified post-LECA to seed a broad family of functionally distinct paralogs, should not this be considered an indication of a strong Asgard association across a wide component of the extant eukaryotic cell?

2. On the work of Vosseberg et al

The core of the reviewer's concern regarding the specific paralogization of Asgard genes references the work by Vosseberg and colleagues (<https://pubmed.ncbi.nlm.nih.gov/33106602/>), which suggested that certain cellular components and pathways were particularly prone to paralogization post LECA. Vosseberg et al used PFAM to define a LECA estimate of ~12,700 genes, quite close to our set of 13,500 EPOCs. However, as a comparison is explicitly requested by the reviewer, we must voice our concerns related to the work of Vosseberg and our reasoning for not applying their conclusions to our data. These are primarily methodological and sampling issues but we gauge would have a large impact on their overall conclusions.

We would first and foremost highlight the narrow representation of Asgard archaea in the work of Vosseberg et al, including only 9 partially complete Asgard genomes. Given the rate at which the Asgard sequence representation has developed, their 2020 Asgard database does not come close to accurately reflecting the diversity of Asgard genomes available now. Thus, all Asgard-related conclusions of Vosseberg et al need to be assessed highly critically. As an example of this limitation, Vosseberg et al only detect 16 of 12,700 families which consisted of only Asgard and Eukaryotic sequences are present. Given the well-established close evolutionary relationship between Asgard archaea and eukaryotes, this tiny number of families uniquely shared by Asgard and eukaryotes is highly suspicious, and indeed in our analysis, we identified 305 such KOGs in our core data (total 2100 KOGs) enriched in pathways related to among other, endocytosis, ER processing, RNA polymerase, the proteasome, ribosome assembly etc, pathways well represented by figure 2. This severe underestimate of the Asgard contribution essentially invalidates the conclusions of Vosseberg et al regarding paralogs in the LECA, at least, for genes of Asgard origin.

Additionally, the work of Vosseberg relies on a sparse and biased representation of eukaryotic biology, a substantial issue given that they aspire to infer details on gene paralogization in eukaryotes post LECA. Their database is derived from Deutekom 2019, <https://doi.org/10.1371/journal.pcbi.1007301>, a eukaryotic dataset that includes 122 Opisthokonta, 6 Amoebozoa, 23 Archaeplastida, 3 Crypto-Haptophyceae, 13 Excavata, 41 SAR, and 1 unassigned species. Even without accounting for annotation bias and sequence diversity, this is a highly inadequate representation of eukaryotic biology which is as much of a problem as the under-representation of Asgard for the identification of ancient duplications.

We also find problems with the phylogenetic inference and interpretation in the work of Vosseberg et al. As a characteristic example, we consider their supplementary list of the LECA families they assign to intracellular processing and sorting (COG category U). For this functional class of genes, they assign 408 of 514 families to diverse bacteria or to more ambiguous taxonomic labels such as "bacteria", "eukaryotic" or "simply cellular organisms" indicating a substantial bacterial presence across this cellular component we detect as Asgard dominated. Their shallow taxonomic annotation of sister clades, ex. "cellular organisms" is also strange indicating a potential mixing of their outgroups. Note that for comparable subsets in our current data, these components are almost entirely Asgard derived. The remaining 116 families are assigned to a mixture of various Archaeal clades as well as mixed clades labeled "Asgard + TACK group", and only 67 of these are to Asgard. Questions as to the taxonomic definitions of Vosseberg et al aside, we also have to emphasize that their analysis 1) includes a data reduction strategy which does not account for intrinsic sequence variability across families, relying on a 60% sequence identity clustering approach, 2) involves no model selection and no explicit rate category estimations and instead uses a blanket LG4X model for all the trees, and

3) includes no error bars on these assignments but assumes single tree topologies to be literally taken as ground truth. All of these practices put into question the validity of the conclusions, especially given the expected wide diversity of MSAs (not deposited anywhere, to the best of our knowledge) and the high noise level that is intrinsic to single-gene trees. For all these reasons, the results of their analysis are, in our view, not directly comparable to our current results, and their reliance on absolute tree topology and potentially inappropriate subsampling represent major liabilities. Based on the aforementioned single-gene trees, Vosseberg et al. attempt to answer the far more difficult question of within Eukaryote paralogization, rather than just sister clade assignment, once again relying on the absolute tree topology. As such we cannot help but seriously question the methodology used for inference of the pre-LECA and post-LECA paralogy in the work of Vosseberg et al.

However, methodological and database issues notwithstanding, how to interpret their results in relation to our data is also not obvious to the authors. Given their final assignment of 67 out of a total of 514 families to Asgard, they conclude that 55 of these 67 include at least one paralog (82%). We note that we cannot find the actual number of paralogs in the data, nor any error estimates for this assignment. The inverse statement is that of the non-Asgard clades, 326 of the remaining 447 include paralogs (79%). Based on these observations, Vosseberg et al. calculate odds ratios for observing duplications to be around 2, statistics we cannot replicate as we do not know the estimated number of paralogs, but for which we trust the authors. Given this skew in Asgard vs bacterial presence it is unclear to interpret this data. As their Asgard contribution is small in the context of the analyzed dataset, how can Vosseberg et al. be certain that the fraction of Asgard descent within intracellular processing and sorting is comparable with the larger bacterial component, or representative of the functional class of genes as a whole?

We definitely agree that post-LECA paralogization in eukaryotes is an interesting and open question in evolutionary biology. However, this question is by no means closed by the work of Vosseberg et al., and given the shortcomings of their data and methodology, must be revisited with modern Asgard representation and demonstrably robust methods. Due to these limitations, we cannot consider using their data in our current manuscript and think that dedicating effort in reviewing their work is beyond the scope of our current manuscript but do so here as requested. We do agree with the reviewer that paralogization might have an effect if indeed the KEGG ontology contains significant mismatches with the true LECA, however since the latter is not known we do not want to speculate on its impact as we might just as well be undercounting paralogs, we briefly elaborate on this in the latest version of the paper but strongly contest the notion that the role of paralogy on LECA is a well-defined and already concluded subject of study. It is not, and the observations of Vosseberg et al. cannot and should not hold any sway over our current study. We already state all our claims with supreme caution, presenting all the supporting data and show that our general interpretation is robust over a wide set of parameter ranges and partitioning of the data, now as per the comments of reviewer 3, extended to include a 20% pangenome filter for eukaryotes, once again transforming our sequence representation but barely affecting the overall results.

3. Taxon sampling

The reviewer further claims that our data suffers from inadequate taxon sampling which can bias the results. To our regret, we missed this criticism in the previous round of review and

sincerely apologize for not including it in our first response. Unfortunately, we are unsure what, exactly, the reviewer is referencing, but we refer to our response to reviewer 3 in the first round of reviews, regarding the representation of Myxococcota. There we raised the point that, even if we use all data present for that clade in GTDB, our actual sequence level representation of the clade would barely increase, adding only about 200 new protein clusters and showing that the coverage of the database relative to all known metagenomic data from that clade, which we can analyze, is robust. In contrast, the addition of metagenomic Asgard data increased our sequence representation substantially, by over 8000 new clusters, highlighting the still growing sequence space within that clade. We hope that this explanation is sufficient to address the respective comments of reviewer 1 as it was for reviewer 3.

4. Independent analysis carried out by the reviewer.

Finally, we must address the independent analysis carried out by reviewer 1 in response to our manuscript. They highlight two examples drawn from our data, cysteine desulfurase NFS1 assigned to Asgards, and FeS assembly protein ISCU, with a mixed assignment, with possible Asgard association. The reviewer claims that our methodology is inadequate given our inability to correctly assign these proteins to Alphaproteobacteria, the claimed ancestry of these proteins. This claim is in turn derived from the paper of Munoz-Gomez et. al 2022 (<https://pubmed.ncbi.nlm.nih.gov/35027725/>) which defines a set of Alphaproteobacterial marker genes for use in phylogenomics.

In evaluating these claims, we start by examining the study of from Munoz-Gomez et. al 2022, an excellent resource, where these proteins were chosen as candidate for Alphaproteobacterial markers. Their initial choice of database contains a narrow but well curated set of Eukaryotes, not overtly biased. Crucially, however, it does not include any Asgard archaea. This is a major issue when referencing the work of Munoz-Gomez et al to claim that NFS1 and ISCU cannot be of Asgard origins. It is beyond the scope of this study to repeat the work presented in Munoz-Gomez et al with our much broader representation of Asgard sequence diversity, but we must stress that, by the design of their study, Munoz-Gomez et al had no way of telling whether these proteins were inherited from Asgards or not. Additionally, considering supplementary figure 2 from Munoz-Gomez et al, attached below, they show that theirs is the only study of four compared that assign either NFS1 and ISCU (labeled NFU1) as Alphaproteobacterial marker genes. Their figure clearly shows that these proteins are not consistently identified as such markers, and that in general there seems to be little consensus in the field as to what constitutes a marker. We have little doubt that, within the parameters chosen by Munoz-Gomez et al, NFS1 and ISCU pass as Alphaproteobacterial markers although it should be noted that NFS1 shows an especially patchy presence in the data presented by Munoz-Gomez et al, missing from more than half of the investigated Alphaproteobacterial species. Regardless, once again, their work does not and could not provide any indications these proteins are not of Asgard origin.

Accordingly, we updated figure 3 to include Alphaproteobacteria in diverse bacteria noting their possible association in line with reviewer one's data.

As for the ISCU tree presented by reviewer 1, we likewise withhold a verdict because we are unsure what we can conclude from these results without further methods or data being provided. However, we must raise our concern that the Alphaproteobacterial sequences highlighted as “real alpha sister” are present as multiple singleton leaves inside the eukaryotic clade, something which seems odd for the proposed ancestral sequences. We managed to identify the source of one of these sequences 434131.NRI_0293.COG0822, as stemming from *Neorickettsia risticii*, an obligate intracellular Alphaproteobacterium inhabiting parasitic flatworms, and the causative agent of neorickettsiosis in horses. The other highlighted accession is from *Anaplasma marginale*, another pathogenic obligate intracellular Alphaproteobacterium inhabiting metazoan cells. As these species are obligate intracellular pathogens with strong evolutionary association with Eukaryotes, it is no surprise that they branch from deep within Eukaryotes, most likely as a result of HGT, which is highly prevalent among Alphaproteobacteria, and pathogens in particular. Issues like these are the reason why in our work we explicitly enforce per clade pangenomes to avoid misinterpreting close specific associations as ancestry. It remains unclear to us why the reviewer claims that these are the ancestral sequences when they are obviously derived from eukaryotes, especially as this is resolved in the very trees the reviewer uses to claim otherwise. We cannot help noting that this raises further questions as to the methodology employed by the reviewer in both examples and the relevance of these examples for the criticism of our work.

5. Further improvement to methodology

Independent on the specifics of the provided trees, we sought to improve the methodology of our study in response to the criticisms of reviewer 1. It is possible that, because our EPOC definition relies on unsupervised clustering, a fraction of our data is improperly partitioned in a way which might affect subsets of the results. So far, we have included different taxonomic definitions and multiple definitions of pangenomes, all changes which fundamentally alter the underlying representation and annotation of the data, yet not substantially affecting our main conclusions. However, from the trees presented by reviewer 1, we observe that there exists an overlap between EPOCs when resolved together on trees. Although most KOGs only include a single EPOC and our methodology is robust to such paralogous clades, those marginal cases are nevertheless suboptimal and might possibly bias the results. We would therefore like to compare our data with EPOCs formed from clades explicitly resolved to be monophyletic trees prior to assigning any prokaryotic associations.

Accordingly, in the second revision of our manuscript, we added another level of validation through the process of cluster “annealing”, including the new supplementary figure 8. As described in detail in the amended Methods section, we start by utilizing the existing clusters derived from mmseqs2 clustering and turn these into a large database of explicit HMM models using muscle5 and HHsuite, the same pipeline as described before for identifying prokaryotic-eukaryotic orthologs. However, rather than directly relying on this data for EPOC generation, we now perform all against all HMM-HMM self-searches for both prokaryote and eukaryotes and subsequently cluster the resulting data using a greedy set cover approach, as per mmseqs2. These superclusters are broader than the initial EPOC clusters but notably do not collapse the data significantly, only from 1.6 million eukaryotic clusters to 1.5 million and from 420.000 to 280.000 prokaryotic clusters, showing that the initial clustering is already rather condensed,

given the sequence diversity. However, clustering from HMM-HMM searches is prone to overclustering of protein families with high pairwise sequence similarity. To avoid such cases, we took a cue from the reviewers' trees and asked whether these superclusters were well specified when resolved in trees, or are there clear sub-clades indicating the existence of distinct gene sets within a supercluster?

To address this question, we align and create FastTrees for all our superclusters (Iqtree is unfortunately prohibitively expensive when handling 1.5 million trees). To identify separated regions with dense leaves, we first rely on our pruning technique (described in the EPOC generation section of the Methods) to remove long branch outliers which might otherwise bias subsequent steps. Then, we calculate all against all pairwise leaf distance matrix and embed it into 2 dimensions using UMAP. This embedded representation reflects the topology of the master tree and can easily be clustered using HDBSCAN to define regions of separate density. Notably, this technique is not dependent on any hard branch length cutoffs or extrinsic parameterization and relies solely on the observed distribution of the data to partition trees. We found that in all cases such clusters represented monophyletic subtrees of the original trees and were well partitioned given the existing diversity. If such sub-clusters were detected, these were extracted separately and used as the basis for new EPOCS, with their own HMM profiles and annotation. This was carried out for both eukaryotes and prokaryotes, under our standard 20 % eukaryotic, 50 % prokaryotic pangenomes criteria. This "annealed" EPOC set is defined much more stringently and has the advantage of being guaranteed to form monophyletic, well separated clades, given the distribution of sequences from HMM-HMM searches. In many cases, these subdivided clades simply correspond to the original EPOC clades they originated from, but in other cases, cluster memberships is refined, with less overlap between clusters. We give a few examples of partitions obtained with this approach in supplementary figure 8A.

As further outlined in supplementary figure 8, comparing the orthogonal EPOC representation of these annealed EPOCs to the original core EPOCs, we observe a broad agreement in the results despite the significant difference in EPOC definition. Overall, the dominance of Asgard archaea remains, being consistently observed across most of the functional classes of eukaryotic genes. Additionally, multiple independent prokaryotic taxa exhibit broad but non-specific associations to multiple metabolic pathways. All this data is available together with the latest Zenodo deposition for general use and comparison.

We hope that this complete reparameterization of the problem, together with previous improvements and the extensive discussion above, shows that our results, although differing from previous studies, are in fact highly robust.

6. Summary

We appreciate the thorough assessment of our work by reviewer 1. However, we also must raise concerns in regard to the critique of our findings which we consider to be partially erroneous and unfair. We apologize for the extensive rebuttal and any wording that might be perceived as harsh but we must state our position firmly in the face of this criticism. Therefore, in conclusion:

- Our work has been criticized as overestimating Asgard contribution to eukaryogenesis due to the allegedly known paralogy within Asgard dominated pathways. This criticism is

invalid because 1) proper normalization was employed to prevent overestimating ELW when multiple EPOCS per KOG are present (Figure 1), and 2) the underlying KOG representation in KEGG merged paralogous protein families under single accessions. Additionally, the underlying premise cannot be considered valid as it relies on the outdated study of Vosseberg et al which cannot be considered accurate given their employed methodology and the lack of representation of Asgard archaea. As such, the impact of their conclusions on our data is not obvious.

- Independent analysis carried out by reviewer 1 is hard to assess as no methodology or underlying data were provided. Claims were made based on the work of Munoz-Gomez 2022, which due to their lack of Asgard sampling, have no our current results. However, from the provided, incomplete information, we can detect significant issues with both trees constructed by the reviewer and must question their validity as detailed above. It should be further noted that in the revised version of our manuscript, we have completely reprocessed the data and increased the stringency of the annotation criteria to avoid any potential errors due to over-clustering or under-clustering of the data.
- The remaining comments in regard to taxonomic sampling or contaminations are either covered in our responses to other reviewers or are presented anecdotally without evidence or examples.

Referee #2

We thank reviewer 2 for their support, constructive critique and time invested in assessing our work.

Referee #3 (Remarks to the Author):

I thank the authors for this revised version and the discussion in the rebuttal, the methods are now more clearly described and the results are more nuanced and discussed in the manuscript.

However, there are some of my concerns about the methodology employed that are still standing.

Major points:

- Effect of the taxonomy

Given the employed approaches that heavily rely on taxonomic assignments, it was much needed to investigate the effect of the taxonomy and test the genome-diversity normalized taxonomy from the GTDB. The authors have established that using the GTDB taxonomy was not affecting their main results, even if I think the proposed origins of eucaryotic core genes would have been more precise if the GTDB taxonomic grouping would have been kept all along, as the NCBI taxonomy can be somehow arbitrary for certain lineages. However, given the high rate of taxonomic assignment to the GTDB mentioned in the Methods (line 705), I do not understand why only a portion of the dataset seem to have been analyzed on Extended Data Figure 1 (all data: “n = 5142”, vs all data in Figure 1B: “n=14353”). Or is this a typo? Could the authors please explain?

The number of reconstructed EPOCs are constrained by several factors relating to the sampling and presence of phylogenetic labels as outlined in the methods. As such, when completely redefining taxonomy, a slight variation is expected in the exact number of EPOCs for which reliable sister-clades could be assigned, especially when those EPOCs have very few prokaryotic sequences present. That being said, the previous number was a regrettable plotting error. In the updated data, which also includes a global 20% pangenome filter on the eukaryotic side, outlined below, there are a total of 13500 EPOCs of which 13250 can be accurately reconstructed in the GTDB data.

- Ability of the phylogenetic method to trace genes back to the LECA.

In my initial review, I pointed out this issue and mentioned as an example the case of the Cyanobacterial contributions that seem to be at least partly false positives. On lines 98-100: “post-LECA horizontal transfer between eukaryotes and prokaryotes » are claimed to be dealt with (“prevent such genes from affecting our analysis »), but this is not the case, since e.g., the Cyanobacteria case is still prominent. The authors somewhat diverted my comment by answering they were not reconstructing the LECA proteome (I apologize for the semantic issue, I meant, as the authors state more precisely, “eukaryotic core genes tracing back to LECA”), leaving the case of transfers vs vertical transmission unanswered. Further, transfers also occur between eukaryotes. In the end, my question is: how can the proposed method really ensure that genes shared by at least 5 distinct taxonomic labels within eukaryotes trace back to their common ancestor if the vertical transmission is not really tested for? To me, this point remains unaddressed, and relying simply on taxonomic distribution to infer ancestry of the genes is too simplistic given the complexity of the question. Showing some phylogenetic trees for several cases could help convince about the validity of the approach (I believe

this was also a request of reviewer #1).

These points constitute severe limitations of the Method, which I think deserve at least a point in Discussion. Thus, the confidence of assessment of genes to FECA should be added in Discussion, along with a discussion on the case of the inference of genes of Cyanobacterial origins, maybe on lines 483-484 where the authors say “Further, our pangenome-based approach to exclude late HGT events from ancestral inference may overlook possible contributions from narrow or severely under-sampled clades”.

We thank the reviewer for these comments and are pleased to hear that our revision has improved the presentation of the manuscript. We do agree with the reviewer that HGT is one of, if not the main methodological hurdle for these deep reconstructions. We approach this issue from two directions, primarily through the implementation of the pangenomes, enforcing that all proteins we include in our data are broadly present across taxa. This is a key improvement from prior studies and makes a large difference as indicated in Supplementary figure 1. Even a mild pangenome definition is sufficient to avoid the numerous (relatively) recent eukaryotic/prokaryotic transfers. Additionally, we enforce a minimum clade diversity on the level of EPOCs, chosen as 5 LECA spanning clades, however, note that our reconstructions are completely independent on this number, see below. We certainly would like to use some “true” statistical test of HGT, however implementing such would require mapping HGT across more than 40,000 Bacterial, Archaeal and Eukaryotic genomes, across 2 billion years of evolution. This would be a massive endeavor even compared to the extensive phylogenetic reconstructions done in this work, and as such, is far beyond the scope of any single paper. We are unaware of any methods which even comes close to achieving this goal and therefore must rely on the more practical definitions within this manuscript. However, although we cannot entirely exclude the presence of HGT, we have to emphasize that our approach provides confidence that the associations we analyze are at least all ancient, predating the LECA, and broad, existing with a wide scope across all prokaryotic taxa.

Nevertheless, in response to the reviewer’s remaining concerns, we sought to further reduce the risk of HGT affecting our conclusions. As such we have now also implemented a pangenome criterion on the eukaryotic databases. In the previous data it was possible, but unlikely, that a protein broadly present in prokaryotes could have been transferred multiple times into several separate eukaryotes, eventually allowing it to pass our criteria for EPOC definition. While we do think such a model is not prevalent enough to affect the overall conclusions it might have bearing on individual maps, especially within the core metabolic scheme. In order to remedy this, we have now implemented an additional pangenome filter on eukaryotes as well. This ensures that not only does any EPOC clade analyzed stem from a broad taxonomic footprint but also that the presence of that gene within that clade is likewise broad. For eukaryotes however, due to the variable protein lengths and domain compositions we must have a slightly less strict definition, compared to prokaryotes, retaining a pangenome cluster if it is present in at least 20 % of species within a clade, rather than the 50 % enforced for prokaryotes. At 50%, we noticed some core and well established KOGs losing a significant number of eukaryotic proteins which based on their annotation should be retained.

This pangenome definition required a recompilation of all data, and as such, all figures are now updated, derived from a eukaryotic database meeting the 20% pangenome criterion. As the reviewer can see in the revised manuscript, although the total number of EPOCs meeting the criteria for further analysis has decreased, and some specific associations have shifted for

individual maps, this modification of the procedure had only a minor effect on the overall results and has not affected the general interpretation which has now proven to be highly robust to various levels of HGT filtering, the choice of taxonomy, database selection etc.

Additionally, to address the concerns regarding the choice of at least 5 distinct taxonomic labels per clade, we present the following plot, also added to supplementary figure 7. Here we show the distribution of aELW values across the number of distinct taxonomic labels per clade, highlighting that our results are mostly independent of this parameter. EPOCs with clades containing exactly 8, 10, 12, 15 or 20 specific labels show the same overall global distribution of ELWs, highlighting the strength of the Asgard signal across a broad set of eukaryotic clades of different diversity.

Total cELW

If we plot the same graph but include the filtering criteria, clades with more than X clades the result is even less variable.

Total cELW

So, although the choice of five clades is in fact not essential, we nevertheless maintain the criterion to avoid analyzing clades with a narrow eukaryotic presence.

This observation is also important in the context of the unexpected prevalence of Cyanobacteria. We expected in the reconstruction of 20% eukaryotic pangenomes to lose some overall cyanobacterial aELW as we could not exclude the possibility of HGT effects. However, the observed number is very persistent between reconstructions with and without pangenomes at any definition. This, as well as the plot above, indicated that the cyanobacterial contribution is associated with eukaryotes independent of taxonomy and on the breadth of the clade. Further, because no specific pathways related to photosynthesis or photosynthesis derived functions are detected in our reconstruction, we strongly doubt that this association is driven by post LECA HGT related to the acquisition of the chloroplast, or dominated by organellar gene transfer from the chloroplast to the nucleus.

We acknowledge that this association is notable and broad and cover it in the paper, elaborating slightly from the previous version. We would like, however, to stress here that, as for almost all prokaryotic association, apart from Asgard and Alphaproteobacteria, there seems to be little to no functional trends here. As such, Cyanobacteria should not be considered different from Actinomycetota or Beta- or Gammaproteobacteria in their mode of contribution to eukaryogenesis.

Under our current parameterization, it would be possible that a gene is repeatedly transferred from the chloroplast to multiple individual eukaryotic lineages, and then spread within those lineages via further transfer to meet the pangenome and clade inclusion criteria. However, in entertaining such a model, given the knowledge of the plots above, we would also be acknowledging that such genes have spread across the vast majority of eukaryotes, and this would have happened a considerable number of times, yet without obvious functional trends, to affect eukaryotes this notably. We consider such a scenario to be highly unlikely. What appears far more likely, is that a non-negligible number of genes derived from cyanobacteria were

present in the Asgard ancestor. For such genes, as we touch upon in the revised Discussion, given the imperfect sampling of Asgard sequences, the closest ancestor for eukaryotes would appear Cyanobacterial rather than Asgard. However, this currently remains a speculation, and as we do not yet have quantitative data to support it, we would prefer not to elaborate on this hypothesis in this work. However, we hope that this explanation, as well as the scope independence of the aELW to be sufficient to alleviate any concerns regarding the potential effect of HGT on our conclusions.

Minor comments:

- **On line 426, the authors write: “Other bacterial contributions appear to be piecemeal, coming from a continuous but non-specific capture of genes from diverse bacteria, occurring along the entire evolutionary path from Asgard to the LECA (Figure 6). ”. I am not sure that the authors have much evidence from their analysis regarding the timing of acquisition, maybe nuance a bit and add “occurring possibly along the entire evolutionary path...”?**

We have changed the language essentially as suggested.

- Some of the figures in Sup Mat are not of good quality, for instance Extended Data Figure 6.

We apologize for this issue and made an effort to improve the quality of the figures in the revision.

- In figure 5C, the legend for “Ribosome” genes is missing.

Figure updated.

- Line 86-: “link” instead of “links”?

Corrected.

- Line 341: remove “astronomic” as unprecise and exaggerated?

We here referred to astronomic as geological time, something independent of evolutionary rate, but nevertheless deleted this word.

- Line 676: “HMM” instead of “HHM”.

Corrected.

- Line 717: typo at “redundant”

Corrected.

Referee #1 (Remarks to the Author):

I thank the authors for their extensive rebuttal and efforts to alleviate the concerns that were raised previously. The authors state in their rebuttal that part of these concerns were considered “erroneous and unfair”, but should realise that my intentions are sincere, and merely aimed to make sure that, given the contrasting results with previous works, the obtained results are not the result of methodological choices and/or artefacts, as this would do the scientific community no good. This is, simply put, my task as reviewer. It is therefore unnecessary for the authors to use a harsh tone in their rebuttal.

I would like the authors to kindly consider the following comments, related to my previous concerns; I apologise if I have not explained myself clearly.

We thank the reviewer for their revised statements and apologize for any perceived animosity. It was merely our intent to clearly state our position and our critical viewpoint on the basis of some of the literature referenced in the critique. We hope that the length of our response did not come across as hostile and apologize if it did.

1. I previously expressed my concerns about paralogy and KEGG representation. In the rebuttal, the authors focus at length on post-LECA duplications. However, my concern are related to stem (pre-LECA) duplications.

Here we would refer to our previous discussion on the mapping between KEGG and LECA. Our data would only be significantly affected by the proliferation of paralogs from FECA to LECA if those are encoded as separate paralogs in KEGG under different KOGs. We previously gave some examples of Ubiquitin, Tubulin and Actin, known to be paralogous in LECA. We noted that these are not encoded under a broad range of KOGs and are therefore not overrepresented in our data. Additionally, we showed that in supplementary figure 3 that Asgard are not specifically enriched in the most paralogous KOGs relative to their global distribution and are as such not overrepresented. Combined with the difficulty to reconcile results from Voseberg as well as the evolving picture of the gene repertoire of Asgard, we are unsure how to provide any more specific analysis related to our data because the history of paralogs from the FECA to LECA is simply not known in detail.

2. As previously indicated, the main result (“dominant contribution of Asgard archaea to eukaryotic gene content”) significantly deviates from other studies that revealed limited Asgard contributions to eukaryotic gene content, which the authors were asked to investigate/explain. The authors point out limitations (limited Asgard taxon sampling amongst others) of a previous study that could explain the obtained differences. However, a more recent study, described in Bernabeu et al, use a comparable Asgard diversity and obtain far less Asgard-dominant contribution to eukaryotic gene content,

indicating that taxon sampling alone cannot explain the difference between both studies. Could the authors comment on this?

We have so far refrained from commenting on the unpublished work of Bernabeu et al, currently available on bioRxiv as “Diverse ancestries reveal complex symbiotic interactions during eukaryogenesis” (<https://www.biorxiv.org/content/10.1101/2024.10.14.618062v2>) as we would not like to reference and discuss a preliminary version of their work in detail. While their approach is similar in spirit to ours, there are, in our opinion, decisions made in database curation and search methodology which are potentially problematic and, based on our experience, would have a significant impact on their data interpretation. These are points we also highlighted to Toni Gabaldon during informal discussions last year as we were both ready to simultaneously publish our preprints. We hope Bernabeu and coauthors address these in the final version of their manuscript; we expect that this to impact their results.

Comparing the results of their preliminary work to ours in any detail appears to be beyond the scope of the main text but we provide our perspective on those key technical differences here for the reference of the editor and the reviewers. As their work is preliminary, we do not have access to the full data and rely entirely on their own description in the preprint.

Database curation and Pangenomes

Bernabeu et al. start their study by curating two databases, one Eukaryotic and one prokaryotic, with a similar scope to ours. While we rely on the Eukprot V3, Bernabeu et al. compile their own sequence dataset from diverse sources, including from EukProt. While this may not be a major issue, they do not appear to clean their eukaryotic database from contaminating prokaryotic sequences, notably present in some Eukprot genomes. We are unsure why this was not done but, in our case, this filter removed around 40,000 sequences, the provenance of which would otherwise be misinterpreted. This issue might be further exacerbated in select genomes from P10K and Ensembl although we have less direct experience here. Bernabeu et al. also rather aggressively remove duplications by clustering at 80% sequence identity and 50% coverage, compared to our 95% identity and 80% coverage, possibly in lieu of a comprehensive tree reduction strategy.

Although it is unlikely that their final Eukaryotic sequence set is qualitatively different from ours (apart from the known presence of prokaryotic contaminants), their Prokaryotic set differs more substantially. Bernabeu et al. start with GTDB r207 and select the most complete genome from each GTDB genus (< 27,000 bacterial and >2000 Archaeal as per r226). Unless they make an undocumented exception for Asgard, at the time of their writing, this would only include 9 Asgard genomes. Although there is a significant amount of diversity present within the GTDB Asgard genera, this is not the same representation of Asgard as we use in our work. Additionally, they carry out pangenome construction following a notably different workflow compared to ours. They cluster sequences using mmseqs, limiting clusters to a 30% identity and a coverage of 50% using the mmseqs implementation of CD hit for clustering. They subsequently retain those clusters which are present in at least 15% of the species per order.

This is a highly permissive definition of a core pangenome, effectively compensating for insufficiently sensitive clustering of orthologs from distant taxa by relaxing the representation threshold, permitting both pseudo-paralogs (unclustered orthologs) and narrow-scope genes (potential eukaryote-to-prokaryote transfers) into their working sequence set. Additionally, because many orders in GTDB encompass fewer than 7 genera, this would automatically allow all the sequences from such genera into the dataset (even orders with up to 20 genera would only require 3 sequences to be included in their core pangenome).

Because of this parameterisation, their core pangenomes are very permissive, and while likely sufficient for sanitizing recent transfers for larger genera, it is highly unlikely to filter out any transfer relevant on the 2 billion year timescale of eukaryogenesis. As such, the comparable data series for our study would be the Global, no-pangenome filtering shown in Supplementary figure 1D. Under that construction approach, which we argue is far from being optimal for the study of eukaryogenesis, we observe ~5% Asgard ELW which is mostly compatible with the observation of 4% Asgard in Bernabeu et al. We have previously raised these concerns with Drs. Bernabeu and Gabaldon but have no way to know if they have revised their results following the initial preprint deposition.

EPOC/LECA-OG creation

While we believe that the majority of the discrepancy comes from the excessively liberal processing of pangenomes by Bernabeu et al, as outlined above, there are a few more aspects of their workflow which we consider likely to systematically bias their results. Here we are most concerned about the consistent lower sensitivity of Bernabeu's et al. search methodology compared to ours. As many Asgard sequences are divergent from eukaryotes, a certain, relatively high level of sensitivity is necessary in order to adequately identify ortholog candidates.

Primarily, we employed a substantially different methodology when it comes to our EPOC (their LECA-OG) construction. Bernabeu et al. rely on Orthofinder to carry out reciprocal best hit sequence/sequence searches using BLAST for sequences across eukaryotes. Initially, we relied on a comparable mmseqs sequence/sequence alignments for clustering, but noticed that this approach led to severe underclustering of a large part of our data, even relative to known annotation. We therefore switched to a much more sensitive, custom profile/sequence approach which helped condense our data further and broaden the scope of our clusters.

Following this step, Bernabeu et al. convert their sequence sets to profiles using MAFFT and HMMER, and perform HMM/sequence searches against their unclustered BroadDB. However, they then carry out another round of BLAST to verify the results. This seems problematic to us as it will effectively nullify the additional sensitivity gained from their profile construction. We are also slightly concerned about their statements that;

- *"We additionally removed hits with disparate sizes (when hit was >2.5 longer or <0.5 shorter than the query) to avoid artefactual clustering of families due to domain sharing."*

- *“We removed sequences with a completeness score lower than 75%, unless this removed more than 10% of the sequences in the alignment, in which case we removed the 10% of the sequences among those with the lowest completeness scores.”*

Given a supposedly proper parameterisation of the coverage during searches, we are unsure why these additional failsafe steps were added. It is possible that these only remove very rare cases, however it is not needed in our procedure and, based on the criteria they describe, should not be necessary in theirs either. We consider this as a possible sign of something problematic occurring in their pipeline, in turn, possibly affecting the integrity of their clusters and downstream profiles; something that should be thoroughly debugged rather than patched over. Additionally, a global coverage of 50% is rather permissive and will likely cause issues with alignment overextension from additional domains once their profiles are generated. As this coverage is maintained for prokaryotic searches, it is further unclear whether Bernabeu et al. can adequately distinguish between cases of correct coverage versus spurious hits against overextended profiles.

It is also notable that Bernabeu et al do not discuss any methodology for data reduction, as many of their clusters are likely too large for rapid and accurate alignment generation by mafft. In comparison, we perform data reduction through tree pruning to construct a set of 500 sequences, evenly sampling the sequence diversity within clusters. If this is not done carefully, Bernabeu et al. could easily bias their profiles or limit their taxonomic scope.

Finally, we are unsure how the procedure described under mLECA-OGs construction affects the partitioning of their data. Inbrief, they use their initial clusters and attempt to resolve paralogy based on the addition of prokaryotic sequences and the identification of monophyletic groups using heuristic values for automatic partitioning. This is conceptually similar to our annealing approach but fundamentally different in implementation. We initially attempted many similar schemes for tree partitioning and were not able to devise a procedure using scalar parameters. The procedure outlined by Bernabeu et al. using iterated reconciliation of sister clades based on a minimal outgroup purity was one of the approaches we tried that, in our hands, caused significant oversplitting in cases of poorly resolved trees or in the presence of deep HGT (both of which we suspect Bernabeu et al. are dealing with). This is the reason why our annealing approach relies exclusively on the internal tree structure of each cluster for partitioning in a parameter-free manner. Of course, it is possible that the different implementation of Bernabeu has somehow avoided this issue, but we suspect that their protocol for identifying monophyly might artificially split their LECA-Ogs even more, further limiting the sequence diversity for HMM profile generation.

As a result, they analyse roughly ~7500 LECA-OGs which should be compared to our 4300 EPOCs; however, based on their coverage of the KEGG metabolic scheme, they appear to identify fewer KOGs compared to us, while having more OGs. This is an entirely qualitative assessment based on the presented coverage of KEGG map1100 in their Figure1 (as the actual

data are not available with the preprint), but in line with their methodology, this would yield more narrow, less sensitive cluster profiles compared to our approach.

To summarize, we are concerned with the LECA-OG sequence representation in the Bernabeu et al. data and the difference in sensitivity between sequence-sequence (BLAST) searches followed by HMM-sequence searches, followed by another “verification” round of sequence-sequence searches, compared to our approach of iterated profile-sequence searches followed by profile-profile searches. While we cannot investigate their data directly and they give no statistics for their database size, specifically, cluster size distributions and profile nEFF, it is likely that they produce sequence sets with less diversity and a more narrow taxonomic scope, a partition which is possibly further diminished by their annealing approach, in turn producing less diverse profiles, primarily identifying prokaryotic sequences with high similarity to eukaryotic homologs. Therefore, we believe that they likely over-detect recent transfers compared to our procedure which, compounded by their very permissive core pangenomes, biases their view of eukaryogenesis. In the revised Discussion, we briefly mention the differences from Bernabeu et al and cite their preprint (new Ref. 72).

3. I previously pointed out two examples of proteins where the authors claimed Asgard ancestry, but for which previous studies have suggested Alphaproteobacterial ancestry (NFS1 and ISCU). Given that these examples featured in a main figure, I found it relevant to follow up on this. By providing a “quick” phylogenetic analyses I merely wanted to clarify my point, and urge the authors to have a more careful look at this. Rather than ‘shooting the messenger’ I was hoping the authors could simply provide a more sophisticated phylogenetic analyses of these protein families to alleviate my concerns. It would be great if the authors consider doing this.

We definitely never intended to ‘shoot the messenger’ and apologize if some of our language made this unfortunate impression. However, we had to critique the message that, according to the reviewer him/herself, was produced by a quick analysis.

We did perform a more detailed and more rigorous phylogenetic analysis of these cases. Briefly, we start with the known KEGG sequences for ISCU and NFS1, construct initial profiles and search against our prok and euk pangenome databases. We then align the resulting hits with muscle5 and construct a tree with IQtree2 and modelfinder. Then, we extracted the main eukaryotic clade and its related prokaryotic sister clades, verifying annotation using blast against clustered NR to ensure that annotation stays consistent for all included sequences. This sequence subset was then realigned and converted to a profile, which was imported into mmseqs and used as the query to carry out a second round of sequence/profile searches. The set of identified sequences was pruned using our tree reduction strategy. Then, we realigned the resulting sequences and constructed the final phylogenetic tree using IQtree2 with modelfinder; the proper annotation of sequences as "cysteine desulfurase" was verified using BLAST against clustered NR.

For ISCU, we cannot find any Alphaproteobacterial hits (Neither could Garcia and Gribaldo in 2023) meeting our criteria, and the tree is diverse including some Asgards. This alignment is only ~150 Aas, so the tree is highly informative. The NFS1 tree is likewise diverse with many eukaryotic clades, all annotated as cysteine desulfurase and showing clear sequence conservation on the MSA level. One of these clades has an Alphaproteobacterial sister (bottom left) but it is joined by many other prokaryotic contenders. Unless one considers only this single eukaryotic clade and takes the tree topology at face value, there is no support for an Alphaproteobacterial ancestry for this protein. Thus, we conclude, as previously suggested, that the ancestry of both NFS1 and ISCU are difficult to ascertain and the origin of these particular genes remains uncertain. . We believe that the current language employed throughout the manuscript reflects this level of confidence in assignments of individual KOGsWe did not consider it necessary to include these trees in the Extended Data but append them to this response.

Referee #3 (Remarks to the Author):

I would like to thank the authors for carefully considering raised methodological criticisms and in making constant efforts to alleviate them along the different rounds of revision, which led among others to include the “annealed clustering analysis” and further analyses that tested whether more stringent definitions of the “pangenome” would affect the results. I am now overall satisfied with the ways the methods are described and used, and the way the results are reported, interpreted and discussed, in the frame of a highly debated question in evolutionary biology.

I would just have one point left about the newly introduced “annealed clustering” analysis and perhaps how its results enlighten the difficulty to automate such complex evolutionary analysis: I believe this should be part of the Discussion.

Based on the new analyses including that of « Annealed cluster of EPOCs » that is supposed to better account for the overclustering of families (Sup Fig 8), it seems that the proportion of EPOCs of Asgard origins tend to decrease, while the relative order of the major bacterial contributors' changes. This is reflected by the low level of correlations between the contributions assessed via the initial and the annealed clustering of EPOCs (Sup Fig 8C) that seem consistent only for Asgard. While a dominant contribution of Asgard does still hold here as pointed by the authors, this demonstrates that when refining the methodology to better account for the complexity of the data, the results change. This point should be raised in Discussion to emphasize the potential limitations of the current methodology (probably in its last section). Hence, I don't totally agree with the conclusion of the authors in the Results (line 167) that there is a «robustness of the results to both cluster specification and taxonomy », while I agree that there is more consistency for the « core data » subset that are subsequently analysed.

We thank the reviewer for their consistent astute and relevant comments. We believe that addressing these comments has significantly improved the manuscript. The reviewer highlights a very important point in regard to the way this type of data can be interpreted, something which we have clarified in the revised Discussion. There is not only a clearer signal from Asgards compared to other taxa, but the consistency of that signal is also more robust to possible representational biases from data partitioning, both of which are noteworthy in their own right. We revised the critiqued line 167 to only comment on Asgards.

Minor points:

- Line 44: Maybe start already including references?

A reference was added.

- Sup Fig 1 => I know there is not an exact match between the NCBI and the GTDB taxonomy, but they mostly overlap... Would have been great to see similar/overlapping groups being coloured the same way between panel E and the others (as the authors did for the Asgard).

We made a further effort of reconciling the colours so that the majority of the equivalent clades among those displayed share a colour between NCBI and GTDB data.

- Sup fig 8A => y axis, an « e » is missing in « pairwise distance »

Revised and figure layout tweaked.

- Sup Figures should be renumbered by order of appearance in main text.

The figure order has been revised.

- Line 324: a space is missing

- Several references are missing from the Methods, in particular for the new section about “Cluster Annealing”. Could the authors please ensure to rightly cite all used software and methods? For instance, on line 599 => should “sklearn” be entirely spelled “scikit-learn” (and cited?). And HDBSCAN also cited? Etc...

- Line 733: please check the sentence are it may include several typos.

The additional sections have been carefully proofread and revised. The relevant software has been cited.

Review report of 2024-10-22122B

Tobiasson et al. have made an effort to clarify and address the issues that were brought forward in the initial round of review. Yet, it appears that some potential methodological weaknesses have been insufficiently addressed, casting doubt on the validity and robustness of the results and claims put forward in the manuscript. Specifically, the analyses seem to suffer from:

- Oversplitting of EPOCs (resulting in inflated taxonomic contributions)
- Inadequate taxon sampling (resulting in taxonomic misclassification)
- Inclusion of outparalogs and contaminated eukaryotic sequences (resulting in taxonomic misclassification)
- Inadequate handling of paralogs, e.g. different small GTPases, actin paralogs and ubiquitin(-related) families in LECA (these form their own KOG and random selection of one EPOC per KOG does not address this properly, leading to incorrect taxon contributions of LECA content).
- The above issues have a bearing on the overall conclusions. Given the strong deviation from previous estimates of Asgard contribution of eukaryotic gene content, the authors need to also compare the effect of previously used methods in order to claim that improved taxon sampling has caused these new, contradictory results.

Until these issues are carefully and convincingly addressed by the authors, the results described in the manuscript cannot be considered robust enough to warrant publication.

Examples:

Below are two examples presented in Figure 3, where Asgard archaeal provenance is proposed for some eukaryotic proteins involved in ISC synthesis that have a clear and previously demonstrated alphaproteobacterial origin (NFS1 and ISCU):

1. NFS1

This eukaryotic protein has previously been shown to be of alphaproteobacterial origin (e.g. it was even used as a marker gene for the mitochondrial placement in Munoz-Gomez et al 2022, *Nature Ecol. Evol.* 6, 253–262). Yet, in the current study, confusingly, an Asgard archaeal origin is proposed for this protein. A phylogenetic reanalysis of the data used in the current study, combined with data from previous studies confirms the alphaproteobacterial origin of NFS1 (see Figure 1 below), and indicates several methodological issues with the approach of the authors:

- In the current analysis, eukaryotic NFS1-like sequences are split in two EPOCs (EP00195P015578 and EP00528P004597), resulting in overestimation. Yet, in the tree below all bona fide eukaryotic NFS1 sequences form one clade, and that they originate from alphaproteobacterial.
- In each EPOC, the authors incorrectly link eukaryotic sequences to different clusters of non-alphaproteobacterial prokaryotic sequences.
- This reanalysis hence indicates that the obtained results are an artefact that is caused by oversplitting (an issue raised during the first review), taxonomic misassignment, and data impurity (contaminating prokaryotic sequences likely originating from transcriptomes; inclusion of molybdenum cofactor sulfurase outparalogs)

Figure 1. Phylogenetic re-analysis of NFS1 sequences in Tobiasson et al. showing alphaproteobacterial origin of eukaryotic NFS1. EP00195P015578 EPOC coloured blue, EP00528P004597 green, sequences from previous studies red (alphaproteobacteria and mitochondria) and black.

2. ISCU

As for NFS1, eukaryotic sequences are spread over two EPOCs (EP00629P016341 and EP00668P041674), each of which are matched with the same cluster of prokaryotic sequences; combined with COG0822 sequences corresponding with ISCU and related prokaryotic sequences, the following artefacts become apparent (see Figure 2 below):

- Clear eukaryotic clade with sequences from both eukaryotic clusters reveals oversplitting (resulting in overestimation)
- With the exception of one betaproteobacterial sequence, eukaryotic sequences are closest related to alphaproteobacterial sequences that were not included in the present analysis, pointing at inadequate taxon sampling (resulting in taxonomic misassignment)

Figure 2. Phylogenetic re-analysis of ISCU sequences in Tobiasson et al. showing alphaproteobacterial origin. Eukaryotic EP00668P041674 EPOC sequences coloured blue, eukaryotic EP00629P016341 EPOC sequences green, prokaryotic EPOC sequences black and sequences from previous studies red.